# Moored observations of mesoscale features in the Cape Basin: Characteristics and local impacts on water mass distributions

Marion Kersalé[1,2], Tarron Lamont[1,3], Sabrina Speich[4], Thierry Terre[5], Remi Laxenaire[4], Mike J. Roberts[3,6], Marcel A. van den Berg[3] and Isabelle J. Ansorge[1]

[1]Marine Research Institute, Department of Oceanography - University of Cape Town, Rondebosch, South Africa
[2]Cooperative Institute for Marine and Atmospheric Studies, University of Miami, and NOAA/Atlantic Oceanographic and Meteorological Laboratory, Miami, Florida
[3]Oceans and Coastal Research, Department of Environmental Affairs, South Africa
[4]Laboratoire de Météorologie Dynamique, UMR 8539 École Polytechnique, ENS, CNRS, Paris, France
[5]IFREMER, Univ. Brest, CNRS, IRD, Laboratoire d'Océanographie Physique et Spatiale (LOPS), IUEM, Plouzané, France
[6]Ocean Science & Marine Food Security, Nelson Mandela University, Port Elizabeth, South Africa

*Correspondence to*: Marion Kersalé (marion.kersale@noaa.gov)

**Abstract.** The eastern side of the South Atlantic Meridional overturning circulation Basin-wide Array (SAMBA) along 34.5°S is used to assess the nonlinear, mesoscale dynamics of the Cape Basin. This array presently consists of current meter moorings and bottom mounted Current and Pressure recording Inverted Echo Sounder (CPIES) deployed across the continental slope. These data, available from September 2014 to December 2015, combined with satellite altimetry allow us to investigate the characteristics and the impact of mesoscale dynamics on local water masses distribution and cross-validate the different data sets. We demonstrate that the moorings are affected by the complex dynamics of the Cape Basin involving Agulhas Rings, cyclonic eddies and anticyclonic eddies from the Agulhas Bank and the South Benguela upwelling front, and filaments. Our analyses show that exchange of water masses happens through the advection of water by mesoscale eddies but also via wide water mass intrusions engendered by the existence of intense dipoles. This complex dynamics induces strong intra-seasonal upper-ocean velocity variations and water mass exchanges between the shelf and the open ocean, but also across the subantarctic and subtropical waters. This work presents the first independent observations comparison between full-depth-moorings and CPIES data sets within the eastern South Atlantic region that gives some evidence of eastern boundary buoyancy anomalies associated with migrating eddies. It also highlights the need to continuously sample the full-water depth as inter-basin exchanges occur intermittently and affect the whole water column.

## 1 Introduction

Mesoscale nonlinear dynamics contribute to large-scale water mass distribution and therefore to the large-scale Meridional Overturning Circulation (MOC) (Gordon, 1985; Biastoch et al., 2008; van Sebille et al., 2012), as they redistribute momentum, heat, and mass between different regions (Robinson, 1983). Nonlinear mesoscale eddies, defined as nonlinear using the advective parameter criteria of Chelton et al. (2011), are found throughout the global oceans, but are very energetic when associated with western boundary currents. The Agulhas Current, the intense western boundary current of the South Indian Ocean, follows the African shelf edge southwestwards along the east coast. When it reaches the southern tip of Africa, at 34°S, the Agulhas Current leaves the continental slope and affected by strong instabilities retroflects at 38°S into the Indian Ocean (Lutjeharms and Cooper, 1996; de Ruijter et al., 1999; Lutjeharms, 2006). These instabilities are responsible for extensive meandering just southwest of South Africa, and shear edge features such as Agulhas rings, eddies and filaments form within this region. These nonlinear, mesoscale features control the Agulhas leakage—defined as the transport of warm and salty Indian Ocean waters into the Atlantic Ocean through the Cape Basin (labeled in Fig. 1; *e.g.*, Gordon et al., 1992; de Ruijter et al., 1999). The Agulhas leakage injects buoyancy anomalies that impact the Atlantic MOC (AMOC) strength, and the Atlantic Multi-decadal Oscillation, with clear implications for climate (*e.g.*, Beal et al., 2011; Biastoch et al., 2015).

Agulhas ring observations, collected from shipboard surveys, were first exploited to quantify the energy, heat and salt fluxes of these mesoscale features (*e.g.*, de Ruijter et al., 1999, and references therein). The first inter-ocean exchange estimates were based on individual hydrographic cruise data combined with altimetric tracking of Agulhas rings (Olson and Evans, 1986; Gordon and Haxby, 1990; Duncombe Rae, 1991; van Ballegooyen et al., 1994; Byrne et al., 1995). These combined data revealed the presence of 4 to 8 rings per year propagating into the South Atlantic with speeds ranging between 5 and 10 km day$^{-1}$, with a diameter up to 400 km and their influence is felt from the surface down to an intermediate depth between 600 and 1100 m. The associated inter-ocean volume transport per ring was estimated between 0.5 and 3 Sv (1 Sv=10$^6$ m$^3$ s$^{-1}$). These first estimates were based on statistics from the early constellation of altimetry satellites that resolved fewer features than the modern constellation, and individual ring observations, which led to a large spread of documented volume fluxes estimates.

A more accurate estimate of the fluxes in the Cape Basin has been accomplished by deploying a line of instruments across the pathway of Agulhas rings (BEST – Benguela Sources and Transports Experiment: Duncombe Rae et al., 1996; Garzoli and Gordon, 1996; Goni et al., 1997) and deep profiling floats (KAPEX – Cape of Good Hope Experiments: Lutjeharms et al., 1997; Richardson et al., 2003 and MARE – Mixing of Agulhas Rings Experiment: Van Aken et al., 2003). From these experiments, the northwestward propagation of the rings in the Cape Basin was characterized through a narrow ring corridor

(Garzoli and Gordon, 1996) or through three different routes on the basis of topographic effects (Dencausse et al., 2010).

Such measurements have also demonstrated the highly turbulent regime of the Cape Basin and the co-existence of cyclonic and anti-cyclonic eddies which can enhance horizontal mixing at intermediate depth (Boebel et al., 2003; Matano and Beier, 2003; Richardson and Garzoli, 2003; Hall and Lutjeharms, 2011; Arhan et al., 2011). Cyclonic eddies are about 50 km in diameter and are formed along the western coast of South Africa or inshore of the Agulhas Current (Lutjeharms et al., 2003). Tracked from altimetric data, recirculating plumes of warm water at the sea surface are often identified around these eddies.


The presence of several Agulhas rings and cyclonic eddies can induce the generation of dipole structures comparable to the Heton model of Hogg and Stommel (1985). These oceanic dipoles have been observed using satellite imagery thanks to their surface signature described as mushroom-like pattern in various regions, and have been characterized with numerical simulations and analytical theories (Ahlnäs et al., 1987; Hooker and Brown, 1994; Hooker et al., 1995; Millot and Taupier-

Letage, 2005; Pallàs-Sanz and Viúdez, 2007; Baker-Yeboah et al., 2010a). These counter-rotating eddies enhance horizontal transport of heat along isopycnals, particularly across frontal structures (Spall, 1995) and vertical diapycnal mixing. Related to dipole dynamics, filaments extending well below the thermocline (de Steur et al., 2004) have been observed during hydrographic cruises southwest of Madagascar and in the Cape Basin (de Ruijter et al., 2004; Whittle et al., 2008).

The mesoscale features in the Cape Basin can also interact with the eastern boundary flow regime (Dencausse et al., 2010). The connection between Agulhas rings and the South Benguela upwelling frontal zone has been evidenced by the propagation of cold filaments from the coastal upwelling system extending hundreds of kilometers offshore the Cape Peninsula (Duncombe Rae et al. 1992; Lutjeharms and Cooper, 1996). Eventually joining this coastal upwelling front, Agulhas filaments, originating from the Agulhas Current, have been observed in the upper part of the water column

(Lutjeharms and Cooper, 1996). They have been associated with a strong equatorward front jet, the Cape Peninsula jet (Bang, 1973; Gordon, 1986; Lutjeharms and Meeuwis, 1987; Nelson et al., 1998). This strong equatorward jet is located over the shelf-break, with a width of 20-30 km and can reach a maximum velocity of 1.2 m s$^{-1}$ (Shannon and Nelson, 1996; Veitch et al., 2017). Its dynamics is suggested to be linked with the local wind forcing during the upwelling season. However, this seasonality is still a subject of discussion as Boyd et al. (1992) show the occurrence of the jet beyond the

upwelling season. This jet has been also linked to anticyclonic circulations associated with frequent passages of Agulhas rings along the continental slope (Dencausse et al., 2010). The issue is further complicated by the regular generation of cyclonic eddies against the continental slope in the area (Penven et al., 2001). These cyclonic eddies have been recently associated with the intensification of a poleward undercurrent that is generally weak to nonexistent in the Cape Basin (Baker-Yeboah et al., 2010b).


Understanding the Cape Basin circulation in the context of this strong nonlinear, mesoscale variability is crucial for many studies. On regional scales, the circulation is important for water mass distribution, local dynamics, air-sea interactions and ecosystem assessments. For instance, the interaction of Agulhas rings and filaments can have crucial implications for cross-shelf exchanges and therefore productivity as water exported offshore is generally rich in nutrients and biota, due to the upwelling regime and the shelf edge jet transporting eggs and larvae alongshore. On large scales, the Cape Basin circulation influences the AMOC and climate. The need of observations in the subtropical South Atlantic at these different scales, led to the establishment of the South Atlantic MOC (SAMOC) international initiative to enhance further observing systems in this area (Garzoli and Matano, 2011). As part of the SAMOC initiative, several efforts to document the inflows of the two main paths of the upper limb of the AMOC have been undertaken across the Drake Passage (*e.g.*, Chereskin et al., 2009; Chidichimo et al., 2014; Donohue et al., 2016) and south of South Africa (*e.g.*, GoodHope line: Ansorge et al., 2005; Speich et al. 2007; Gladyshev et al., 2008; Swart et al., 2008; Hutchinson et al., 2016). A trans-basin AMOC array, South Atlantic MOC Basin-wide Array (SAMBA), which began as a pilot array in 2008-2009 (Speich et al., 2010; Meinen et al., 2013), continues to grow along 34.5°S, a crucial latitude to evaluate the MOC variability and the impact of inter-ocean exchanges (Perez et al., 2011; Schiermeier, 2013). SAMBA is a collaboration between Argentina, Brazil, France, South Africa and the United States, with moorings located on the western and eastern boundaries (Speich et al., 2010; Meinen et al., 2013; Ansorge et al., 2014; Meinen et al., 2018). Since the pilot deployment described in Meinen et al. (2013), the number of moorings on the boundaries has increased dramatically. The transport and water mass anomalies associated with the deep current and migrating eddies near the western boundary have been well studied (*e.g.*, Meinen et al., 2012, Meinen et al., 2017; Valla et al., 2018) but the eastern boundary anomalies have not yet been examined along 34.5°S.

In the framework of the SAMOC international project and national programmes (SANAP - South African National Antarctic Programme), our work focuses on the eastern part of the SAMBA array (hereafter SAMBA-east) offering an ideal position to observe and characterize mesoscale dynamics—a key link between Indian and Atlantic water exchanges (Fig. 1). The analysis of the SAMBA-east moored data sets, from September 2014 to December 2015, provides evidence of mesoscale features passing through the Cape Basin. Experimental evidences of advected mesoscale eddies through moored arrays along a constant line of latitude or longitude in various parts of the ocean have been presented (Ursella et al., 2011; Sutherland et al., 2011; de Jong et al., 2013) and used to estimate eddy parameters such as their strengths and sizes (Lilly and Rhines, 2002). The main focus of this study is to characterize such mesoscale structures in the Cape Basin region from different data sets obtained along the SAMBA-east array using the Lilly and Rhines (2002) technique, and to quantify their impact on local water mass distributions.

## 2 Data and Methods

The eastern part of the SAMBA array during September 2014 to December 2015, consisted of four full depth current meter moorings (hereafter full-depth-moorings), eight Current and Pressure recording Inverted Echo Sounder (CPIES), and two bottom mounted Acoustic Doppler Current Profiler (ADCP) moorings that were deployed from the shelf to near the Walvis
Ridge offshore along 34.5°S (Fig. 1). In this study, we will focus on the four full-depth-moorings (hereafter named M1, M2, M3 and M4) extending from the continental shelf edge, at 1121 m of depth, to 15°E, at 4474 m of depth, deployed in September 2014, by the Department of Environmental Affairs (DEA) and the University of Cape Town (UCT), South Africa from the R/V *Algoa*, and the four CPIES nearest to the full-depth-moorings (hereafter named P1, P2, P3 and P4), deployed in September 2013, by IFREMER, France and DEA, South Africa from the R/V *SA Agulhas II* (Fig. 1, Table I).


The characteristics of the moorings and CPIES used in this study are summarized in Table I. All full-depth-moorings (M1-M4) have an upward looking 75kHz RDI ADCP, deployed at the uppermost float at about 400-500 m of depth, set to sample the upper water column at hourly intervals. At selected depths along the mooring lines, SBE 37 MicroCat's, a high-accuracy conductivity and temperature recorder, were deployed. Some of these instruments also have pressure recorders and optical
oxygen sensors. The oxygen sensors will not be used in this study. All of the full-depth-mooring instruments (ADCP and SBE Microcat's) were still recording on recovery, with the exception of two SBE37 MicroCat's that stopped recording, in November 2014, due to low battery power. These two units were both from M4, at depths of 1340 m and 3985 m. The sampling period for the full-depth-mooring instruments was 1h. The collected data were tidally filtered using a second-order Butterworth filter with a 3-day cut-off period. CPIES recorded hourly measurements of round-trip acoustic travel time,
bottom pressure, and the velocity at 50 m above the sea-floor. All of the CPIES were recovered and re-deployed, with the exception of P3 which was unfortunately lost during the recovery. Typically, the pressure sensor of the CPIES exhibits either a linear or an exponential plus linear type of drift (Watts and Kontoyiannis, 1990; Donohue et al., 2010; Meinen et al., 2013). After removing the drift following these traditional methods, the same tidal filter has been applied on the CPIES data. The CPIES moorings records were sub-sampled to one value per day at noon UTC. An empirical lookup table for hydrographic
property profiles, known as the "Gravest Empirical Mode" (GEM) method (Meinen and Watts, 2000; Watts et al., 2001) was constructed from 2213 CTD and Argo profiles in the region (Fig. 2a). These two-dimensional look-up tables of temperature and salinity as functions of depth and travel time are shown in Fig. 2b,c. The scatter between the original hydrographic data and the GEM fields (Fig. 2d,e) provide an estimate of the accuracy. Following the new thermodynamic equation of seawater, all temperature and salinity referenced as conservative temperature [°C] and absolute salinity ($S_A$ [g kg$^{-1}$]), (IOC et al., 2010).
The temperature and salinity scatter is less than 1.5°C and 0.1 g kg$^{-1}$, respectively. In the deep ocean, the scatter around the GEM field is a larger fraction of the observed variability, however this is because the signals at depth are smaller relative to the noise, and not because the scatter around the GEM field itself is significantly larger. The GEM method has been

successfully applied in the eastern South Atlantic (Meinen et al., 2013; Meinen et al., 2018) and across the Agulhas Current off the southeast coast of South Africa (Beal et al., 2015; Elipot and Beal, 2018). Combining the measured travel time from the CPIES (Table I) with the GEM look-up tables produces daily full-water-column hydrographic profiles. We will primarily look at results from the full-depth moorings (Sections 3.1 - 3.3), and secondarily look at the CPIES moorings (Section 3.4). The set of records presented hereafter were all tidally-filtered as mentioned above.

Following Lilly and Rhines (2002), a set of metrics used to characterize eddies in mooring data were defined. The strength and size of an eddy can be defined by the peak azimuthal velocity (V), the "apparent" radius or half-width of the eddy chord passing by a mooring (X), the temporal center of the eddy ($t_0$), and of the eddy's influence on the mooring ($\Delta T$). For an advected eddy, the quantification of V needs a separation of the eddy flow from the mean advection flow (magnitude U and direction $\Theta$). The first estimate of V (hereafter $V_n$) is made maximizing the velocity difference over a time window encompassing the eddy and giving initial guesses for $t_0$ and $\Delta T$. The mean advection flow direction $\Theta$ is then estimated as it is expected to be perpendicular to the direction of $V_n$. A second measure of V (hereafter $V_U$) can be obtained by subtracting an estimate of the advecting flow. For this, two methods exist: the "angle" ($U_a$ and $\Theta_a$) and the "filtering" ($U_f$ and $\Theta_f$) techniques. The first method minimizes the angle difference between the observed eddy currents and the observed vertical shear. The second method is useful when the advection flow is large compared to the eddy, and estimates $U_f$ and $\Theta_f$ by low-pass filtered velocity time series at the eddy center $t_0$ using a Hanning window whose length is three times the eddy duration $\Delta T$. More details about these methods can be found in Lilly and Rhines (2002) and the Appendic C of Lilly et al. (2003). For the purpose of this study, the duration of the eddy ($\Delta T$) and the magnitude of the advection flow ($U_a$ and $U_f$) are used to estimate the "apparent" radius of the eddy: $X= \Delta T \times U$. Finally, the Rossby number can be assessed as $(2 \times |V|)/(X \times f)$. In a purely linear flow, the Rossby number is equal to zero and implies that the momentum equation is dominated by the geostrophic balance between the pressure gradient force and the Coriolis acceleration.

Hydrographic sampling has been conducted along the eastern section of the SAMBA transect during five cruises on board the R/V *SA Agulhas II* (September 2013, September 2014, and July 2015) and the R/V *Algoa* (September 2014, and December 2015). During each cruise, Conductivity-Temperature-Depth (CTD) casts were carried out at several stations along the transect using multiple SeaBird Electronics SBE 911+ CTD systems and sensors, in order to measure temperature and conductivity. Conductivity was used to derive salinity, and then, combined with temperature, to determine density. CTDs were conducted from the surface to within 5-10 m of the sea floor on the R/V *SA Agulhas II*, with generally five casts offshore (0°E - 15°E) and five casts over the continental slope (15°E-17°E35'). CTDs on the R/V *Algoa* were conducted to a maximum depth of 1000 m, with 8 casts over the continental slope (15°E-17°E35') and zonal resolution of 0.3-0.5°. Discrete seawater samples were collected at selected depths for analysis of salinity with a Guildline Portsal salinometer (van den Berg, 2015) and used after the cruise to calibrate the CTD conductivity sensors. Following the procedure of Kanzow et al.

(2016), the SBE MicroCat's sensors were attached to the rosette and data was recorded simultaneously between the surface and 1000 m depth due to the limitation of the vessel. As the CTD sensors on the R/V *Algoa* had not been recently calibrated by the manufacturer, and the surface mixed layer did not provide a stable environment to assess the differences, the correction coefficients applied to the MicroCat's data can introduce additional bias. To assess the accuracy of these measurements, we compared the MicroCat's data with World Ocean Atlas (WOA) 2013 climatology profiles (Locarnini et al., 2013; Zweng et al., 2013). We compared the mean value of temperature and absolute salinity between the MicroCat's records and the climatology profiles below 2000 m, to avoid depth-dependent offsets to the comparison due to strong vertical gradients in the thermocline. The mean differences of 0.14°C for the temperature and -0.01 g kg$^{-1}$ for the absolute salinity were estimated between WOA and the MicroCat's below 2000 m. There differences are large, one or two orders of magnitude larger than the instrument accuracies, but they provide an upper bound for the biases in the MicroCat's data in lieu of calibrated CTD sensors. In any case, these mean differences are small compared with the robust signals analyzed in this study.

Satellite imagery was analyzed to give an overview of the surface dynamics in the Cape Basin region. Sea Surface Temperature (SST) was derived from ODYSSEA, a Group for High Resolution Sea Surface Temperature (GHRSST) regional product interpolated on a 0.02° grid for the South African area (Piollé and Autret, 2011; http://cersat.ifremer.fr/data/). The Sea Surface Height (SSH) and geostrophic velocity fields are derived from the Delayed Time Maps of Absolute Dynamic Topography (MADT) mapped daily on a 1/4° Mercator grid (Pujol et al., 2016). An eddy detection method based on the algorithm developed by Chaigneau et al. (2008, 2009) was applied to these ADT fields. This new method detects local ADT extrema to identify potential eddy centers. For each potential eddy centers, the algorithm looks for closed ADT contour lines with a increment of plus or minus 1 mm (*e.g.*, Chaigneau et al., 2011). This 1 mm threshold step is a bit smaller than the typical value used in the literature, however recent work has shown that it leads to more accurate eddy sizes and amplitudes (Faghmous et al., 2015). This criterion prevents the misidentification of large eddies as two smaller ones (Chaigneau et al., 2008, 2009). The spatially largest closed ADT contour line encompassing one extrema corresponds to the eddy edge. Following the filtering criteria of Faghmous et al. (2015) for regional study, we only consider eddies with a minimum lifetime of 7 days. The method provides the radius ($R_{out}$) and the amplitude of the eddies associated with the outermost contour of ADT. In addition, the eddy ADT contour along which the norm of the azimuthal geostrophic velocity is the highest ($V_{max}$) is calculated to provide the radius of the coherent core of the eddy ($R_V$; *e.g.* Nencioli et al., 2010; Chelton et al., 2011). We validated our results with the well-known Mesoscale Eddy Trajectory Atlas (META) product produced recently by CNES/CLS in the DUACS system and distributed by AVISO which is based on an eddy-tracking methodology developed by Chelton et al. (2011). The main difference between the META eddy-tracking and our methodology is the use of an improved eddy-tracking method which takes into account both merging and splitting events. In both products, the method is based on the low displacement of eddies compared to their size. Each eddy was

tracked within the boundary of the eddy defined at the previous time step following the method of Pegliasco et al. (2015). In our version, no cost function was applied to the eddies in cases of multiple association but a minimum of overlapping surfaces is added. Consequently, splitting events are identified when one eddy is associated to two or more eddies in the next day as well as merging events in the reverse case. The eddies that are discussed in the present paper are all identifiable by both the new method and by the META product. In this study, the eddy statistics are based on our method because it better tracks splitting/merging of eddies, and we only show the META results to validate the new methodology.

## 3 Results

### 3.1 Upper ocean properties and dynamics

To characterize the hydrodynamical properties of the mesoscale features passing through the SAMBA-east array, the measurements of the upward-looking ADCP are analyzed together with concurrent altimetry data. From September 2014 to December 2015, the mean and the standard deviation of the near surface velocities recorded at the shallowest depth (between 40 and 60 m depth) by the upward-looking ADCP is between $24.6 \pm 11.4$ cm s$^{-1}$ at the mooring nearest to the shelf (M1 - thin blue line, Fig. 3a,b) and $29.9 \pm 22.0$ cm s$^{-1}$ at the offshore mooring (M4 - thin red line, Fig. 3g,h). Typically, the shallowest depth sampled was near the surface but below the Ekman layer (mean value of about 36 m along the section) allowing a comparison with the surface geostrophic velocity derived from satellite altimetry. Statistical comparisons between M1-M4 ADCP near-surface velocities and the surface geostrophic velocity derived from satellite altimetry have been made (Table II, Fig. 3). The comparison has been undertaken mostly in terms of correlation and rms differences between the zonal and meridional components of M1-M4 ADCPs ($u_m$, $v_m$) and those from altimetry at the nearest gridded location ($u_{alti}$, $v_{alti}$):

$$\Delta V_{rms} = \left[ \frac{1}{N} \sum \left[ \left( V_m - \overline{V_m} \right) - \left( V_{alti} - \overline{V_{alti}} \right) \right]^2 \right]^{1/2} \quad (1)$$

with $V$ the zonal or the meridional components of the velocity and $m$ the index of the mooring. Additionally, comparisons were made by computing the bias (Eq. 2).

$$bias = \overline{V_m} - \overline{V_{alti}} \quad (2)$$

Correlations between ADCP and altimetry velocity estimates (R) are significant and fell in the range 0.29-0.83 (Table II). To determine the significance of the correlations, the number of independent samples, also known as the number of degrees of freedom (Thomson and Emery, 2014), is estimated by dividing the record length by twice the integral time scale calculated from the variability of each respective quantity (~40-50 days). The integral time scale is estimated from the integral of the auto-correlation function to its first zero-crossing (see Appendix B in Meinen et al., 2009 for more information). $\Delta V_{rms}$ varied

from 12.9 to 15.9 cm s$^{-1}$ (Table II). The absolute values of the biases were typically between 1 and 6 cm s$^{-1}$, with a maximum of 8.8 cm s$^{-1}$ at M2. Correlation between the u-component of these velocity measurements showed values greater than 0.7 for the three offshore moorings. Weak correlation coefficients for both components were observed for the mooring closest to the shore (M1, 1121 m of depth). The coefficient for the v-component gradually increases across the continental slope, moving away from the shelf. The correlations for sites away from the shelf are significant considering the mean zonal correlation scales of the satellite product of 150 km at that latitude (Pujol et al., 2016). The poor correlation for both velocity components at M1 can be mainly attributed to its position too close to the coast (~160 km off shore, at the shelf break), and therefore embedded in a different dynamical regime than purely geostrophic. The comparisons reveal that satellite data provides a reasonable description of the upper ocean circulation (between 40 and 60 m depth) across the continental slope along our mooring arrays, except near the shelf-break (i.e., well inshore of the 1200 m isobath).

In light of these results, the dynamics inferred from satellite altimetry can provide a basis for understanding the variability of the zonal and meridional components of the upper-layer velocity between 40 and 60 m depth for the offshore moorings (M2 to M4). Using the new eddy detection technique, we can first estimate statistical characteristics of mesoscale eddies passing through the mooring line during the measurement period of the full-depth-moorings along 34.5°S (September 2014 to December 2015). A total number of 16 Agulhas rings, defined as anticyclones that enter the Cape Basin crossing the C-line (Fig. 1), were detected and confirmed by the META product. This line extends from the southernmost tip of Africa (Cape Agulhas) and, after crossing various seamounts, ends at 45°S in the Southern Ocean. From our altimetric tracking, the median radius and standard deviation of these features are equal to 85 ± 43 km for $R_{out}$ and 66 ± 38 km for $R_V$. The azimuthal speed of these Agulhas rings ($V_{max}$) is equal to 0.49 ± 0.24 m s$^{-1}$ with a translation speed, mainly northwestward of 11 ± 6 km day$^{-1}$. Considering our observing system along the SAMBA-east line, 7 anticyclonic and 6 cyclonic eddies influenced the mooring measurements during the ~14 months period of recording at sites M2 (P2) to M4 (P4) (Fig. 4, Table III). All these mesoscale features passed over one of the moorings considering their outermost contours of ADT, and have a closed contour in the satellite dynamic-height amplitude of at least 1 cm. Two of these eddies (anticyclonic A4 and cyclonic C4) were generated at the Agulhas Retroflection and propagated into the Cape Basin through the northern route defined by Dencausse et al. (2010) (dashed black line - Fig. 4a). These two eddies have the largest azimuthal speed ($V_{max}$) compared with other eddies detected in the area (Table III). Four of the anticyclonic eddies (A1, A2, A5 and A7) observed in this study are generated by the splitting of an Agulhas ring and two are generated north of the SAMBA line (A3 and A6). Cyclonic eddy C5 is generated by the splitting of C4 and the other four cyclonic eddies are generated over the slope off the Cape Peninsula.

The currents measured during September 2014-December 2015 in the top 500 m of the water column contain a number of sudden rotational events, seen in the time series of the current vector stick-plot (Fig. 5). These transitions in the velocity field are associated with mesoscale eddies passing through the mooring line. The altimetry data allows us to examine the impact

of 6 cyclonic (blue shaded areas - Fig. 5) and 7 anticyclonic eddies (red shaded areas - Fig. 5) on zonal and meridional velocity at the three mooring sites. Moreover, the data records show that the presence of one dipole (counter-rotating eddies) affects the velocity measurements (yellow shaded area - Fig. 5). The presence of this dipole is confirmed in the ODYSSEA SST field by its mushroom-like pattern, the typical surface signature of these features.

To have more of a consistent and complete picture of the mesoscale dynamics in the Cape Basin, the position of these eddies observed by satellite altimetry were analyzed in relation to variations in the ODYSSEA SST field (Fig. 6). On December 1, 2014, both eddy identification methods detect one cyclonic eddy (C1) and one anticyclonic eddy (A1) close enough to the mooring line to affect the measurements (Fig. 6a). Approximately one month later (Fig. 6b), C1 and A1 move westward. At that time, a new cyclonic eddy (C2) was generated on the slope, at the South Benguela upwelling front, and a new anticyclonic eddy (A2) by the splitting of A1.

On February 20, 2015 (Fig. 6c), an anticyclonic eddy (A3) is present between M3 and M4 and a new cyclonic eddy (C3) is generated at 35°S. This cyclonic eddy merges afterwards with an intense cyclonic eddy (C4) coming from the Agulhas retroflection. After the merging, the cyclonic eddies C3 and C4 (called C4) affect the measurements of M2-M3 until the end of March (Fig. 5c,d). This strong northeastward current is also intensified by an intense cross-shelf density front that is enhanced at that time due to an upwelling event and the northward migration of an Agulhas ring A4 (Fig. 6d). On March 21, 2015, A4 splits to generate A5. This anticyclonic eddy was close enough to M4 to induce the northward propagation of a warm filament on April 9, 2015 (Fig. 6e).

At the end of April, an intense dipole is observed due to the interaction of eddies A5 and C5. Note that cyclonic eddy C5 is generated by the splitting of C4 on April 18, 2015. The dipole (A5, C5) induced a strong northward current that injected cold surface water across the mooring array in April 25 (Fig. 6f) and warmer Agulhas Current water during the month of May (Fig. 6g).

After this intense event, the satellite imagery still reveals the presence of cyclonic eddy C5 and a new anticylonic eddy (A6) in July between M3 and M4 (Fig. 6h). On September 15, 2015 (Fig. 6i), the presence of a cyclonic eddy (C6) over the mooring line and an anticyclonic eddy (A7) generate a warm filament propagating toward M4.

In summary, the moorings over the slope (M2 and M3) are affected by five cyclonic eddies identified by the two methods of eddies detection (Table III, Fig. 5 and Fig. 6) over the measurement period from September 2014 to December 2015. The largest velocity perturbations over the slope are associated with cyclonic eddy C4. M4 records show the influence of many anticyclonic eddies and three cyclonic eddies. Largest velocity perturbations are seen at M4 during the presence of the dipole

A5 and C5. The presence of several of these mesoscale features induces the propagation of two warm filaments and two injections of relatively colder surface water and relatively warmer Agulhas Current water.

**3.2 Case studies - Upper water column**

From the combined analysis of satellite and mooring data in section 3.2, we selected the period between March and June 2015 to analyze eddy-like features, filaments, and strong intrusions of cold water or Agulhas Current water due to eddy-eddy interactions. We focus on the variability at M4 which exhibits the strongest variations in responses to passing mesoscale features.

Following the method of Lilly and Rhines (2002), the coherent eddies, dipoles or filaments in mooring data can be detected and assessed in a more quantitative fashion. The zonal and meridional components of the velocity measured at the mooring can be placed on a hodograph plane (Fig. 7a,b,c). In this plane, a combination of a straight line with a segment of a circle (called D-shaped) reflects an eddy structure. The nearer the eddy's center passes close to the mooring, the more the

hodograph appears as a straight line. An eddy sliced through its exact center or a filament lead to the same type of hodograph (straight line). Eddies can be further distinguished from fronts and filaments using progressive vector diagrams (PVD) (Fig. 7d,e,f). The eddy structure presents bends unlike fronts or filaments which result in straight lines. Concerning the hodographs for a dipole, it is similar to that of an eddy if one of the cores passes over the mooring. If the mooring measures the velocity in between the two cores, the hodograph shows a pulse with strong increase and decrease of the currents.

In accordance with the Lilly and Rhines (2002) detection method, we have isolated three case studies between March and June 2015. Each case study is associated with one feature which can generate different events. The first case study focused on anticyclonic eddy A4, its splitting which generates A5 and the propagation of a filament along its border. The second case study detailed the dynamics of cyclonic eddy C5, and the third one described the dynamics of a dipole (A5/C5) generating

two current pulses.

The first case study between March 9 and April 6, 2015 (Fig. 7a,d) illustrate the presence of anticyclonic eddies A4 and A5. The D-shape structures on the hodograph (Fig. 7a) and the bends to the right associated with anticylonic eddies on the PVD (Fig. 7d) appear clearly. To better illustrate these events, the temperature recorded at the first SBE Microcat's (depth between

430 and 750 m) de-meaned over the time period of each case study are presented. Within the period during which a positive temperature anomaly associated with the anticyclonic eddies appears in the record, the hodograph tends to be relatively straight. At the end of the time period, the impact of a filament propagating northward due to the presence of A5 is observed as a straight line on both planes (Fig. 7a, d). The second hodograph and PVD between May 28 and June 10, 2015 (Fig.7 b,e) display a D-shaped structure and a bend to the left associated with a negative temperature anomaly typical of a cyclonic

eddy. According to its time of occurrence, this mesoscale feature corresponds to C5. Finally, the final case study between April 8 and May 29, 2015 (Fig. 7c,f) exhibits very strong currents with two acceleration and deceleration phases that are associated with a dipole.

Following these three identified cases studies, the vertical–temporal structure of eddy-like features and filaments, and strong intrusions of cold water or Agulhas Current water due to eddy-eddy interactions are analyzed. Moreover, the estimated sizes and strength of eddies (A4, A5 and C5) can be defined (Table IV) according to the Lilly and Rhines (2002) method detailed in Section 2.

Case of Anticyclonic Eddies: From the middle to the end of March (Fig. 8a,b), M4 is impacted by Agulhas rings A4 and A5. The vertical-temporal section of the u- and v-component of the current speed (Fig. 8c,d) shows an eddy-like structures on March 14 and March 24, 2015. The vertical structure of the first Agulhas ring A4 is not evident as it is associated with an advecting flow directed to the southwest ($\Theta$ – Table IV). For the second anticyclonic eddy, A5, the direction of the mean flow shifts toward the west ($\Theta$ – Table IV) and so the eddy-like structure is much clearer in the perpendicular direction (v-component, Fig. 8d). The azimuthal eddy velocities reach 26.0 cm s$^{-1}$ for A4 and 21.7 cm s$^{-1}$ for A5 ($V_U$ – Table IV) from the surface to 200-250m depth. During this period, an increase of temperature (salinity) of 2.4°C (0.35 g kg$^{-1}$) is recorded at the first SBE37 MicroCat's between 430 and 450 m depth (Fig. 8e).  We can also evaluate a down-shift of the isopycnal layer of about 200 m for both eddies (Fig. 8f). This quantity is found by interpolating the potential density within the mean vertical profile derived from CTDs on the R/V *SA Agulhas II*. At the end of the time period (April 2), the propagation of a warm filament at the eastern side of A5 is identified. The measured v-component of the velocity (Fig. 8d) show at that time a strong northward flow. From the satellite altimetry (colored circles at the top of Fig. 8c,d), the velocity component magnitudes are lower than the ones from the upward-looking ADCP but show similar order changes.

Case of Cyclonic Eddy: The same analysis is undertaken for a second event occurring between May 28 and June 10, 2015. On June 3, 2015 (Fig. 9a), a cyclonic eddy (C5) affects the circulation along the SAMBA line. At this date, the vertical-temporal section of the u-component of the current speed (Fig. 9b) shows an eddy-like structure. During the time period the eddy passes across the mooring, a strong northward flow is recorded (v-component – Fig. 9c, $\Theta$ and U– Table IV).  The core of this strong cyclonic eddy (-20.0 cm s$^{-1}$ azimuthal velocity) extends from the surface down to 200 m depth. A decrease of salinity and temperature between 470 and 500 m depth of 0.13 g kg$^{-1}$ and 1.7°C respectively is recorded (Fig. 9d). This decrease can be partly explained by the 100 m uplift of the isopycnal layer (Fig. 9e). From the satellite altimetry (colored circles at the top of Fig. 9b,c), the meridional component of the current speed is in good agreement with the upward-looking ADCP compared to the zonal that shows a lower velocity.

Intrusions - Dipole dynamics: From the middle of April to the end of May (Fig. 10), anticyclonic eddy A5 and cyclonic eddy C5 affect the circulation around M4 as a dipole. This intense dipole induces two intense northward pulses of current with meridional velocities exceeding 100 cm s$^{-1}$ on April 22 and May 14 throughout the entire depth of the sampled water column. These two pulses last about 22 and 15 days respectively. Their vertical influence is deeper than 600 m depth. The vertical mooring motion as evidenced by the downward shifts in the range of depths resolved by the ADCP is very intense during these two events. Although depth changes during these events (~ 300 m) were substantial, the minimal variations in pitch and roll (maximum change of 5 and 4 degrees, respectively), were well within accepted limits and indicated that the performance of the mooring is satisfactory. The altimetry data show the same intense northward current at the surface, however there is a pronounced lag of 7 days for the second intrusion. This lag is no longer present if we consider the altimetric velocity one grid eastward (0.25° of resolution). These two pulses of currents impacts the measurements at the first SBE Microcat's sensor between 435 and 700 m depth (Fig. 10e,f) with large variations of temperature (more than 6°C), salinity (~0.7  g kg$^{-1}$) and a down-shift of isopycnal layers of about 150-200 m.

Note, the temperature and salinity anomalies for all three of these cases are much larger than the upper bound of the MicroCat's temperature and salinity biases, 0.14°C and 0.01 g kg$^{-1}$, documented in Section 2.

### 3.3 Full water column water masses distribution and variability

We used the daily-averaged temperature and salinity data obtained from SBE37 MicroCat's instruments on moorings to recover the regional water masses present at M4. Similar to Lamont et al. (2015), the distribution of water masses was determined according to conservative temperature, absolute salinity, and density layers, as illustrated in Fig. 11a and described in Table V. Modified Upwelled Water (MUW) was defined according to Duncombe Rae (2005) as central shelf water upwelled along the coast and modified due to solar heating and freshwater flux. Oceanic Surface Water (OSW) was defined with the criteria of Donners et al. (2005) as salinity maximum water subducting in the western tropical Atlantic. The criteria of Donners et al. (2005) were also used to define light South Atlantic Central Water (lSACW - defined as Indian Central Water brought into the South Atlantic Ocean by Agulhas Current intrusions), South Atlantic Subtropical Mode Water (SASTMW), and Subantarctic Mode Water (SAMW- with a vertical temperature gradient less than 1.6 °C/100 m (Roemmich and Cornuelle, 1992)). Local ventilation of Indian Central waters have been firstly identified by Arhan et al. (2011) inside different sampled Agulhas rings (Arhan et al, 1999;  Gladyshev et al., 2008). This water mass has been recently defined in the study of Capuano et al. (2018) as Agulhas Ring Mode Water (ARMW) including in addition to the above two other Agulhas rings previously sampled (Ducombe Rae et al., 1996; McDonagh et al., 1999). Three different varieties of Antarctic Intermediate Water (AAIW), namely Indian AAIW (I-AAIW), Indo-Atlantic AAIW (IA-AAIW) and Atlantic AAIW (A-AAIW) were characterized according to Rusciano et al. (2012). The highest salinity values in the I-AAIW variety are likely associated with Red Sea Intermediate Water (RSIW) which have been shown traveling down the Agulhas current as

discontinuous filaments, or confined within anticyclonic and cyclonic eddies (Roman and Lutjeharms, 2007). Upper Circumpolar Deep Water (UCDW), North Atlantic Deep Water (NADW), and Lower Circumpolar Deep Water (LCDW) were defined according to Heywood and King (2002).

Overall, the vertical distribution of water masses at M4 (colored dots - Fig. 11a) shows that SAMW is present on the
415 southwestern African continental slope around 500 m depth, I-AAIW and IA-AAIW between 500 and 1000 m, UCDW from 1000 to 1500 m depth, NADW between 1600 and 3000 m, and finally LCDW below 3000 m.

The mean vertical and zonal distributions are affected by the regional mesoscale dynamic described in the previous section. Typically a cyclonic (anticyclonic) eddy causes uplift (suppression) of isopycnal layers. The temperature and salinity
relationship is highlighted for these two types of events. During the period of the anticyclonic eddies, the temperature and salinity, and density values at the shallowest SBE Microcat sensor at M4 (red dots - Fig. 11b) show the highest range of values of the all of the time series records. During the time period of the cyclonic eddy, the values are at the opposite end, i.e. the lowest recorded densities (blue dots - Fig. 11b). While the signature of these two features is clearly separated for the two shallowest SBE Microcat sensors, we do not definitively prove if these changes are associated with a thermohaline anomaly
or a simple heave. During the two successive intrusions of waters due to the presence of the dipole at M4, the temperature and salinity relationship (orange and green dots - Fig. 10c) highlights a wide range of values. For this type of event the deepest sensors recorded water masses characteristics different than the ones usually sampled over the time period (Fig. 11a). The pressure record from the sensor at M4 exhibits the largest vertical variation during the time period of the dipole (Fig. 10c,d). This large vertical mooring motion adds another level of complexity to interpreting this relationship. To understand
the origin of these variations, which can be associated with water trapped inside or around the eddies, vertical movement of isopycnal layers and/or mooring motions, the full-water column characteristics will be analyzed with different data sets onto neutral density surfaces.

### 3.4 Full-water-column analysis – Focus on the three case studies

The different data sets allow us to analyze in more detail the effects of the mesoscale features described in the previous
sections and successfully cross-validate the measurements made by the different types of moored instruments (full-depth-moorings *vs.* CPIES).

The profiles estimated via the GEM method can be compared with the single point values of temperature and salinity from M4 sensors (8 sensors recording over the common time period, from September 20, 2014 to August 11, 2015). The
440 reconstructed field captures the major changes in temperature and salinity variability in the upper water column. Indeed, significant correlation coefficients between these two independent data sets show values higher than 0.93 for salinity and

temperature at the shallowest sensors (~ 450, 900 m depth). At the deepest levels, the correlation coefficients range between 0.14 and 0.83 and they are not significant at the 95% confidence level. This can be due to the small variability at the deepest levels compared to the scatter around the GEM field (Fig. 2) and the correction applied to some of the Microcat's data without pressure recorder during intense vertical mooring motion.

Measurements from these two data sets are compared for the three case studies described in Section 3.3. The conservative temperature and absolute salinity anomalies at M4 due to the presence of the eddies (anticyclone and cyclone: A5 and C5, Fig. 12a,b) and the intrusion of water that arise because of the presence of the dipole (Fig. 12c,d) are calculated relative to the hydrographic properties in "normal conditions" (just before each event occurred).

During the occupancy of A5 at M4, defined as the temporal range from $t_0$ - $3\Delta T$ to $t_0$ (Table IV - March 18-24, 2015), the measurements at the SBE37 MicroCat's show an increase of temperature of 1.54ºC and salinity of 0.22 g kg$^{-1}$ along the 26.73 kg m$^{-3}$ neutral surface (Fig. 12a,b - red dots). The second sensor recorded a temperature anomaly of 0.48ºC. No significant anomaly is recorded for the salinity along the 27.33 kg m$^{-3}$ neutral surface. The reconstructed temperature profile via the GEM method in the upper part of the water column captures a maximum of temperature anomaly of 1.42ºC along the 27.03 kg m$^{-3}$ neutral surface and salinity of 0.17 g kg$^{-1}$ along the 26.7 kg m$^{-3}$ neutral surface (Fig. 12a,b - red line) which is relatively well captured with the single point measurements of the full-depth-mooring. Negligible anomalies are detected in the deeper part of the water column with neutral density larger than 27.7 kg m$^{-3}$.

During the time period of C5, defined as before from $t_0$ - $3\Delta T$ to $t_0$ (Table IV – May 31 - June 3, 2015), the shallowest SBE37 MicroCat records a negative temperature (-1.43ºC) and salinity (-0.17 g kg$^{-1}$) anomalies (Fig. 12a,b - blue dots). Along the 27.55 kg m$^{-3}$ neutral surface, the second sensor records saltier water. Anomalies of 0.04 g kg$^{-1}$ for the salinity are characteristics of I-AAIW. The hydrographic data, estimated from the CPIES, show anomalies associated with this feature of the same order. The transition between the different sign of salinity anomalies is evidenced at 27.35 kg m$^{-3}$. Negligible anomalies are detected in the deeper part of the water column.

For both features (cyclonic and anticyclonic eddies), the similar temperature and salinity anomalies between the two data sets in the upper water column (26 – 27.7 kg m$^{-3}$) provide evidence for the presence of different water masses trapped inside the observed mesoscale features.

During the two intrusions of water due to the presence of the dipole (first intrusion: April 11 - 22, 2015; second intrusion: May 7-14, 2015 – green and orange dots, Fig. 12c,d), the temperature and salinity show large anomalies at all neutral densities sampled. The largest anomalies are recorded at the first SBE37 MicroCat's for both intrusions. During the second

intrusion, a warmer ($\Delta T$= 4.03ºC) and saltier ($\Delta S$= 0.41 g kg$^{-1}$) water of Indian origins appears in the upper part of the water column (<27 kg m$^{-3}$ neutral surface). Negative anomalies during the first intrusion are very likely associated with water of Subantarctic origins (anomalies of -5.19ºC for temperature and -0.56 g kg$^{-1}$ for salinity). As for the cyclonic eddy, during the cold intrusion the presence of high salinity I-AAIW (0.28 g kg$^{-1}$ in salinity) is observed along the 27.52 kg m$^{-3}$ neutral surface. Interestingly from the SBE37 MicroCat's, we observe strong anomalies of salinity at the lowest neutral surface (>28 kg m$^{-3}$). The presence of such saltier ($\Delta S$=0.16 g kg$^{-1}$) water anomalies at these depths reveal the presence of large pulses of NADW. This event influences not only the upper layers waters but its also affects most of the water column, down to the bottom.

From the reconstructed CPIES fields, the temperature and salinity signatures of the two intrusions (Fig. 12c,d – green and orange lines) show anomalies of the same sign but two times smaller than the ones recorded at the MicroCat's. During these intrusions, the vertical mooring motion is very intense and is larger than the isopycnal displacement. This result is supported by the large range of neutral densities sampled by the SBE37 MicroCat's (colored boxes, Fig. 12c,d). This range is even larger for the first two sensors, as at these shallow depths the vertical gradients of salinity and temperature are larger than horizontal gradients.

**4 Discussion and concluding remarks**

Since 2010, several efforts have been undertaken to enhance further the AMOC observing in the South Atlantic. This strategic monitoring system continues to grow along 34.5°S, a crucial latitude to evaluate the AMOC variability and the impact of inter-ocean exchanges (Drijfhout et al., 2011). As a consequence of the limitations of high spatial and temporal resolution in *in situ* observations, the quantification of inter-ocean exchange is still an ongoing work and many key questions and issues remain open such as what are the characteristics of mesoscale structures and what is their impact on local water masses exchange and distribution. In the framework of the SAMOC initiative, we provide here further investigation using a combination of satellite altimetry, full-depth-moorings measurements and CPIES records.

Focusing on the SAMBA-east array region, the general circulation around South Africa has been rather well described in previous studies. Two main processes have been observed to influence this area: an equatorward shelf-break frontal jet off the Cape Peninsula (Lutjeharms and Meeuwis, 1987; Nelson et al., 1998) and the instabilities of the Agulhas Current responsible for the spawning of mesoscale eddies propagating into the Cape Basin (Lutjeharms and Cooper, 1996; de Ruijter et al., 1999; Lutjeharms, 2006).

During the time period of our study (~14 months), we  used a newly developed method to identify a number of eddies in our region using satellite imagery; these eddies where then confirmed to also be present in the well-known META atlas (Chelton et al., 2011). Further analysis showed that 16 large eddies were Agulhas rings, and the satellite data show that one of these rings later passes through the SAMBA-east array during the mooring time period, as well as four anticyclonic eddies generated by the splitting of one of the rings. The Agulhas ring statistics (diameter of 170 ± 86 km with translation speeds of

11 ± 6 km day$^{-1}$) are in good agreement with previous estimates (diameter of 200-400 km – *e.g.* Arhan et al, 1999; translation speed of 2.9-7.3 km day$^{-1}$ *e.g.* Olson and Evans, 1986; Byrne et al., 1995; Goni et al. 1997; Schouten et al. 2000). The comparisons reveal that satellite data provides a reasonable description of the upper ocean circulation along the continental slope at our mooring locations, except near the shelf-break (i.e., along and inshore of the 1200 m isobath).

Analyses of both satellite and mooring data show that the eastern mooring array is strongly affected by the intense regional mesoscale variability. Previous studies (Arhan et al., 1999, 2011; Schouten et al., 2000; Boebel et al., 2003) have shown that Agulhas rings coexist with cold-core (cyclonic) eddies, which can contribute directly to the input of Indian water in the Atlantic (Lutjeharms et al., 2003; Arhan et al., 2011; Capuano et al., 2018). The resulting interaction of cyclonic and anticyclonic eddies can be also responsible for the extraction of warm Agulhas water filaments (Lutjeharms and Cooper,

1996; Whittle et al., 2008). These filaments may provide not more than 15% of the total mass flux of Indian Ocean waters into the South Atlantic (Lutjeharms and Cooper, 1996).

Over the measurement period from September 2014 to December 2015, the slope moorings (M2 and M3) show to be affected essentially by cyclonic eddies of different origins. Indeed, these moorings are affected by one cyclonic eddy

generated at the Agulhas Bank (C4), one by the splitting of C4, and four along the South Benguela upwelling front. The off-shore mooring (M4) is affected by the more complex dynamics characterizing the Cape Basin involving five Agulhas Rings, and two anticyclonic and cyclonic eddies both generated along the South Benguela upwelling front. The propagation of two warm surface filaments have been highlighted. The presence of these several mesoscale features induce intense intra-seasonal upper-ocean velocity variations and water masses exchanges across both, the shelf and the open ocean and between

the subantarctic and the subtropical frontal zones.

Our study indicates that exchanges of water masses across the continental slope happen through water advection not only via mesoscale eddies but also wide filaments engendered by the interaction among eddies, and in particular, through the existence of intense dipoles. As illustrated in previous studies, such filaments can extend well into the thermocline and can

be related to dipole dynamics (de Steur et al., 2004; Baker-Yeboah et al., 2010b). These wide intrusions cause intense north-northwestward currents affecting the whole water column. These injections are different from ordinary filaments, which exhibit a much smaller vertical extension (around 300 m of the water column). Among the different processes observed

along the SAMBA-east array, the most significant event is the intrusion of waters of Indian Ocean and subantarctic origin due to the presence of intense dipole.


In terms of number of occurrence of each type of event during the 14 months of record, we account for two intrusions of waters associated with the presence of a dipole, five Agulhas rings, six cyclonic eddies, two anticyclonic eddies and two warm filaments. Our work suggests that not only the advection of water within Agulhas rings or cyclonic eddies is important but that also dipole intrusions and filaments have a significant impact on the total mass, heat, and salt fluxes and therefore,

they all need to be better accounted for.

The presence of eddies, filaments and interaction of cyclonic and anticyclonic eddies have been also described in more detail in this study with our three case studies. Following the Lilly and Rhines (2002) method, an assessment of coherent eddies, dipoles and filaments in mooring data have been achieved. This method allowed us to evaluate the eddy parameters, such as

the eddy apparent radius, the direction of the mean flow, and the Rossby number, all essential elements to characterize eddies in a single point measurements. The estimation of small Rossby number (~0.1) associated to these features reveals that eddies are not highly nonlinear or ageostrophic by this measure, but the features are nonlinear in the sense that the corresponding Rossby number is not nil, so some ageostrophic processes are occurring. The momentum equation is then dominated by the quasi-geostrophic balance between the pressure gradient and the Coriolis forces and implies that altimetry

data is adequate for investigating the dynamics of the observed mesoscale features. From altimetry data, these eddies have a maximal azimuthal velocity exceeding their translation speed, confirming that the observed mesoscale features are nonlinear by the metric of Chelton et al. (2011). This definition is maybe the most pertinent in the context of this study, since it determines the ability of the observed mesoscale features to advect a parcel of trapped fluid as they translate (Fierl, 1981).

During the first two case studies, the typical impact of cyclonic and anticyclonic eddies causing, respectively, an uplift and downward motion of isopycnal layers are revealed. For the dipole case study, a downward motion of ispoycnal layers is recorded associated with a vertical movement of the mooring.

The different properties of each type event (cyclonic eddy *vs.* anticyclonic eddy; cold *vs.* warm dipole intrusions) have been

compared between full-depth-moorings and CPIES measurements in density space allowing a better characterization of the full-water column hydrographic properties and the opportunity to distinguish the changes of temperature and salinity due to vertical motion (isopycnal and/or mooring displacement) versus trapped water masses. In the upper part of the water column, the presence of Indian water trapped inside the Agulhas rings or advected within dipoles have been identified by both data sets. The intrusion of subantarctic water in the upper water layers due to the dipole dynamics is also highlighted. Associated

with these upper intrusions of Indian and subantarctic waters due to the presence of dipoles, high-salinity I-AAIW at

intermediate depth and NADW at the deepest level are also illustrated. The presence of intermediate high-salinity I-AAIW is also evidenced during the period of the cyclonic eddy crossing.

It has been shown that the trapping depth of rings can reach the sea floor (Van Aken et al., 2003). The analyses of our tall mooring deep SBE MicroCat's data show that not only Agulhas rings but also water intrusions due to presence of dipoles, extend to 4400 m of depth impacting the NADW layers and even deeper layers.

This study presents the first combination of full-depth-moorings and reconstructed fields from CPIES data combined with a GEM technique in the upper water column within the eastern South Atlantic region. Here properties are well resolved by the combination of local sensors and by the GEM reconstructed fields. The reconstructed fields capture the same changes of the temperature and salinity variability in the upper water column as do the local sensors data when mooring motion is smaller than the isopycnal displacement. Relatively small differences can be attributed to the limited vertical sampling resolution in the upper 500 m, and the mixing associated with the mesoscale activity in the Cape Basin. When vertical mooring motion is larger than the isopycnal displacement, the local sensors sample a large range of neutral density and can overestimate the anomalies during the event compared to the GEM reconstructed fields. However, the deep-water column properties remain to be analyzed with local temperature and salinity sensors. Moreover, the distance between CPIES and full-depth-moorings (*e.g.* 210 km between M3 and M4) does not allow precise transport estimates - as the typical velocity decorrelation length scale is smaller than this distance (~100 km – *e.g.*, Donohue et al., 2010; Meinen et al., 2017).

Finally, this work presents the first independent observations comparison between full-depth-moorings and CPIES data sets along the SAMBA-east array that gives some evidence of eastern boundary buoyancy anomalies associated with migrating eddies. It also highlights the need to continuously sample the full-water depth as inter-basin exchanges occurs intermittently and affect the whole water column. Future investigations with longer time series at these existing sites will lead to a better understanding of the eastern boundary current variability and Indo-Atlantic exchanges. The impact of each isolated mesoscale eddy will not be adequately resolved at this scale but the global eddy thickness flux anomalies can be improved. The CPIES records used in combination with the moored instruments in the western part of the SAMBA array will improve of our understanding of the strength and variability of the AMOC.

**Acknowledgment.** The authors would like to express their great appreciation to the Captain, officers and crew of the research vessels, which have supported this program to date, including the South African research vessels the RV *Algoa*, and the RV *SA Agulhas II*. We are warmly grateful to the technical staff who worked on the preparation, deployment and the recovery of the instruments. And our thanks to those who have helped coordinate these challenging international cruise collaborations. The authors acknowledge the support of grants from NRF/SANAP – SAMOC-SA programme. MK

acknowledges support from a NRF grant via a South-African post-doctoral fellowship. MK work on this study was carried out in part under the auspices of the Cooperative Institute for Marine and Atmospheric Studies (CIMAS), a Cooperative Institute of the University of Miami and the National Oceanic and Atmospheric Administration (NOAA), cooperative agreement NA10OAR4320143. MK also acknowledges support from the NOAA Atlantic Oceanographic and Meteorological Laboratory. This work was also supported by the European Union's Horizon 2020 research and innovation programme under

grant agreement no. 633211 (AtlantOS) and the 11- ANR-56-004 grant for SS. Users can access the CPIES data available at LOPS from T. Terre (thierry.terre@ifremer.fr). Data from the hydrographic casts and the full-depth-moorings can be available at DEA from T. Lamont (tarron.lamont@gmail.com) under certain data sharing policies. The altimeter products were produced by Ssalto/Duacs and distributed by Aviso, with support from CNES (http://www.aviso.oceanobs.com/duacs/). The Mesoscale Eddy Trajectory Atlas products were produced by SSALTO/DUACS and distributed by AVISO+

(http://www.aviso.altimetry.fr/) with support from CNES, in collaboration with Oregon State University with support from NASA. The authors thank Renellys C. Perez, Chris S. Meinen and Jonathan Lilly for precious help, comments and useful discussions. Finally, we thank the editor and the three reviewers of this manuscript for their constructive comments.

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

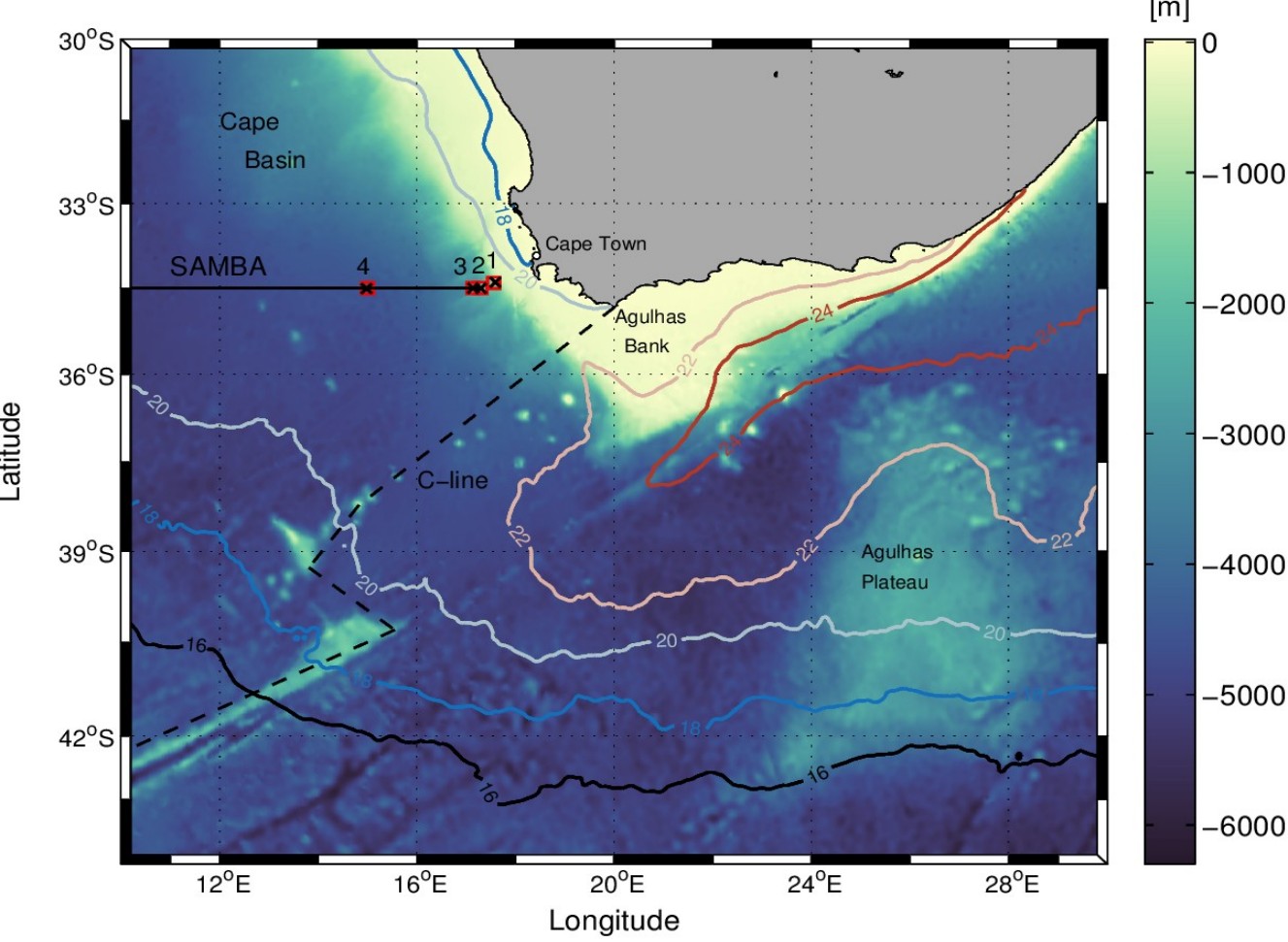

**Figure 1: Study area with shaded color representing the bathymetry [m] from ETOPO1. The thin black line denotes the position of the SAMBA-east transect and the black crosses (red squares) represent the mooring (CPIES) positions with their associated numbers (1, 2, 3 and 4). The dotted black line denotes the C-line position. 2015 annual averaged SST isotherms from the ODYSSEA dataset are plotted with colored contours.**

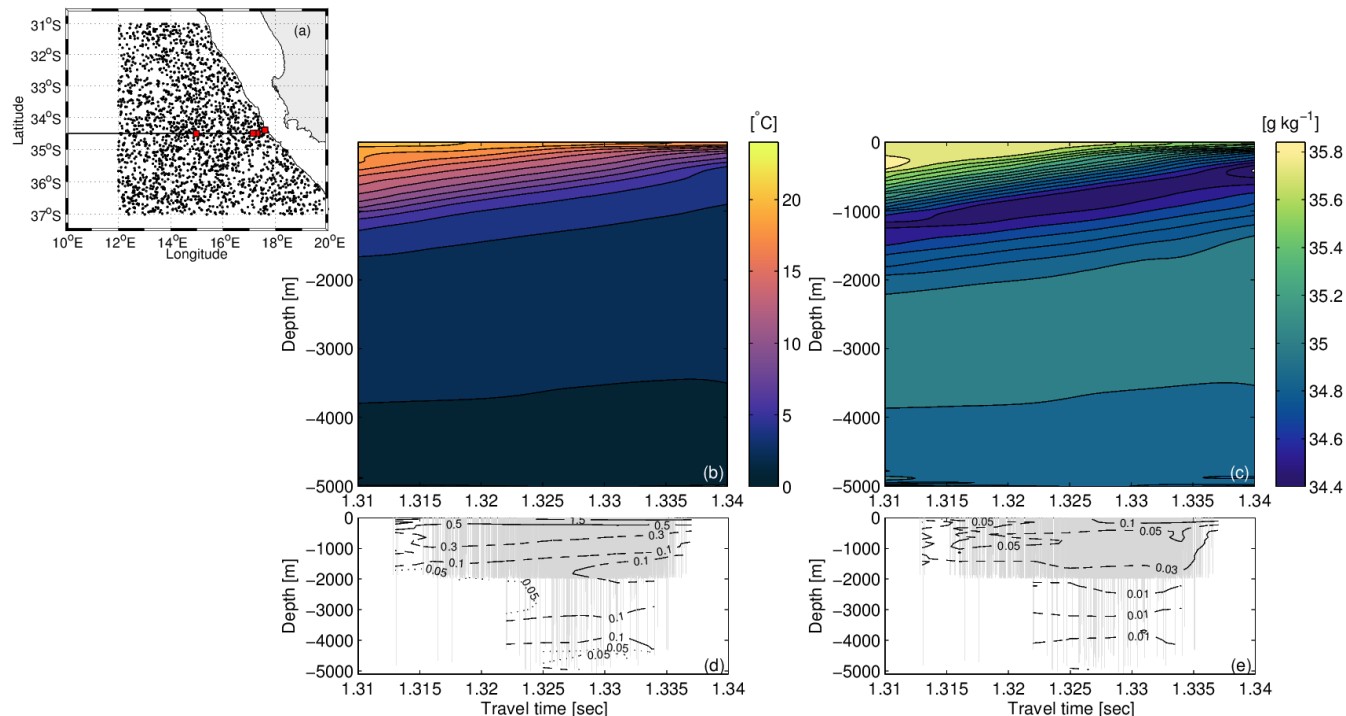

**Figure 2: (a) Positions of the CTD and Argo casts from 1983 to 2018 collected around the SAMBA-east transect and used to create the GEM field. The CPIES are identified by the red squares along the transect. Gravest empirical mode (GEM) fields of conservative temperature (b) and absolute salinity (c) determined for the SAMBA-east line. The root-mean-squared (rms) differences between the original hydrographic measurements and the smoothed look-up table values are shown in the lower plots (d, e). The solid, dashed, and dotted contours represent progressively smaller contour intervals. The gray vertical lines show the**
**locations of the hydrographic measurements.**

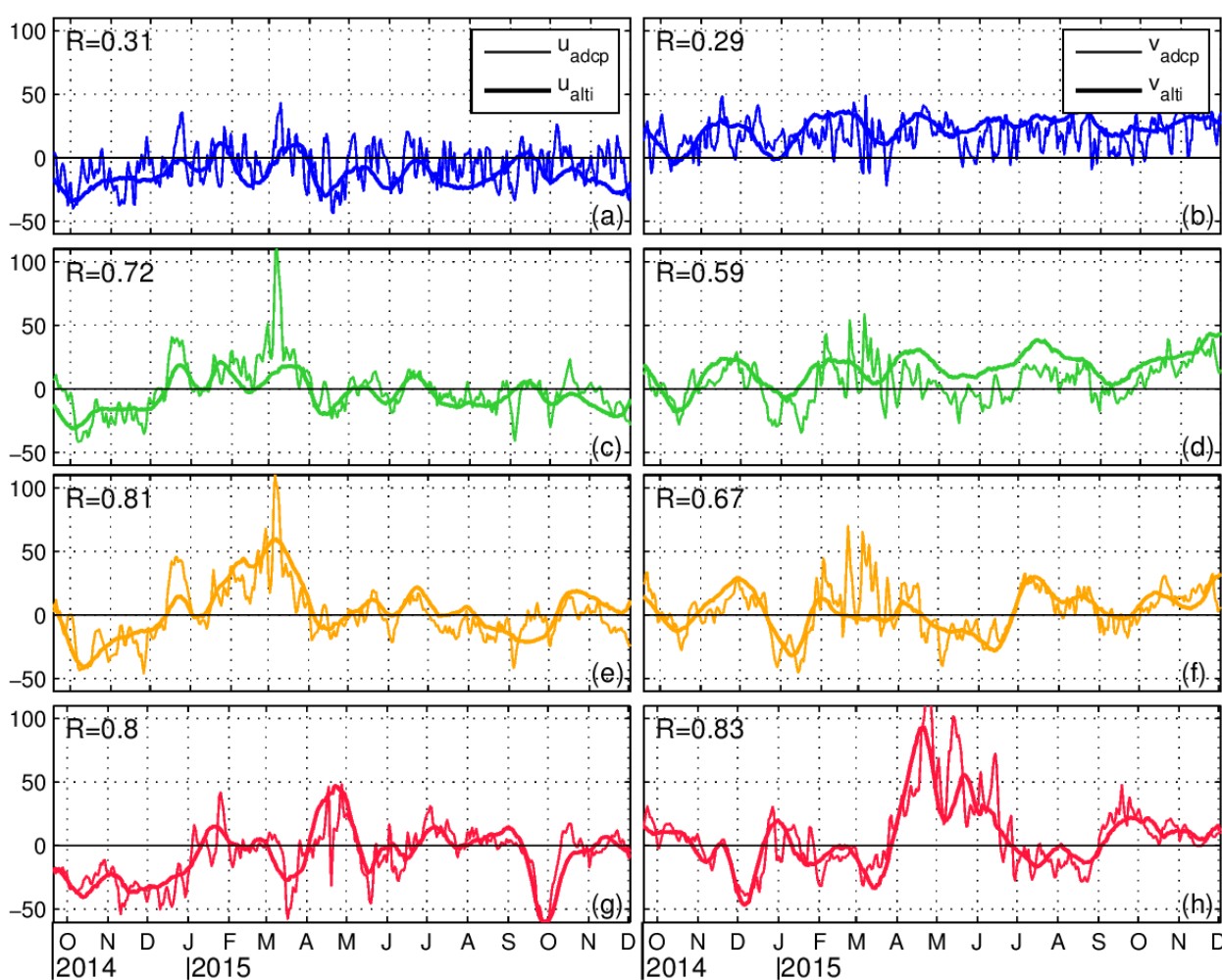

**Figure 3:** Temporal evolution of the near-surface velocities recorded at the last bin by the upward-looking ADCP (thin lines) and the surface geostrophic velocity derived from satellite altimetry (bold lines). The u (left-column) and v (right column) component of the current velocities [cm s$^{-1}$] are represented at M1 (a,b - blue line), M2 (c,d - green line), M3 (e,f - yellow line) and M4 (g,h - red line). The correlation coefficient value (R) is indicated in the top left corner of each panel.

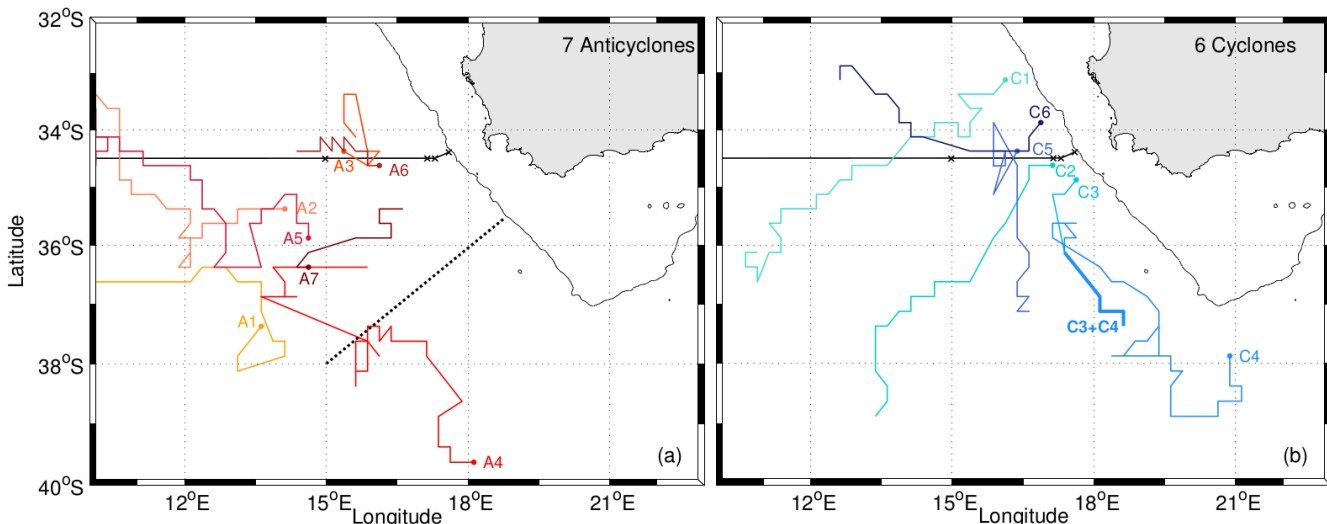

**Figure 4: Eddy trajectories from September 15, 2014 to December 15, 2015 for eddies influencing the moorings measurements. The eddies are named as AN (anticyclonic eddies - a) and CN (cyclonic eddies - b), with N a number assigned in chronological order by the eddy tracking scheme. The circles indicate the starting positions of an eddy track. Bold blue line represents the trajectory of an eddy after a merging. (a) The dashed black line represents the northern route of Agulhas rings defined by Dencause et al. (2010).**

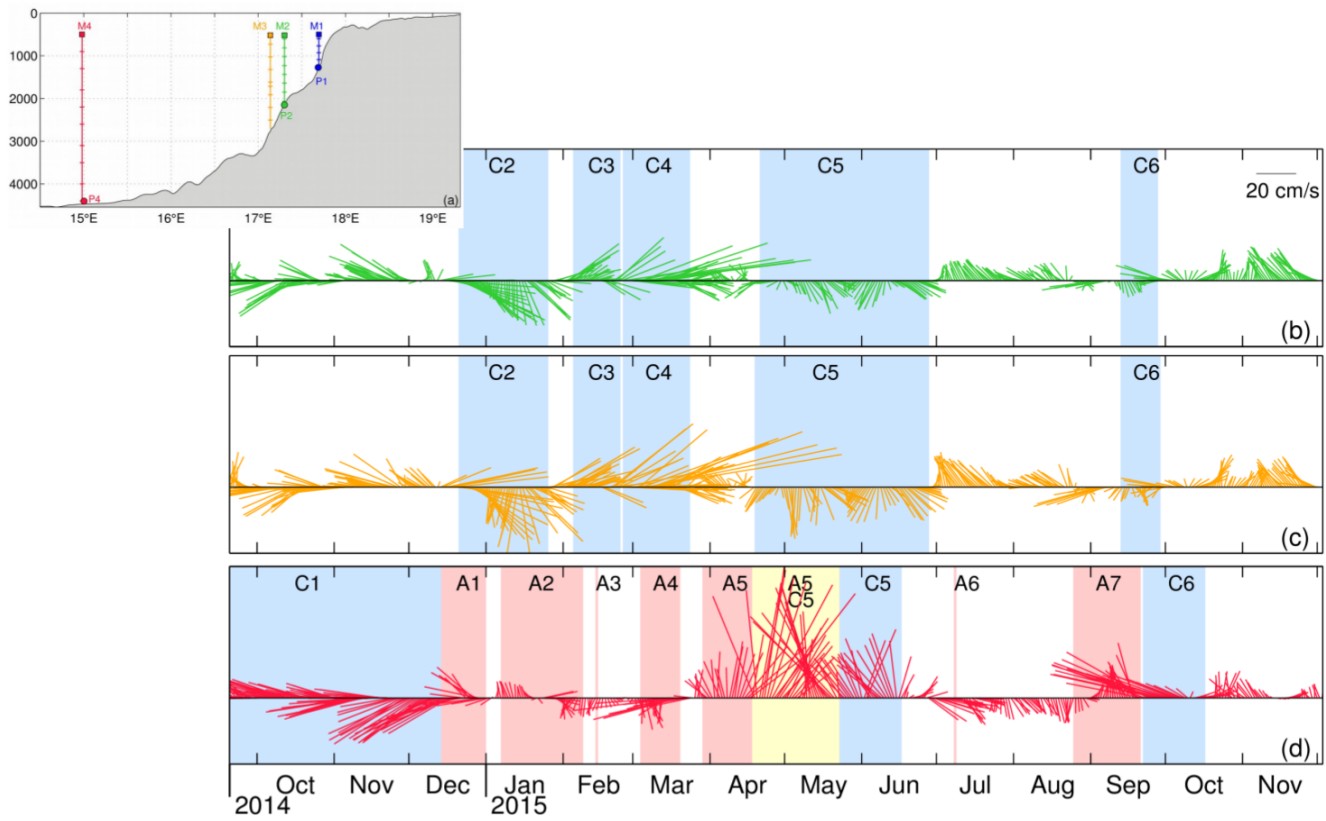

**Figure 5: (a) Vertical scheme of the mooring lines positions along the latitude 34.5°S. The squares represent the upward-looking ADCP positions, the small horizontal lines indicate the SBE Microcat's sensors and the circles at the bottom show the CPIES. Temporal evolution of the current vector stick-plot at M2 (b), M3 (c) and M4 (d). One vector per day is shown. The shaded areas show the eddy events defined with altimetry data (red: anticyclonic eddies, yellow: dipole event and blue: cyclonic eddies).**

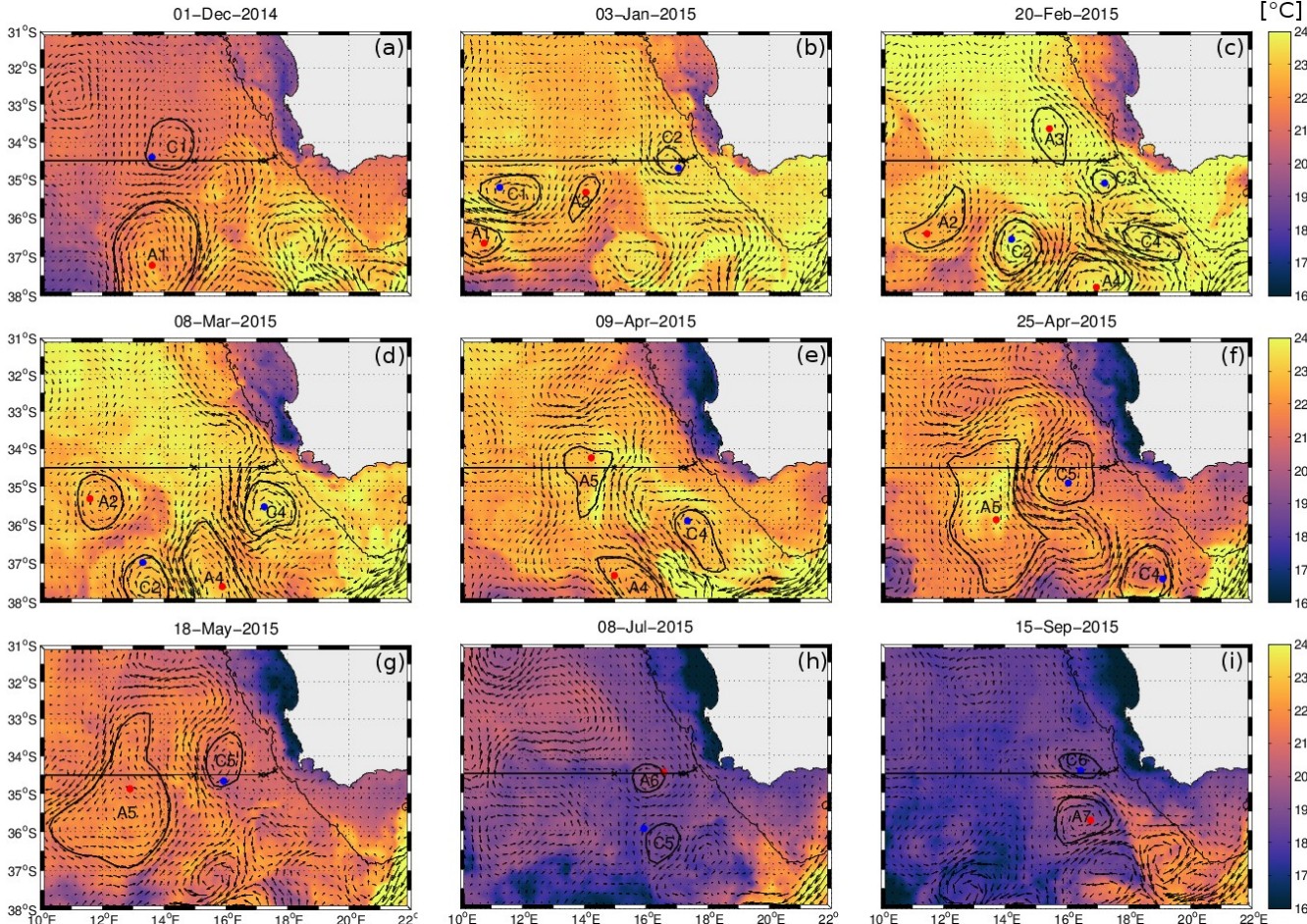

**Figure 6: SST satellite images for selected dates between December 1, 2014 to September 15, 2015. The black line denotes the position of the SAMBA line and the crosses represent the mooring positions. Black contours show the coherent core of the eddies identified by the new method of detection. The colored dots (red for AN and blue for CN) are the eddy centers detected in the META product.**

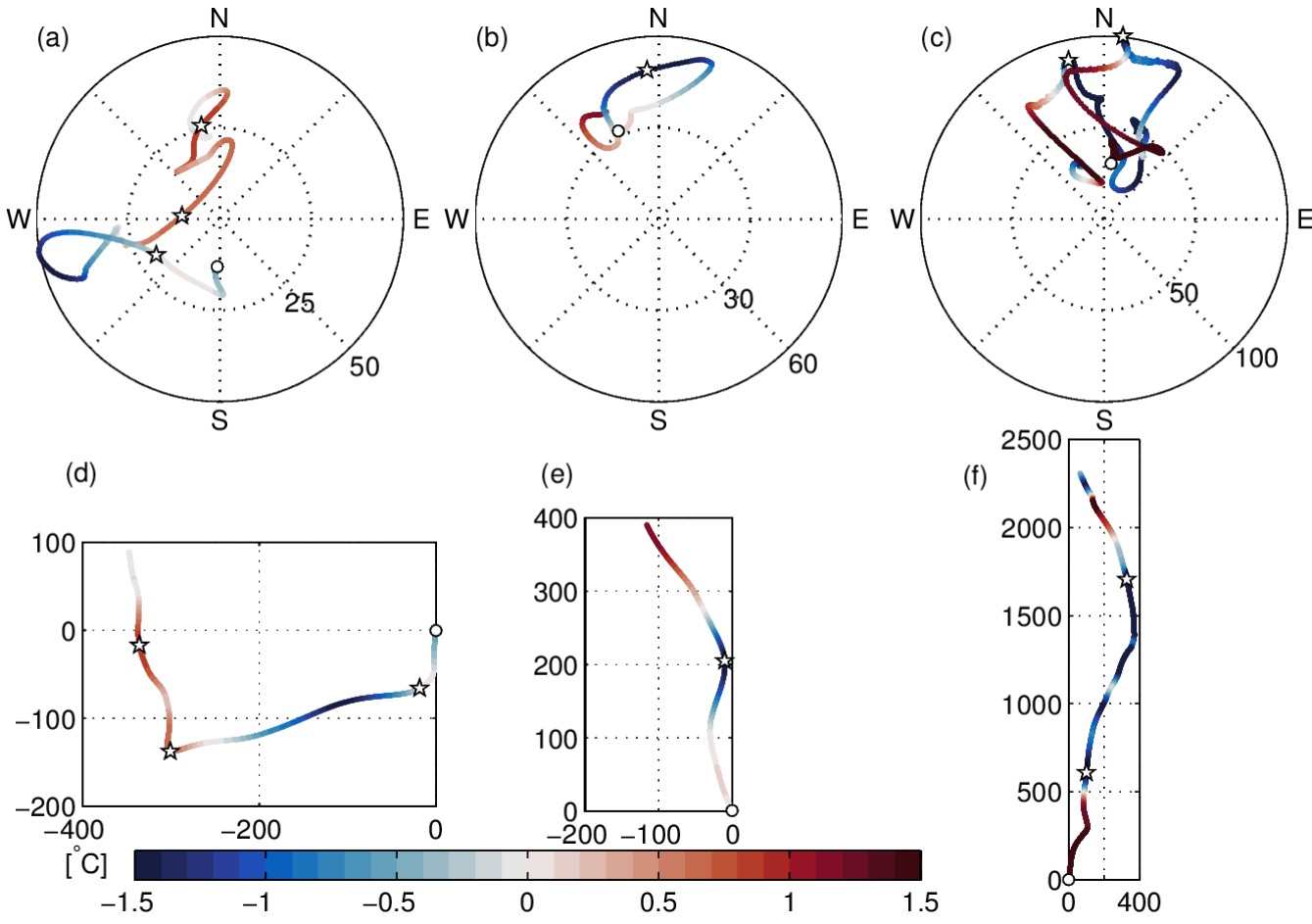

Figure 7: (a, b, c) Hodograph of the mean currents of the upper-water column observed at M4 for three case studies (a-Anticyclonic eddy; b- Cyclonic eddy; c- Dipole). Color represents the temperature recorded at the first SBE37 Microcat's de-meaned over the time period of each case studies (a: March 9 – April 6, 2015; b: May 28 – June 10, 2015; c: April 8 – May 29, 2015). The estimated centers of 6 events have been marked with white stars. The beginning of each time series has been marked with white dots. (d, e, f) Progressive vector diagram for the mean currents of the upper-water column observed at M4 for three case studies. Color, time period and symbols are the same as the hodograph.

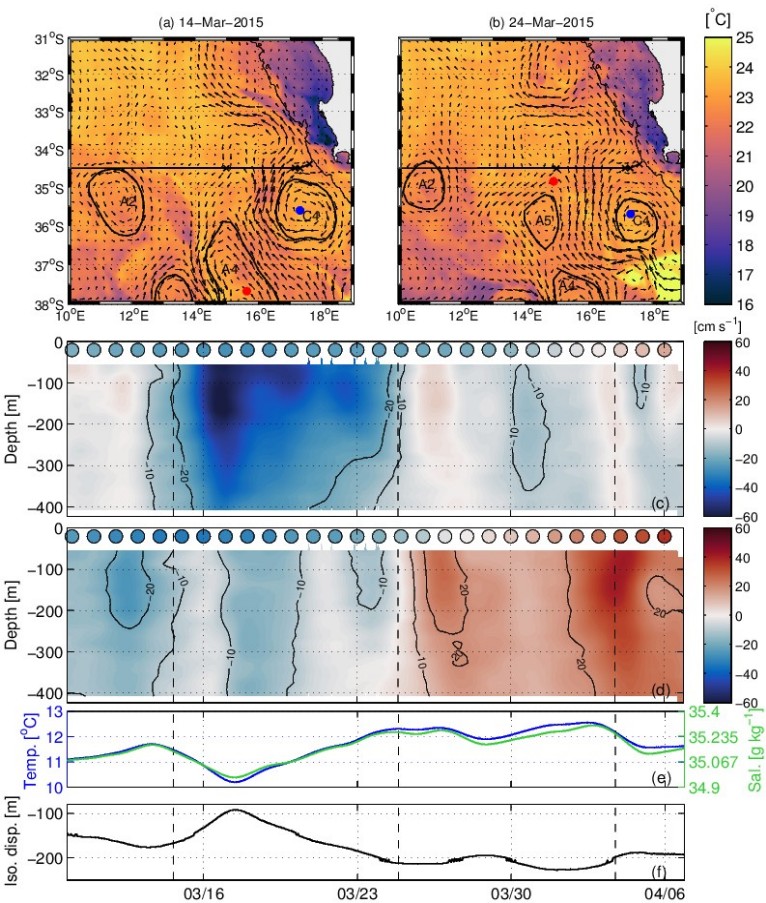

Figure 8: Anticyclonic eddy: (a, b) SST satellite images for selected dates (vertical black lines on the time series). Vertical-temporal section of the u-component (c) and v-component (d) of the current speed measured from the upward-looking ADCP at M4. The colored circles at the top of the section show the u-component and v-component of the current speed from altimetry data. (e) Temporal evolution of temperature (blue line) and salinity (green line) and (f) the isopycnal displacement recorded at the first SBE37 MicroCat's between 430 and 450 m depth.

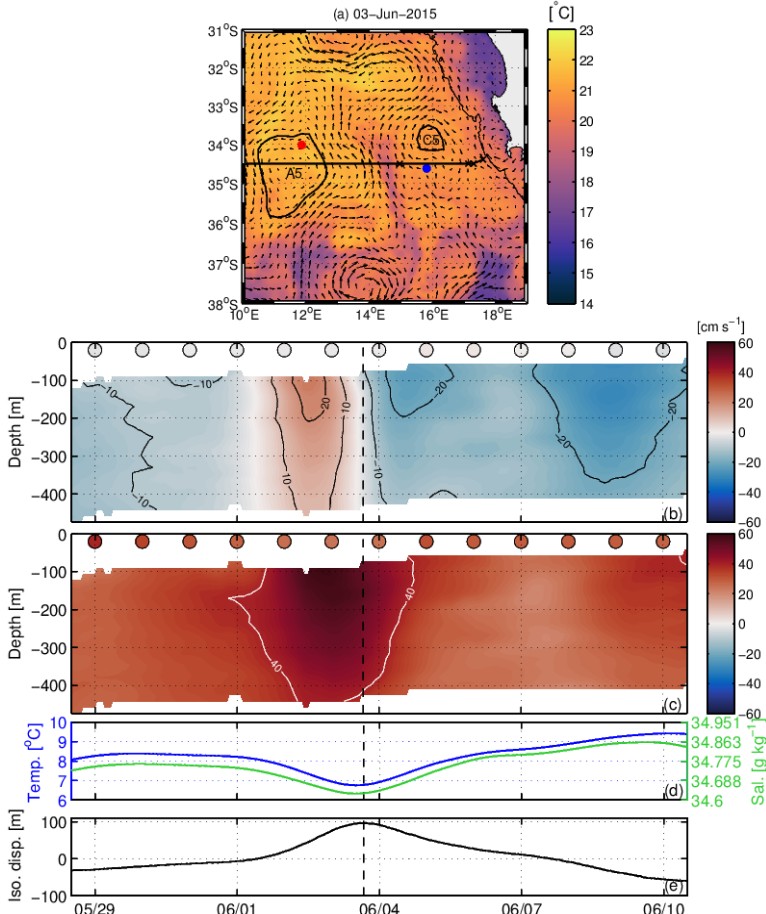

**Figure 9: Cyclonic eddy: (a) SST satellite images for selected date (vertical black lines on the time series). Vertical-temporal section of the u-component (b) and v-component (c) of the current speed measured from the upward-looking ADCP at M4. The colored circles at the top of the section show the u-component and v-component of the current speed from altimetry data. (d) Temporal evolution of temperature (blue line) and salinity (green line) and (e) the isopycnal displacement recorded at the first SBE37 MicroCat's between 470 and 500 m depth.**

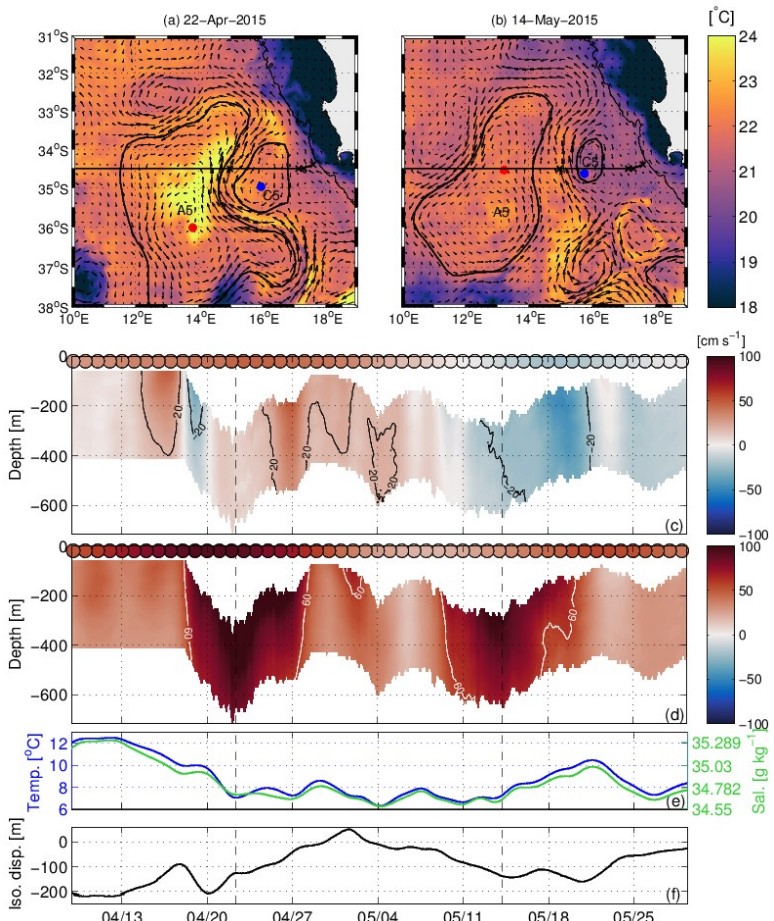

**Figure 10: Intrusion water: (a, b) SST satellite images for selected dates (vertical black lines on the time series). Vertical-temporal section of the u-component (c) and v-component (d) of the current speed measured from the upward-looking ADCP at M4. The colored circles at the top of the section show the u-component and v-component of the current speed from altimetry data. (e) Temporal evolution of temperature (blue line) and salinity (green line) and (f) the isopycnal displacement recorded at the first SBE37 MicroCat's between 435 and 700 m depth.**

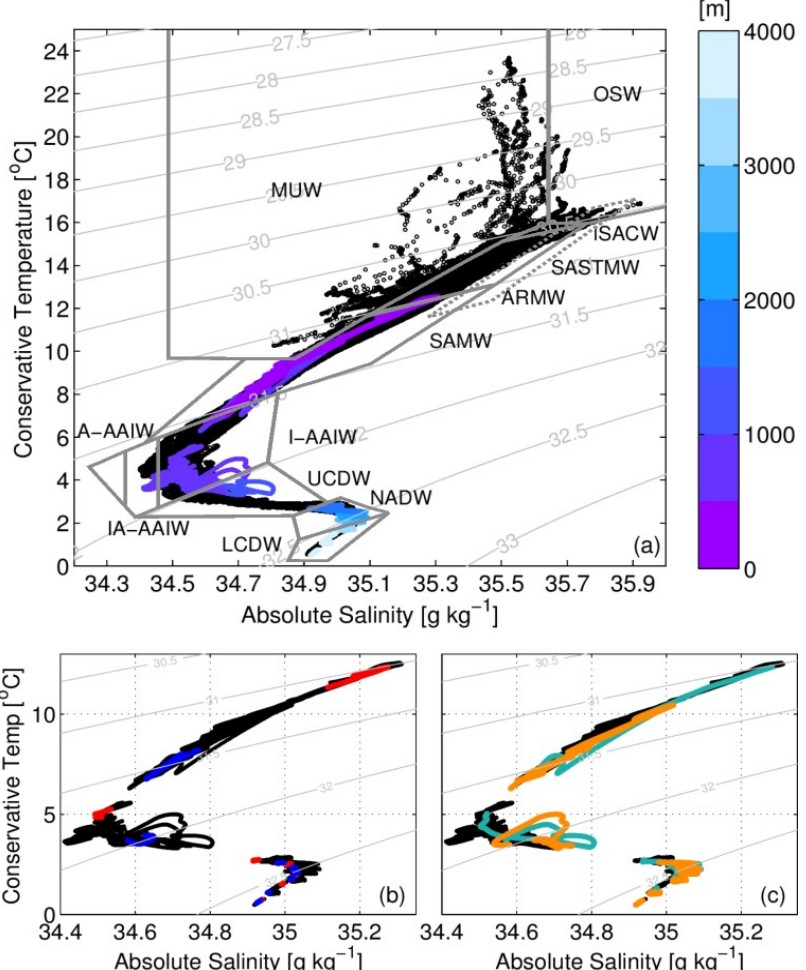

**Figure 11: (a)** Conservative Temperature (°C) – Absolute Salinity (g kg$^{-1}$) relationship from the hydrographic cruises sampling along the SAMBA-east array on board the RV *SA Agulhas II* (black dots) and from the SBE37 MicroCATs measurements at M4 (depth-colored circles). The gray contours represent the potential density anomaly with a reference pressure of 1000 dbar. Water masses are indicated by gray boxes (OSW – Oceanic Surface Water, MUW – Modified Upwelled Water, lSACW – Light South Atlantic Central Water, SASTMW – South Atlantic Subtropical Mode Water, SAMW – Subantarctic Mode Water, ARMW – Agulhas Ring Mode Water, I-AAIW – Indian Antarctic Intermediate Water, IA-AAIW – Indo-Atlantic Antarctic Intermediate Water, A-AAIW – Atlantic Antarctic Intermediate Water, UCDW – Upper Circumpolar Deep Water, NADW – North Atlantic Deep Water, and LCDW – Lower Circumpolar Deep Water). **(b,c)** Close-up views of the deep water recorded with the MicroCat's illustrating the distributions observed in the anticyclonic/cyclonic cases **(b)** and intrusion cases **(c)**. The colors represent the measurements during each case study(anticyclonic eddy: red; cyclonic eddy: blue; Intrusions: cold water - green, warm water – orange).

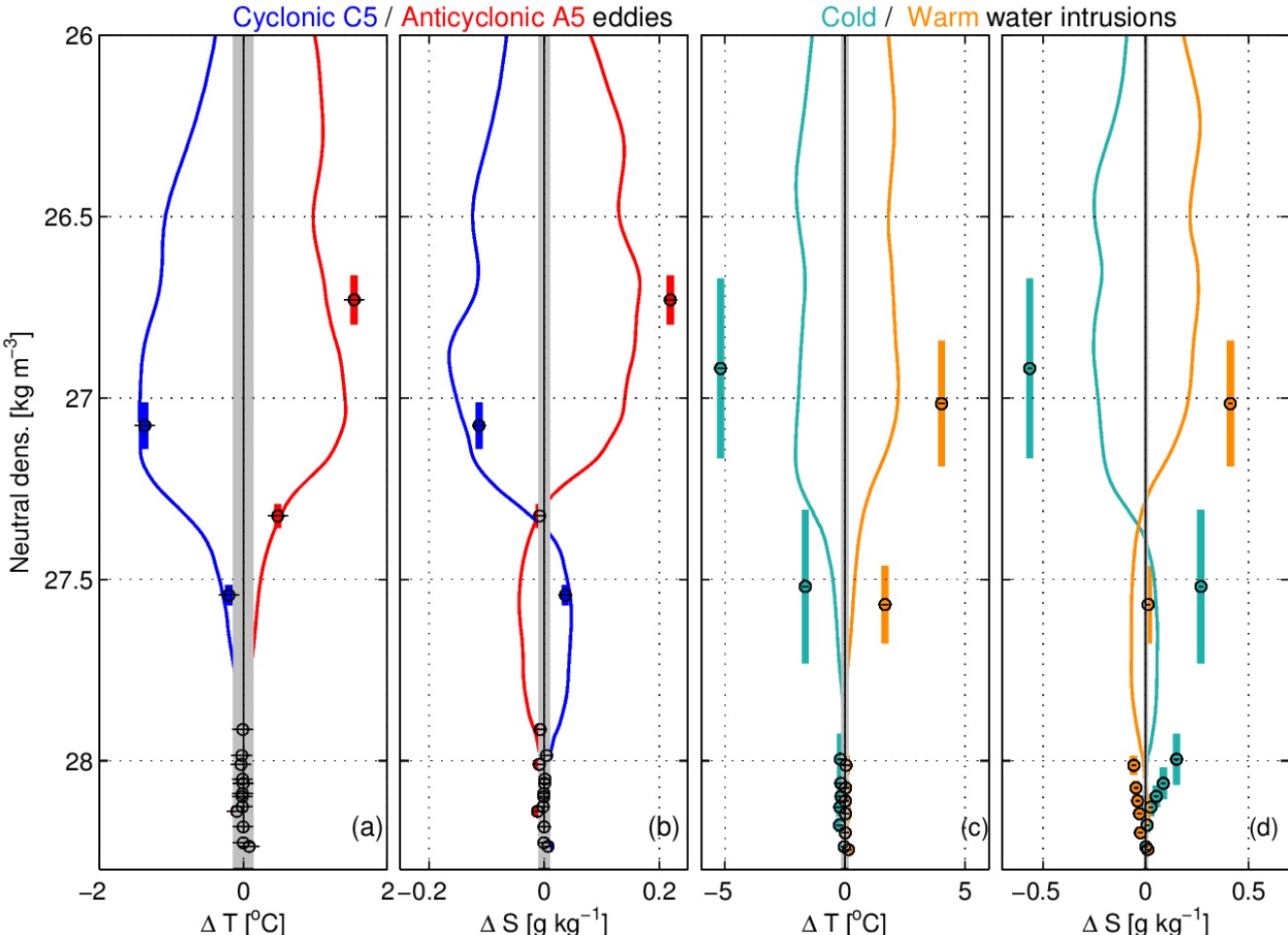

**Figure 12: Vertical profiles of the conservative temperature (a,c) and absolute salinity (b,d) anomalies at M4 due to the presence of the anticyclonic and cyclonic eddies (a,b) and the two intrusions associated with the dipole (c,d). The colored dots represent the anomalies from the SBE37 Microcat's with their estimated accuracy (horizontal black line) and their range of neutral density variability (colored boxes). The solid colored lines represent the anomalies from the reconstructed GEM fields. The gray shaded areas around the horizontal zero axis show the range of value below the estimated measurement's accuracy.**

**Table I: Moorings and CPIES characteristics.**

| | Mooring 1 - M1 | Mooring 2 - M2 | Mooring 3 - M3 | Mooring 4 - M4 |
|---|---|---|---|---|
| Latitude | 34ºS 23.636' | 34ºS 29.960' | 34ºS 30.010' | 34ºS 30.360' |
| Longitude | 17ºE 35.664' | 17ºE 18.064' | 17ºE 8.3640 | 14ºE 58.810' |
| Water depth (m) | 1121 | 2094 | 2970 | 4474 |
| Start | 17/09/2014 08:00 | 17/09/2014 11:00 | 17/09/2014 16:00 | 19/09/2014 23:00 |
| End | 01/12/2015 05:00 | 01/12/2015 13:00 | 02/12/2015 10:00 | 03/12/2015 05:00 |
| ADCP | 75kHZ RDI ADCP | 75kHZ RDI ADCP | 75kHZ RDI ADCP | 75kHZ RDI ADCP |
| Depth (m) | 600 | 605 | 435 | 450 |
| Cell size (m) | 16 | 16 | 16 | 16 |
| Number bins | 34 | 34 | 24 | 24 |
| Depth 1st bin (m) | 582.85 | 588.65 | 410.95 | 413.78 |
| Number of records | 10 558 | 10 566 | 10 587 | 10 606 |
| Depth (m) SBE 37 SMP MicroCat's C-T ($P$) Recorder | **600**, 770, **930**, 1090 | **605**, 810, 1020, 1230, 1435, 1642, 1850, 2070 | **435**, 730, 1030,1325, 1620, 1720, 1915, 2210, 2505, 2880 | **450**, 895, 1340, 1780, 2250, 2665, 3110, 3550, 3985, 4360 |

| | CPIES 1 - P1 | CPIES 2 - P2 | CPIES 3 - P3 | CPIES 4 - P4 |
|---|---|---|---|---|
| Latitude | 34ºS 24.348' | 34ºS 29.813' | 34ºS 29.964' | 34ºS 30.252' |
| Longitude | 17ºE 33.456' | 17ºE 18.036' | 17ºE 8.3100' | 15ºE 0.161' |
| Water depth (m) | 1266 | 2129 | 2850 | 4482 |
| Start | 07/09/2013 00:00 | 07/09/2013 00:00 | No data | 07/09/2013 00:00 |
| End | 11/08/2015 00:00 | 11/08/2015 00:00 | No data | 11/08/2015 00:00 |

**Table II: Summary of comparison statistics for satellite altimetry and moored current meter, with R the correlation coefficient, $\Delta V_{rms}$ the root mean square differences between the zonal and meridional components of ADCPs and those from altimetry (Eq. 1), and the bias between those components (Eq. 2).**

| | R | | $\Delta V_{rms}$ [cm s$^{-1}$] | | Bias [cm s$^{-1}$] | |
|---|---|---|---|---|---|---|
| | u | v | u | v | u | v |
| M1 (54.9 m) | 0.31 | 0.29 | 15.4 | 13.3 | 5.3 | -1.5 |
| M2 (60.7 m) | 0.72 | 0.59 | 14.0 | 12.9 | 3.0 | -8.8 |
| M3 (43.0 m) | 0.81 | 0.67 | 13.3 | 13.9 | -5.8 | 1.2 |
| M4 (45.8 m) | 0.80 | 0.83 | 13.0 | 15.9 | -4.3 | -4.9 |

**Table III: Characteristics of the eddies passing over the moorings from satellite altimetry. The time period, the radius and the azimuthal velocity of the different eddies are derived from our eddy detection method. The eddies are named as CN (cyclonic eddies) and AN (anticyclonic eddies), with N a number assigned in chronological order by the eddy tracking scheme. Eddies generated at the Agulhas retroflection are identified with a star.**

| $\Delta t$ [days] <br> $R_V$ [km] <br> $V_{max}$ [cm s$^{-1}$] | C1 | C2 | C3 | C4* | C5 | C6 | |
|---|---|---|---|---|---|---|---|
| M2, M3 |  | 28-37 <br> 35-36 <br> -36.0 | 20 <br> 36 <br> -38 | 27-28 <br> 76 <br> -66 | 55-71 <br> 61-65 <br> -41 | 11-17 <br> 48-56 <br> -26 | |
| M4 | 93 <br> 62 <br> -34 |  |  |  | 61 <br> 65 <br> -43 | 26 <br> 68 <br> -23 | |
| $\Delta t$ [days] <br> $R_V$ [km] <br> $V_{max}$ [cm s-1] | A1 | A2 | A3 | A4* | A5 | A6 | A7 |
| M4 | 19 <br> 104 <br> 54 | 34 <br> 80 <br> 33 | 2 <br> 92 <br> 21 | 17 <br> 110 <br> 59 | 56 <br> 154 <br> 37 | 2 <br> 44 <br> 32 | 28 <br> 77 <br> 51 |

**Table IV: Estimate eddy parameters. t0: the eddy's center apparent at M4, Δt: temporal half eddy duration, U/$\Theta_{a/f}$ ($X_{a/f}$): the estimated mean flow magnitude and direction (eddy apparent radius) using the angle method (subscript "a") or the filtering method (subscript "f"). $V_n$/$V_U$: the estimated maximum eddy currents using the normal (subscript "n") or the U subtraction method (subscript "U"). Rossby number using the different value for V and X.**

|  | A4 | A5 | C5 |
|---|---|---|---|
| t0 | March 14, 2015 - 3pm | March 24, 2015 - 9pm | June 3, 2015 - 4pm |
| Δt [h] | 54 | 48 | 25 |
| $U_a$ [cm s$^{-1}$] | 27.0 | 18.3 | 50.7 |
| $\Theta_a$ | 110 | 77 | 2 |
| $U_f$ [cm s$^{-1}$] | 28.9 | 16.5 | 49.5 |
| $\Theta_f$ | 150 | 77 | 2 |
| $V_n$ [cm s$^{-1}$] | 25.8 | 20.6 | -19.2 |
| $V_U$ [cm s$^{-1}$] | 26.0 | 21.7 | -20.0 |
| $X_a$ [km] | 52.4 | 31.6 | 45.7 |
| $X_f$ [km] | 56.2 | 28.4 | 44.5 |
| Rossby number | 0.11-0.12 | 0.15-0.17 | 0.10 |

**Table V: Table with the different water masses and their definition in terms of conservative temperature, absolute salinity and density layers.**

| Water mass | | Definition |
|---|---|---|
| MUW | | $9.71 \leq T \leq 21.3$°C; $34.661 \leq S_A \leq 35.647$ g kg$^{-1}$ |
| OSW | | $\gamma^n < 26.2$ kg m$^{-3}$; $T \geq 16.0$°C; $S_A \geq 35.647$ g kg$^{-1}$ |
| lSACW | | $15.15 \leq T \leq 16$°C; $35.506 \leq S_A \leq 35.764$ g kg$^{-1}$ |
| SASTMW | | $11.69 \leq T \leq 16$°C; $35.49 \leq S_A \leq 35.764$ g kg$^{-1}$ |
| SAMW | | $6.32 \leq T \leq 13.18$°C; $34.78 \leq S_A \leq 35.49$ g kg$^{-1}$ |
| ARMW | | $11.60 \leq T \leq 17.00$°C; $35.18 \leq S_A \leq 35.81$ g kg$^{-1}$ |
| AAIW | I-AAIW | $S_A \geq 34.47$ g kg-1 |
| | IA-AAIW | $34.37 \leq S_A \leq 34.47$ g kg-1 |
| | A-AAIW | $S_A \leq 34.37$ g kg-1 |
| UCDW | | $27.55 < \gamma^n < 27.92$ kg m$^{-3}$; $T < 4.23$°C; $34.696 \leq S_A \leq 35.916$ g kg$^{-1}$ |
| NADW | | $27.92 < \gamma^n < 28.11$ kg m$^{-3}$; |
| LCDW | | $28.11 < \gamma^n < 28.26$ kg m$^{-3}$; |