# Peer review of "Moored observations of mesoscale features in the Cape Basin: Characteristics and local impacts on water mass distributions"

_Ocean Science, 2017_

## Referee Comment (RC1) · Anonymous Referee #1 · 28 Nov 2017

The manuscript by Kersale et al. is based on the analysis of a unique data set from a mooring array in the eastern South Atlantic in combination with PIES and satellite data. The authors identify four categories of mesoscale eddy events in the data ("cases") and provide an in-depth analysis of these cases. They make use of time series records from individual instruments (MicroCat, ADCP) and also used GEMS reconstructed hydrography information from depth integrating PIES observations.

I have a number of comments below that I think are important to address before the manuscript can be published.

First of all, I think the introduction needs re-writing. It is irritating that so much is

written about MOC and the different MOC programs in the Atlantic and so little about the problem the study is concerned with. Why do we need to know that there is a MOVE or RAPID when analysing and interpreting TS anomalies in eddies in the Cape Basin? – How does that really is connected to the study here? If it would be I would expect that at the end of the paper I know how this study contributes to the problem. Maybe I overlooked it – but I can't find a contribution of the findings presented here to the MOC. I understand that the complete SAMBA/SAMOC array (full basin width) aims to measure the MOC variability at the respective latitude but really this study is looking into a "local" aspect of hydrography and flow variability in the Cape Basin. It just makes use of the mooring infrastructure/data. I strongly suggest to shorten the AMOC part (to maybe one sentence only) but to expand a lot more on what the study is concerned with (and what is also reflected in the title): A LOCAL study on the structure of eddies (on the mesoscale only) in the Cape Basin. To me this would also mean to expand in the introduction on eddy/eddy interaction (dipole discussion), eddy detecting in mooring data (quite a bit of studies have worked on that), calculation of anomalies (please pay attention to pressure versus density space (see comments below). I would also suggest a brief introduction to GEMS in this set up – I understand that you used the T/S/pressure at the grid points (moored instrumentation) to set the local sound velocity points – the unknown would then most of the water column upward from the ADCP (upper 500m or so) – and which is were likely a lot of warm water (and thus sound speed) variability lies? Do you use satellite SST to give you one sound speed gridpoint at the surface?

Specific comments:

Line 40 to 50: why do we need to know these details? see my general comments above

Line 55-60: if you decide to leave in this part it would be useful to introduce HOW the mechanisms are behind MOC and water mass transformation and how precisely does that link to the analysis of eddies in the Cape Basin?

[Figure]

Line 64: are these "anomalies" similar to the anomalies that will be studies here? or is the first a transport anomaly in the DWBC (western array) and here local anomalies due to mesoscale eddies in the eastern boundary current are studied?

Line 88: I am missing all references to research on the circulation elements along the eastern boundary and in the Cape Basin. Are there undercurrents? Is there a coastal current? Does it connect to the north poleward or south? I would expect that the South Africans and Namibians have worked quite a bit here and know about the local/boundary circulation?

Line 89: Given the discussion later about eddies, dipoles, filaments etc. could you please introduce a bit more general wisdom on eddies and eddy/eddy interaction and inclduing filamentation and how all that links to water mass transformation (isopycnal heave versus water mass anomalies)?

Line 103: the MicroCat's have no pressure? How did you add pressure information? (it is required for calculating salinity). What will be the uncertainty in Salinity? Please indicate that you did not use the oxygen (or did you?)

Line 108: why do you know that the PIES was destroyed if it was lost?

Line 109: different approaches have been used in the past (e.g. creep function) - why linear? and, does it make a difference using other techniques? or maybe it is not relevant for your specific study (if so – make clear).

Line 120: What is the conclusion on accuracy after the calibration? Is it homogeneous across the data sets? How have the Microcats been quality controlled? And what do you expect for their accuracy?

Line 132: Please provide information about ADCP configuration and processing - depth allocation, compass calibration, depth cell length and ensemble lengths, number of bins, burst or spread mode?

Line 134: for my understanding most of the content in this paragraph (3.1.) must be

part of the introduction.

Line 139: is there maybe seasonal or other variability? What drives it? wind, topographic waves? Maybe a connection to the equatorial belt via waves?

Line 155-156: I would expect that to be part of the following chapter (3.2)

Line 156: why is there such a large std on the propagation? Could you add a few words explaining that. Many stationary eddies? interaction with mean flow?

Line 157: please add time period: "in 14?? months we observed...."

Line 169-170: how do you know that this is "usually below the Ekman layer"? what is your critieria for "Ekman layer"? did you compare ADCP shear with geostrophic shear? Did you calculate the Rossby number for the eddies? how important are the non-linear terms (Rossby number)? and how would that impact a conclusion on Ekman layer and other dynamics?

Line 170: is only one (the last?) bin used or a mean over a number of bins (which ones)? has the data been corrected for tidal effects via a tidal model or only through the butterworth filter? what is the bin length? (ADCP configuration should be added to Data and Methods section – see comment above)

Line 189: could you pls provide a depth range here?

Line 195: what do you mean by "dipole"? is that just neighbouring eddies or do the centres need to be at a minimal/critical distance? are there only cyclone/anticyclone dipoles or can there by a anticyclone/anticyclone dipole? more to add to Introduction. . .

Line 197: Referenced to a discussion below in this section: It may be helpful to show the nature of the variability discussed here in the context of the 4 "cases" - also in a T/S diagram of the time series - maybe embedded in the background TS. This would reveal immediately if this is a thermohaline anomaly of simply heave.

Line 198: in which respect "consistent" and "complete"? maybe saying "consistent and

complete picture of the mesoscale dynamics in the Cape Basin"? Even if the basin would be filled with eddies it is not clear that the basin dynamics as a whole is similar to the dynamics of the individual components.

Line 209: you differentiate between propagating (transient?) and "quasi stationary" eddies? what are the thresholds applied for that? or does transient means that the eddy dissolves when crossing the array?

Line 221: what does "associated" mean for you?

Line 227: in this summary you leave out all information about the temperature/filaments you discuss before - why? isn't that the key for the MOC connection?

Line 232: it would be helpful to get an idea about how well the centre of the eddy crossed the mooring. Progressive vector diagram type techniques can help to analyse that (see Lilly, J., and P. Rhines (2002), Coherent eddies in the Labrador Sea observed from a mooring, J. Phys. Oceanogr., 32, 585–598). Very much of your conclusion depends on who well the eddy centre was observed.

Line 251: what is the difference between a "core" speed and the maximum velocity magnitude? at which depth is the maximum velo. found?

Line 253: How much of this decrease is caused by TS anomalies (relative to ambient water) versus the vertical displacement of isopycnals in the eddy caused by rotation (heave)?

Line 257: is this a one-record-only speed? what depth? (this applies to all the numbers presented here (see cases above). with tidal current or not?

Line 266: This is a bit of a water mass zoo – you provide references but you may say in one sentence the origin of MUW? and OSW?

Line 283: in which respect "stabilizes" the SAMW?

Line 287: besides the changes in the vertical structure due to the dynamical adjustment

of the density field - you need to also consider the movement of the mooring - it can be seen that he mooring moved down by several hundreds of meters - for a cyclone that means that sampling is done at very different isopycnals (as uplift and mooring subduction operate into a similar direction)

Line 291: is that the bottom? or how many meters above the bottom?

Line 294: please give again reference that provides the information why you would expect to find A-AAIW – has it been observed in the Cape Basin before?

Line 309: is the high correlation maybe because you used the data to estimate the GEMs?

Line 313: how is this number estimated and what tells us the number?

Line 320: This looks strange - 0.38% - I guess that is because you use the salinity and not a salinity anomaly. Would also be low for temperature if you use absolute temperature instead of temperature on the Celsius scale. So my question is what we should take home from a 0.38%? to me it first looks like insignificant... You may think of a better way in showing what you want to show. using anomalies??

Line 326: sometimes you give meters (m) sometimes dbar (not db please). Please use either one or the other (convert m in dbar or vice versa).

Line 330: to me this is very confusing - what is uplift, what is real warming? are these anomalies detectable in GEMS if they are only isopcynal heave?

Line 333: again, how much is heave and how much is a real anomaly? In case of an isopycnal heave the "release" of the anomaly in the event of a dissolution of the eddy would have 0 (zero) impact on the environment.

Line 385: uplift of isopycnals has nothing to do with water masses variability - the local uplift of isopycnals in a vertical homogenously stratified fluid (e.g. through a local geostrophic adjustment of the density field) will create an anomaly when looking in

depth/pressure space but no anomaly when looking in density space. This is of fundamental importance when it comes to discussion of transport of water mass anomalies. Water mass anomalies are only "real" when looking in density space. Note, it may be helpful to show the nature of the variability discussed here in the context of the 4 "cases" - also in a T/S diagram of the time series (Figure 5-8) - maybe embedded in the background TS (see my comment above). This would reveal immediately if this is a thermohaline anomaly of simply a heave.

line 393: what does "direct" and "indirect" mean - please precise.

Line 396: Indian Ocean

Line 407: could you please provide reference for the decorrelation scales

Figure 5-8: a TS diagram oft he anomaly and a progressive vector plot to determine how close tot he centre the crossing took place would be helpful.

Figure 11: anomalies versus density would be of more help to show

---

## Referee Comment (RC2) · Anonymous Referee #2 · 13 Dec 2017

This paper presents a description of mooring, CPIES, altimetry, and SST data in order to study the impact of eddies on water masses within the Cape Basin. The paper uses clear language and is easy to read. Unfortunately, I find that the analyses of the data presented do not generally support the conclusions, and that some interpretations (especially of the altimetry data) are rather questionable.

As an example, the abstract claims that the data of the SAMBA array are used to assess the "nonlinear, mesoscale dynamics of the Cape Basin". I am having difficulty to find where exactly in the manuscript such nonlinear dynamics are described and studied. Another example found in the abstract is that "future investigations with longer

time series ... will ... ultimately improve our understanding of the strength and variability of the Meridional Overturning Circulation". I am really wondering where in this article such a thing is suggested from the results. In summary, while the title reflects the content of the paper, most of the abstract does not. About the introduction: it is useful to provide some context about the MOC and the SAMBA experiment, but there is too much of it since this is not the topic of the paper. More background should be given about the hydrography and oceanography of the region, as well as the characteristics of eddies in this region.

Another aspect of the study that I am very concerned about is the use of the eddy detection algorithm of Laxenaire et 2017, a paper not yet accepted for publication at the time of submission of this paper. An example of puzzling result is found in Figure 7: are we to understand that anticyclonic eddy A19 has grown in size by an order of magnitude from 22-April-2015 to 26-April-2015, and that cyclonic eddy C14 suddenly appeared between these dates? Even if C14 is generated at the Benguela front, it does not appear at all like a coherent structure shed from the Benguela Current. I realize that the Cape Cauldron is a very energetic region where tracking eddies (in the sense of coherent structures) is difficult, but maybe another algorithm could be tested against the one of Laxenaire? Or maybe the atlas of eddies by Chelton et al. could be used? Also, does not it look observations at mooring M4 are more representative of the meandering Benguela current?

Another and final aspect of the study that I find worrying are the conclusions based on the temperature and salinity changes (Figure 11). I understand that the authors attempt to show more clearly the potential changes from mooring and CPIES data by showing percentages of changes (compared to which means?), but are we to be compelled by changes of salinity of the order of less than 0.5%? Or even temperature changes of the order of less than 10%?). Often, when alleged eddies are passing the moorings, I am not seeing much in the time series of Figure 3. How large are these changes compared to, let's say, the total standard deviation of the records, or again the standard deviations

outside of eddy "events"?

The authors have at their disposal a very valuable dataset of mooring and CPIES observations that are clearly under-utilized with this study. There have been methods developed to detect eddies in mooring data, the paper Lilly and Rhines 2002, Coherent Eddies in the Labrador Sea Observed from a mooring, comes to mind (https://doi.org/10.1175/1520-0485(2002)032<0585:CEITLS>2.0.CO;2). The CPIES data should also be able to tell you a lot more about the processes taking place such as transport, even if, as you state, the decorrelation length state is smaller (what is it?)

In summary, even if this paper is elegantly written, I am afraid that its scientific significance is poor to fair at this stage, that its scientific quality is poor to fair, and that its presentation quality is good. An important concern is that the abstract is not representative of the results.

Some more detailed comments:

l 15: nonlinear: what is demonstrated to be non linear here?

l 18: these mesoscale features: which ones?

l 24: substantial role of these mesoscale features: you have not shown how large are the "eddy"-related changes in T and S compared to ther sources of variability.

l 71: lead -> led?

l 101: moorings have a sub-surface depth of 500 m: I understand what you mean but this is an odd way of describing the mooring.

l 115-116: the authors know very well that a CTD does not measure salinity and density, please rephrase.

l 140: The Agulhas is traditionally said to retroflect, not reflect

l 165-66: what are the tolerances given? standard deviations or standard errors of the

mean or confidence intervals? please specify.

l 186: accurate: i would temper this by something like "reasonable". Do not forget that a correlation of 0.72 implies that about only 52% of the variance can be explained.

Figure 5 & 6 and others: please add units close to colorbars

Figure 10 is introduced before Figure 9? and Figure 10 is really not insightful.

Figure 10 should contain the definition of the water mass acronyms.

line 283: stabilizes? what do you mean? is present?

Figure 1: you should state in the caption that the CPIES estimate for the cyclone is missing

line 331: after its recovery and re-deployment: do you mean before the re-deployment?

line 365: what are the previous estimates?

line 408: the decorrelation length scale is smaller than this distance: and what is it?

line 411-413: "Future investigations with longer time series at these existing sites will lead to a better understanding of the eastern boundary current variability and Indo-Atlantic exchanges, and ultimately improve our understanding of the strength and variability of the AMOC."

Ok, this is for further investigation, but you are not even hinting how this could be done. How are your mooring observations going to inform you on the variability of the AMOC? How are you results on water masses and eddies going to be utilized when constructing the trans-basin array. If you want to talk about the AMOC this is what you need to discuss.

---

## Referee Comment (RC3) · Anonymous Referee #3 · 21 Dec 2017

Review of Moored observations of mesoscale features in the Cape Basin: Characteristics and local impacts on water mass distributions By M. Kersale, T. Lamont, S. Speich, T. Terre, R. Laxenaire, M. J. Roberts, M. A. van den Berg and I. J. Ansorge

This manuscript outlines the analysis of 14 months of data from an array of moorings and CPIESs deployed in the South Atlantic near the Cape Basin as part of a basin-wide South Atlantic MOC array along 34.5°S, together with satellite observations. Their analysis focuses on the identification of mesoscale features impacting the moored observations. The in situ data set is unique, and the analyses presented provide some useful insights about the regional circulation at the Cape Basin providing eddy statis-

tics that are in agreement with previous studies, and detailed analyses of case studies (anticyclonic eddy, cyclonic eddy, cold and warm water intrusions).

One concern is that there are many instances where the manuscript is imprecise and more details are needed for some of the methods. There is little (or no) information about configuration and set up of the instruments and calibration issues, how have the authors performed the quality control of the different types of data, or which are the associated error bars of the different measured variables. For the CPIES/GEM analysis the robustness of the relation between vertical profiles of temperature, salinity, density and measured tau is not shown, even though it is key to the analysis presented comparing mooring and CPIES data. For instance, it would be good to know how does the GEM perform in the deep ocean, how many deep casts are used to build the GEM, etc. The manuscript relies on an unpublished method (Laxenaire et al. 2017 submitted) for the eddy analyses. I suggest that the eddy detection method should be carefully explained to provide more guidance to the reader (for example, lines 128 and 154).

Another concern is that the authors express many times the significance of understanding these mesoscale features to understand MOC variability without really explaining the reason for that, and in particular (if it's meant for future work) how that will be accomplished. Most of the introduction is dedicated to the MOC and MOC arrays, but there is very little explained about the connection between the findings in this study and MOC variability. The main objective of the manuscript should be clearly stated and the introduction must be targeted to provide the necessary background for what will be presented in the following sections. If the focus of the paper will be on the local dynamics (as the title and results suggest) more background on regional aspects is needed. I suggest putting this work in the context of the SAMOC/SAMBA project strategy/objectives making it clear what the eastern part of the SAMBA array will address and how is it likely to contribute to improve the understanding of MOC variability and, specially, on which time scales.

My overall recommendation is that the manuscript needs to be revised and strengthened before publication.

Please find specific comments below.

The abstract needs to be revised in order to reflect the main findings of this work.

Line 14: change to South Atlantic Meridional Overturning circulation Basin-wide Array (SAMBA).

Please include previous literature regarding the main circulation patterns/boundary current systems in the Introduction. A schematic of the main currents/circulation patterns on Figure 1 could be useful.

Line 57: there are updated ACC transport estimates through Drake Passage

Line 64: It is not clear how the western boundary buoyancy anomalies have been looked at. Did these studies looked at the mesoscale eddy field?

Line 85: '. . .some accurate information' I suggest rephrasing this to try to make a better statement.

Lines 86-88: This a quite strong statement. I suggest leaving it for the outlook at the end of the manuscript with some explanation for why do you think that is true.

Line 93: The usage of the term 'tall' moorings is not clear. Would 'full depth' mooring suit better?

Line 108: 'destroyed and lost'. Maybe it was just 'lost'?

Line 113: replace 'various' with 'five'. Please add information about horizontal resolution of the hydrographic sections.

Line 116: Salinity and density are not directly 'measured' by the CTD. Please clarify.

Line 123: It may be worth clarifying here that satellite will give information of the dynamics at the surface.

Lines 135-152: Part of this could go to the Introduction.

Line 153: what is the 'C line'? Laxenaire et al method should be explained if their paper is still unpublished during the revision of this manuscript.

Line 169: How do you know this? Which is the approximate depth of the Ekman layer here? Are there any seasonal variations?

Lines 180-186: Which type of filtering is applied to compare the velocities from altimetry and velocity from ADCPs? How is aliasing by the sampling resolution of the altimetry addressed? The conclusion about the validation of the altimetry-derived velocities with the ADCP velocities should be approached with more caution, given the values for the correlation coefficients. A correlation coefficient of 0.7 leaves about 50% of the variance unexplained. It could be useful to show a comparison between the time series of the velocities from altimetry and from the ADCP to provide more faith in the validation analysis.

Line 189: Does the upper layer include subsurface waters? Please give a depth range

Line 276-277: Is ARMW a newly identified water mass or has it been identified before in all the studies mentioned there?

Line 229: Given the text structure below, I would replace 'four' case studies by 'three' case studies

Lines 266-267: '. . .to recover the regional water masses for each mooring'. This is not clear, what is shown in Figure 10? Is it the depth range occupied by each water mass? Figure 10 is not very informative. The main information from the figure could go to a Table (see below).

Lines 270-282: It would be nice to condense the information about water masses on a Table to make it easier for the reader to follow.

Lines 294 -295: Please add an interpretation

Line 310: Please add an interpretation about the low correlation coefficients found for the deeper levels.

Line 311: There are correlation coefficients values before in the text (Section 3.3), the description of the method for estimating their significance should be provided before.

Lines 318-322: It is not clear with respect to which means are these percent changes estimated from.

Line 353: I suggest changing 'high spatial and temporal scales' with 'high spatial and temporal resolution'

Line 365: Please add references for the previous estimates

Lines 366-367: 'accurate description'. . . this needs more evidence

Line 396: add 'Ocean' after 'Indian'

Line 408: what is the decorrelation scale number here?

Line 410: 'these first independent observations comparisons' Do you mean between ADCP and altimetry, and between moorings/CPIES?

Line 411: 'some confidence' Please try to rephrase.

In the Discussion, I think that a better case could be made describing which are the essential new findings of this study compared with previous studies, and what have we learned from this new data set.

Lines 411-413: It is not clear how this will be accomplished. How is the eastern boundary current system variability is going to be estimated from the present array configuration? How are these mesoscale features likely to impact MOC strength/variability? For instance, do you expect them to average out (or not) for the basin-wide MOC calculation?

Some general comments:

In general: To define an acronym first it is necessary to use the full words and then in brackets the acronym should be defined. For instance, Acoustic Doppler Profiler (ADCP).

I find it confusing that the authors use the same nomenclature for an instrument's site (for instance C1) and an eddy indentified as a cyclone (also C1).

Figure 4: For the caption I suggest adding 'for selected dates between' September 19, 2014 and October 15, 2015 ' . . .

Figures 6-8: This is perhaps a matter of style, but I strongly suggest providing complete figure captions.

Figure 9: Please specify that this diagram corresponds to the eastern part of the SAMBA array

Figure 11: For identifying water masses it is more appropriate to use neutral density as the vertical coordinate instead of depth.

---

## Author Comment (AC1) · 25 Feb 2018

**Answer to: Anonymous Referee #1**

Thank you for your detailed, helpful suggestions on our manuscript. As we noted in the letter to the editor, we found the comments of the editor and reviewers to be very constructive, and we feel that the revised manuscript is greatly improved now that we have addressed these suggestions.

Below we detail the changes we have made to the manuscript, addressing point by point all the issues raised. The comments from the reviewers are in bold, while our responses are interspersed between the comments in non-bold text. All line numbers indicated in our responses correspond to the new version of the paper.

**General comments**

First of all, I think the introduction needs re-writing. It is irritating that so much is written about MOC and the different MOC programs in the Atlantic and so little about the problem the study is concerned with. Why do we need to know that there is a MOVE or RAPID when analysing and interpreting TS anomalies in eddies in the Cape Basin? - How does that really is connected to the study here? If it would be I would expect that at the end of the paper I know how this study contributes to the problem. Maybe I overlooked it – but I can't find a contribution of the findings presented here to the MOC. I understand that the complete SAMBA/SAMOC array (full basin width) aims to measure the MOC variability at the respective latitude but really this study is looking into a "local" aspect of hydrography and flow variability in the Cape Basin. It just makes use of the mooring infrastructure/data. I strongly suggest to shorten the AMOC part (to maybe one sentence only) but to expand a lot more on what the study is concerned with (and what is also reflected in the title): A LOCAL study on the structure of eddies (on the mesoscale only) in the Cape Basin. To me this would also mean to expand in the introduction on eddy/eddy interaction (dipole discussion), eddy detecting in mooring data (quite a bit of studies have worked on that), calculation of anomalies (please pay attention to pressure versus density space (see comments below). As suggested, we have considerably revised and shortened the description of the MOC (**l. 34-36**), focusing mainly introducing the South Atlantic MOC (SAMOC) observing systems used in this study (1. 86-94). We added some additional words in the Introduction to explain why we mention this (l. 44-46), and at the end of the discussion (**l. 512-516**) we add some additional words explaining the contribution of the mesoscale nonlinear dynamics we are discussing to larger scale processes such as the AMOC.

Following your advice, we have also introduced in more detail the dynamical background of the Cape Basin, the processes and the local water masses exchanges associated with eddies, dipoles, and filaments **(l. 64-83)**. Finally, we introduce the different studies observing eddy with mooring data focusing on those using the technique of Lilly and Rhines (2002) **(l. 104-106)**.

**I would also suggest a brief introduction to GEMS in this set up - I understand that you used the T/S/pressure at the grid points (moored instrumentation) to set the local sound velocity points - the unknown would then most of the water column upward from the ADCP (upper 500m or so) - and which is were likely a lot of warm water (and thus sound speed) variability lies? Do you use satellite SST to give you one sound speed gridpoint at the surface?**

It appears that our initial version of the paper was not clear on this issue, our apologies. The T/S/P data from the moorings are not used in building the GEM fields. Instead, hydrographic profiles from the region (CTD and Argo) are used to build these simple two-dimensional look-up tables. Based on your suggestion, the introduction to the GEMs creation has been rewritten to clarify this point and moved to the Data and Methods section (l. 131-134). We have also added a new figure presenting the GEM fields of temperature and salinity (Figure 2), and we've added discussion about the scatter around the GEM fields (i.e. the estimated accuracy of the GEM values; see the new Figure 2c,d) to the text of the paper (l. 134-138).

**Specific comments**

**140-50: why do we need to know these details?**

**155-60: if you decide to leave in this part it would be useful to introduce HOW the mechanisms are behind MOC and water mass transformation and how precisely does that link to the analysis of eddies in the Cape Basin?**

We have considerably shortened this description as suggested, and we have introduced some additional wording to explain the contribution of the mesoscale nonlinear dynamics we discuss in this paper to larger scale processes like the MOC **(I. 34-36).**

**l64: are these "anomalies" similar to the anomalies that will be studies here? Or is the first a transport anomaly in the DWBC (western array) and here local anomalies due to mesoscale eddies in the eastern boundary current are studied?**

This sentence has been rewritten to clarify this point **(l. 97-99)**: "The transport and water mass anomalies associated with the deep current and migrating eddies near the western boundary have been well studied (e.g., Meinen et al., 2012, Meinen et al., 2017; Valla et al., submitted, 2018) but the eastern boundary anomalies have not yet been examined along 34.5°S."

**188: I am missing all references to research on the circulation elements along the eastern boundary and in the Cape Basin. Are there undercurrents? Is there a coastal current? Does it connect to the north poleward or south? I would expect that the South Africans and Namibians have worked quite a bit here and know about the local/boundary circulation? 189: Given the discussion later about eddies, dipoles, filaments etc. could you please introduce a bit more general wisdom on eddies and eddy/eddy interaction and including filamentation and how all that links to water mass transformation (isopycnal heave versus water mass anomalies)?**

As suggested, we have added a number of citations to literature linked to the dynamics of the Cape Basin and the processes and the local water mass exchanges associated with eddies, dipoles, and filaments **(l. 64-83)**.

**1103: the MicroCat's have no pressure? How did you add pressure information? (it is required for calculating salinity). What will be the uncertainty in Salinity? Please indicate that you did not use the oxygen (or did you?)**

We agree, our earlier version did not provide detailed information about which instruments included pressure data, and we did not mention that the oxygen data was not critical for this study. We have modify the text to clarify these issues **(l. 121-122)**. We have also added additional information in Table I to distinguish which sensors have a pressure recorder (depths in italics).

**1108: why do you know that the PIES was destroyed if it was lost?**

The PIES crashed hard against the hull of the vessel before being lost back into the sea, and subsequent attempts to contact it failed, so we are fairly confident that the glass sphere was breached and the instrument was destroyed. Nevertheless, we have simplified this sentence as the ultimate state of that instrument is not essential to this paper (**l. 127-128**).

**1109:** different approaches have been used in the past (e.g. creep function) – why linear? and, does it make a difference using other techniques? or maybe it is not relevant for your specific study (if so – make clear). Looks like we were not clear on this issue either - thank you for pointing this out. The type of pressure sensors used in the CPIES have been in use for some decades, and they have a well established method for correcting the drift (e.g., Watts and Kontoyiannis, 1990; Donohue et al., 2010; Meinen et al., 2013). Based on this previous work, we know this type of sensor typically exhibits either a linear or an exponential plus linear type of drift. We have followed the traditional technique discussed in the above studies for our pressure records. We have revised these sentences **(l. 128-131)** to better explain our approach.

**1120:** What is the conclusion on accuracy after the calibration? Is it homogeneous across the data sets? How have the Microcats been quality controlled? And what do you expect for their accuracy? We have now detailed the calibration process and specify the errors associated to each variables (**l. 167-171**).**

**1132: Please provide information about ADCP configuration and processing – depth allocation, compass calibration, depth cell length and ensemble lengths, number of bins, burst or spread mode?**

This information about the ADCP configurations has been added in Table I. The ADCP was configured in bursts mode and the compass calibration was done in September 2014, before the initial deployment of the tall moorings, with the software *RDI WinSC*. We have not repeated the calibration due to servicing the instruments at sea and the influence the vessel's metal would have on the calibration. When we did the initial calibration all the parameters were within the set ranges and the total error less than 5°.

**1134: for my understanding most of the content in this paragraph (3.1.) must be part of the introduction. 1139: is there maybe seasonal or other variability? What drives it? wind, topographic waves? Maybe a connection to the equatorial belt via waves?**

Part of the original paragraph 3.1 has been moved in the introduction as suggested. We also added **(l. 80-83)** more information about the forcing mechanism and the variability of this equatorward jet.

**1155-156: I would expect that to be part of the following chapter (3.2)**

The paragraph concerning the eddy statistics has been moved **I. 227-238**.

**1156: why is there such a large std on the propagation? Could you add a few words explaining that. Many stationary eddies? interaction with mean flow?**

Thanks for pointing this out – you are correct, some of the 'speeds' that we had estimated were artificially high because we had some spurious features that were being tracked that exaggerated our standard deviation. We have redone this calculation and we have been careful to avoid artificial translation due to the intense merging and splitting in the area. We have now a comparable mean and standard deviation  $(11 \pm 6 \text{ km day}^{-1})$  to that found in previous estimates (**l. 440-442**).

**1157: please add time period: "in 14?? months we observed...."**

The time period (~14 months) has been added **l. 231.**

**1169-170: how do you know that this is "usually below the Ekman layer"? what is your critieria for "Ekman layer"? did you compare ADCP shear with geostrophic shear? Did you calculate the Rossby number for the eddies? how important are the non-linear terms (Rossby number)? and how would that impact a conclusion on Ekman layer and other dynamics?**

CTD observations and shipboard ADCP measurements conducted during the various cruises along the eastern section of the SAMBA transect have been used to estimate the mean depth of the Ekman layer. From these measurements, we have noticed that the mean ADCP shear deviates from the mean geostrophic shear above 36 m depth, indicating the presence of ageostrophic wind-driven Ekman currents. We have used this criterion to define the base of the mean Ekman layer along the section as 36-m depth. The estimated mean value of the Ekman layer depth has been added to the text **l. 198**.

Concerning the Rossby number for the eddies, we have estimated this number. We have now added the Rossby number estimates from the Lilly and Rhines (2002) techniques to Table IV. A discussion about the importance of the non-linear terms has been added in the discussion (**l. 475-481**).

**1170: is only one (the last?) bin used or a mean over a number of bins (which ones)? has the data been corrected for tidal effects via a tidal model or only through the butterworth filter? what is the bin length? (ADCP configuration should be added to Data and Methods section – see comment above)**

Only the last bin has been used for the comparison with the surface geostrophic velocity derived from satellite altimetry. The data have been smoothed with a Butterworth filter, but it has been shown that the tidal currents are of relatively small amplitude in this region (Steele et al., 2010). Information about the ADCP configuration has been added to the Data and Methods section in Table I.

Steele, J. H., Thorpe, S. A. and Turekian, K. K. (Eds.). (2010). *Ocean Currents: A derivative of the encyclopedia of Ocean Sciences*. Academic Press.

**1189: could you pls provide a depth range here?**

We have modified this sentence to make clear that we are speaking about the upper-layer dynamics between 40 and 60 m depth (**l. 221-222**).

**1195: what do you mean by "dipole"? is that just neighbouring eddies or do the centres need to be at a minimal/critical distance? are there only cyclone/anticyclone dipoles or can there by a anticyclone / anticyclone dipole? more to add to Introduction.**

We agree with the reviewer, the definition of a dipole was not explicit in the earlier version. We have addressed this point by providing the definition of, and additional information on the dynamics of, a dipole in the Introduction (**l. 69-77**) and in Section 3.1 (**l. 244-246**).

**1197: Referenced to a discussion below in this section: It may be helpful to show the nature of the variability discussed here in the context of the 4 "cases" - also in a T/S diagram of the time series - maybe embedded in the background TS. This would reveal immediately if this is a thermohaline anomaly of simply heave.**

As suggested, the T/S diagrams of the time series for each case study have been embedded/added to the plot of the background T/S in the new version of the manuscript (Figure 10). During the period of the anticyclonic eddies, the temperature and salinity, and density values at the shallowest SBE Microcat sensor at M4 show the highest range of values of the all of the time series records. During the time period of the cyclonic eddy, the values are at the opposite end, i.e. the lowest recorded densities. While the signature of these two features is clearly separated, we do not definitively proves if these changes are associated with a thermohaline anomaly or a simple heave. We have added text to explain our assessments in the new version. (**I. 367-380**)

**1198:** in which respect "consistent" and "complete"? maybe saying "consistent and complete picture of the mesoscale dynamics in the Cape Basin"? Even if the basin would be filled with eddies it is not clear that the basin dynamics as a whole is similar to the dynamics of the individual components. As suggested we have modified this sentence to make clear that we are speaking about the mesoscale dynamics of the Cape Basin **1. 247**.

**1209:** you differentiate between propagating (transient?) and "quasi stationary" eddies? what are the thresholds applied for that? or does transient means that the eddy dissolves when crossing the array? Transient was used for describing the relatively short life duration of this eddy (less than a month) compared to Agulhas rings. We agree that this adjective has a wide application and we chose to delete it, as it was not essential.

**l221: what does "associated" mean for you?**

This sentence has been modified and "associated" has been deleted as it was potentially confusing. (l. 272)

**1227: in this summary you leave out all information about the temperature/filaments you discuss before - why? isn't that the key for the MOC connection?**

As suggested, we have added in the summary a sentence to review the number and characteristics of the filaments and intrusions detected. **1.278-279**

**1232: it would be helpful to get an idea about how well the centre of the eddy crossed the mooring. Progressive vector diagram type techniques can help to analyse that (see Lilly, J., and P. Rhines (2002), Coherent eddies in the Labrador Sea observed from a mooring, J. Phys. Oceanogr., 32, 585–598). Very much of your conclusion depends on who well the eddy centre was observed.**

We thank the reviewer for suggesting this additional eddy detection technique. We now applied the technique of Lilly and Rhines (2002) to detect eddies, filaments and dipole from the mooring measurements. Using this new method we have evaluated the Rossby number, apparent eddy radius, azimuthal velocity, mean direction, and velocity of the background flow for each of the features we discuss in the paper. Unfortunately, the result were indeterminate with regards to the distance of the eddy center from the mooring position. Nevertheless, the method did allow us to explicitly confirm whether the mooring was inside the core of the eddy or outside at each

time step. More details about this method and its application have been added to the text in several places. **l. 143-157** and **l. 285-308**.

**1251: what is the difference between a "core" speed and the maximum velocity magnitude? at which depth is the maximum velo. Found?**

We agree with the reviewer that the core of the eddy should not be dissociated from the maximum of the azimuthal velocity. We have deleted the previous definition of the core and applied the resulting estimates from the method of Lilly and Rhines (2002) in the revised manuscript.

**1253: How much of this decrease is caused by TS anomalies (relative to ambient water) versus the vertical displacement of isopycnals in the eddy caused by rotation (heave)?**

In this section (**l. 309-341**) and the related Figures 7, 8 and 9, we have now added the isopycnal displacements associated with each of the features that we discuss. Furthermore, by shifting our analysis onto neutral density surfaces, we can more explicitly identify which signals are associated with heave. We have revised our discussion in Section 3.4 to explain this.

**1257: is this a one-record-only speed? what depth? (this applies to all the numbers presented here (see cases above). with tidal current or not?**

All of the moored time series data presented in this study have been smoothed with a 2nd order Butterworth filter with a cut-off period of 72 hours, which will effectively remove all of the major tide constituents aside from the fortnightly tide. We have clarified this point in the Data & method section (**l.125-126**; **l. 141-142**). All of the maximal values listed are a one-record-only speed, but we have now extended the description in the text to include discussion of the vertical shear of the velocity. All the numbers presented in this section have now been more carefully explained, e.g. **l. 314-315**.

**1266: This is a bit of a water mass zoo – you provide references but you may say in one sentence the origin of MUW? and OSW?**

We agree with the reviewer than the origins of MUW and OSW were not clear in the previous manuscript. We added this information (**l. 347-350**), and to clarify this paragraph we synthesized all the information about the characteristics of the water masses in Table V.

**1283: in which respect "stabilizes" the SAMW?**

"Stabilizes" was replaced with "is present" (l. 363)

**1287: besides the changes in the vertical structure due to the dynamical adjustment of the density field you need to also consider the movement of the mooring - it can be seen that he mooring moved down by several hundreds of meters - for a cyclone that means that sampling is done at very different isopycnals (as uplift and mooring subduction operate into a similar direction)**

The different properties of each event from full-depth-moorings and CPIES measurements are now made in density space in the revised manuscript, allowing us to more explicitly distinguish between property changes due to mooring motion from those associated with ocean changes.

**l291: is that the bottom? or how many meters above the bottom?**

We deleted this sentence as the related Figure 10 was deleted based on our changes due to some of the other reviewer comments.

**1294: please give again reference that provides the information why you would expect to find A-AAIW – has it been observed in the Cape Basin before?**

We did not expect to find A-AAIW, but the absence of this water mass during the whole record confirms the different varieties of AAIW characterized by Rusciano et al. (2012). Nevertheless, we have deleted this sentence as it is not essential to this paper.

**1309: is the high correlation maybe because you used the data to estimate the GEMs?**

As explained in our reply to the general comments above, it appears that our initial version of the paper was not clear on this issue. The T/S/P data from the moorings are not used in building the GEM fields. Instead, hydrographic profiles from the region (CTD and Argo) are used to build these simple two-dimensional look-up tables, so the GEM fields and the T/S/P data from the moorings are completely independent.

**1313: how is this number estimated and what tells us the number?**

We agree, in the earlier version we were not clear about the method we used to estimate the number of degrees of freedom, nor did we mention its implications. We have now modify the sentence to explain that the number of degrees of freedom represents a measure of the autocorrelation in a time series, and hence the true number of 'independent' observations, and we added more background for its estimation. **I. 204-210**

**1320: This looks strange - 0.38% - I guess that is because you use the salinity and not a salinity anomaly. Would also be low for temperature if you use absolute temperature instead of temperature on the Celsius scale. So my question is what we should take home from a 0.38%? to me it first looks like insignificant... You may think of a better way in showing what you want to show. using anomalies??**

We totally agree, the percent of changes on temperature and salinity were not relevant in the previous version of the manuscript. We followed your suggestion and the ones of the editor in modifying this figure, i.e. we modified this figure to illustrate conservative temperature anomalies in °C and absolute salinity in g kg-1 with neutral density as the vertical coordinate instead of depth (Figure 11). The maximal anomalies reach 0.5°C in temperature and 0.15 g kg-1 in salinity (**l. 397-425**), which are much more significant for our conclusions.

**l326: sometimes you give meters (m) sometimes dbar (not db please). Please use either one or the other (convert m in dbar or vice versa).**

We agree, and we now referenced all the vertical levels according to depth (meters) values in the text and the figures.

**1330: to me this is very confusing - what is uplift, what is real warming? are these anomalies detectable in GEMS if they are only isopcynal heave?**

Concerning the distinction between uplift and real warming, we have now addressed this point by switching to a density vertical coordinate (Figure 11). Concerning this specific anticyclonic eddy, from the SBE37 MicroCat measured data, we have evaluated the changes in temperature and salinity and concluded that they are mostly influenced by the down-shift of the isopycnal layers (**1. 397-404**). From the reconstructed properties (GEM field), the warmer and salitier water mass trapped inside the Agulhas ring is visible in the upper part of the water column.

The GEM field technique will capture whatever is in the original hydrographic data sets (CTD and Argo). So to the extent that the hydrographic data sets capture these anomalies, the GEM field should be able to reproduce them. One limit, of course, is whether the anomalies are smaller than the scatter around the GEM field or not. We have now added this figure (Figure 2), and we've added discussion about the scatter around the GEM fields (Figure 2c, d) to the text of the paper (**l. 134-138**).

**1333: again, how much is heave and how much is a real anomaly? In case of an isopycnal heave the "release" of the anomaly in the event of a dissolution of the eddy would have 0 (zero) impact on the environment.**

We agree, and as explained in the comments immediately above, we now addressed this point by discussing the anomalies in density space. For this specific event (cyclonic eddy), we now show that the variability is mostly influenced by the up-lift of the isopycnal layers and water mass transport characteristics of I-AAIW. We have explained this in more detail in the revised text. (**1. 405-412**)

1385: uplift of isopycnals has nothing to do with water masses variability - the local uplift of isopycnals in a vertical homogenously stratified fluid (e.g. through a local geostrophic adjustment of the density field) will create an anomaly when looking in depth/pressure space but no anomaly when looking in density space. This is of fundamental importance when it comes to discussion of transport of water mass anomalies. Water mass anomalies are only "real" when looking in density space. Note, it may be helpful to show the

**nature of the variability discussed here in the context of the 4 "cases" - also in a T/S diagram of the time series (Figure 5-8) - maybe embedded in the background TS (see my comment above). This would reveal immediately if this is a thermohaline anomaly of simply a heave.**

We agree, and as explained in the comments immediately above we now address this point by evaluating the anomalies in density space (Figure 11). As suggested and detailed in our response to one of the earlier comments n, the T/S diagrams of the time series for each case study have been embedded/added to the plot of the background T/S in the new version of the manuscript (Figure 10). We have added text to explain our assessments in the new version. (**l. 367-380**)

**1393: what does "direct" and "indirect" mean - please precise.**

Direct or indirect were not properly applied in this sentence. We decided to delete these two adjectives.

**1396: Indian Ocean**

Fixed as suggested **l. 466**

**1407: could you please provide reference for the decorrelation scales**

We added this information to the new version of the manuscript (l. 507).

**Figure 5-8: a TS diagram of the anomaly and a progressive vector plot to determine how close tot he centre the crossing took place would be helpful.**

The T/S diagram of each case study is now shown in Figure 10. The hodographs and the progressive vector diagrams are now illustrated in Figure 6.

**Figure 11: anomalies versus density would be of more help to show**

Thanks for pointing it out. As noted in our response to one of the earlier comments, we have changed this figure to represent the anomalies versus neutral density.

---

## Author Comment (AC2) · 25 Feb 2018

**Answer to: Anonymous Referee #2**

Thank you for your detailed, helpful suggestions on our manuscript. As we noted in the letter to the editor, we found the comments of the editor and reviewers to be very constructive, and we feel that the revised manuscript is greatly improved now that we have addressed these suggestions.

Below we detail the changes we have made to the manuscript, addressing point by point all the issues raised. The comments from the reviewers are in bold, while our responses are interspersed between the comments in non-bold text. All line numbers indicated in our responses correspond to the new version of the paper.

**General comments**

**As an example, the abstract claims that the data of the SAMBA array are used to assess the "nonlinear, mesoscale dynamics of the Cape Basin". I am having difficulty to find where exactly in the manuscript such nonlinear dynamics are described and studied. Another example found in the abstract is that "future investigations with longer time series ... will ... ultimately improve our understanding of the strength and variability of the Meridional Overturning Circulation". I am really wondering where in this article such a thing is suggested from the results.**
It appears that our initial version of the abstract did not reflect the main findings of our work, our apologies. The abstract has been revised in the new version to order to clarify these different points better focus on the key results of the paper. We have added a bit more discussion of the "non-linear" terms in response to some of the other comments from the reviewers, as we have now evaluate the Rossby number associated to each feature (Table IV) and added in the text (**l. 475-481**) a discussion about its meaning. Also, we have considerably shortened the description of the AMOC in the paper, and so we removed the sentence you mentioned from the abstract. We kept at the end of the discussion a general perspective of this work in the context of Meridional Overturning Circulation future studies.

**It is useful to provide some context about the MOC and the SAMBA experiment, but there is too much of it since this is not the topic of the paper. More background should be given about the hydrography and oceanography of the region, as well as the characteristics of eddies in this region.**
As suggested, we have considerably revised and shortened the description of the MOC (**l. 34-36**), focusing mainly introducing the South Atlantic MOC (SAMOC) observing systems used in this study **(l. 86-94)**. Following your advice, we have also introduced in more detail the dynamical background of the Cape Basin, the processes and the local water masses exchanges associated with eddies, dipoles, and filaments **(l. 64-83)**.

**Another aspect of the study that I am very concerned about is the use of the eddy detection algorithm of Laxenaire et 2017, a paper not yet accepted for publication at the time of submission of this paper. An example of puzzling result is found in Figure 7: are we to understand that anticyclonic eddy A19 has grown in size by an order of magnitude from 22-April-2015 to 26-April-2015, and that cyclonic eddy C14 suddenly appeared between these dates? Even if C14 is generated at the Benguela front, it does not appear at all like a coherent structure shed from the Benguela Current. I realize that the Cape Cauldron is a very energetic region where tracking eddies (in the sense of coherent structures) is difficult, but maybe another algorithm could be tested against the one of Laxenaire? Or maybe the atlas of eddies by Chelton et al. could be used?**
It is true than the eddy detection algorithm of Laxenaire et al. (submitted, 2018) is still in review at this time as the paper had been re-submitted to JGR. We believe that method is good, but to address this comment and other comments from the reviewers, we chose a version of the algorithm comparable with the ones of Chaigneau et al. (2008, 2009) and Pegliasco et al. (2015). We have also added, in the Data and method section, the main changes compared with these original algorithms **(l. 178-188)**. This has led to some small changes in our interpretation (**l. 259-260**; **l. 265-266**) mainly associated with A19 that you mentioned in your comment. In the new version of the algorithm, A19 is not considered as a new structure but is still tracked as A16. This anticyclonic eddy shows a better coherence with time and similar size at the two dates mentioned (Figure 9). On April 26, the cyclonic eddy C14 is already detected with the new version of the algorithm (Figure 9-a). More details about the generation of

this eddy have been added to the text (**l. 265-266**). Finally, the presence of the structures at the mooring is supported by the Lilly and Rhines (2002) method, used to to detect eddies, filaments and dipole from the mooring measurements. More details about this method and its application have been added to the text in several places. **l. 143-157** and **l. 285-308.**

**Also, does not it look observations at mooring M4 are more representative of the meandering Benguela current?**
We agree with the reviewer, however this fact is not inconsistent with the presence of mesoscale eddies as most of the energy of the Benguela Current is supplied by eddy fluxes (Matano and Beier, 2003). South of 30°S, the observed Benguela Current is characterized by two ''streams'' separated by a conspicuous cyclonic meander (Veitch et al., 2010). The offshore stream, passing through mooring M4 is situated on the Agulhas eddy corridor (Garzoli and Gordon 1996). Indeed, passing anticyclonic eddies enhance the mean northwestward flow in this region. The meandering nature of the mean flow is therefore a manifestation of the preferential path of transient eddies in this region.

*Veitch, J., Penven, P., & Shillington, F. (2010). Modeling equilibrium dynamics of the Benguela Current System. Journal of Physical Oceanography, 40(9), 1942-1964.*

**Another and final aspect of the study that I find worrying are the conclusions based on the temperature and salinity changes (Figure 11). I understand that the authors attempt to show more clearly the potential changes from mooring and CPIES data by showing percentages of changes (compared to which means?), but are we to be compelled by changes of salinity of the order of less than 0.5%? Or even temperature changes of the order of less than 10%?). Often, when alleged eddies are passing the moorings, I am not seeing much in the time series of Figure 3. How large are these changes compared to, let's say, the total standard deviation of the records, or again the standard deviations outside of eddy "events"?**
We totally agree, the percent of changes on temperature and salinity were not relevant in the previous version of the manuscript. We followed the suggestions of the editor and other reviewers in modifying this figure, i.e. we modified this figure to illustrate conservative temperature anomalies in °C and absolute salinity in g kg$^{-1}$ with neutral density as the vertical coordinate instead of depth (Figure 11). The maximal anomalies reach 0.5°C in temperature and 0.15 kg m$^{-3}$ in salinity (**l. 397-425**), which are much more significant for our conclusions. The temperature and salinity anomalies due to the presence of the structures are calculated relative to the hydrographic properties in "normal conditions" (just before each event occurred).
To have an idea about the changes of T/S compared to the all time series, the T/S diagrams of the time series for each case study have been embedded/added to the plot of the background T/S in the new version of the manuscript (Figure 10). We have added text to explain our assessments in the new version. (**l. 367-380**)

**There have been methods developed to detect eddies in mooring data, the paper Lilly and Rhines (2002).**
We thank the reviewer for suggesting this additional eddy detection technique. We now applied the technique of Lilly and Rhines (2002) to detect eddies, filaments and dipole from the mooring measurements. More details about this method and its application have been added to the text in several places. **l. 143-157** and **l. 285-308.**

**The CPIES data should also be able to tell you a lot more about the processes taking place such as transport, even if, as you state, the decorrelation length state is smaller (what is it?)**
We added the decorrelation length state to the new version of the manuscript (**l. 505**). We totally agree with the reviewer that the CPIES data can be a lot more exploited for other processes as transport estimation. Nevertheless, this analysis is beyond the scope of the present study. We actually working on the variability of the Eastern Boundary currents, where the full data set from the 8 CPIES will be analyzed in this way.

**Specific comments**
**l15: nonlinear: what is demonstrated to be non linear here?**
As explained in our reply to the general comments above, we have added a bit more discussion of the "non-linear" terms in response to some of the other comments from the reviewers, as we have now evaluate the

Rossby number associated to each feature (Table IV) and added in the text (**l. 475-481**) a discussion about its meaning.

**l18: these mesoscale features: which ones?**
This sentence have been rewritten to clarify this point **(l. 18).**
**l24: substantial role of these mesoscale features: you have not shown how large are the "eddy"-related changes in T and S compared to their sources of variability.**
We agree, we now address this point by evaluating the anomalies in density space (Figure 11), allowing us to more explicitly distinguish which signals are associated with a thermohaline anomaly or a simple heave or down-shift of the isopycnal layers.

**l71: lead -> led?**
We agree, we deleted this sentence based on our changes due to some of the other reviewer comments.

**l101: moorings have a sub-surface depth of 500 m: I understand what you mean but this is an odd way of describing the mooring.**
This sentence have been rewritten. (**l. 118-120**)

**l115-116: the authors know very well that a CTD does not measure salinity and density, please rephrase.**
We agree, we have rephrased this statement (**l. 160-162**).

**l140: The Agulhas is traditionally said to retroflect, not reflect**
Thanks for pointing this out, we changed accordingly. **l. 40**

**l165-66: what are the tolerances given? standard deviations or standard errors of the mean or confidence intervals? please specify.**
The sentence has been rephrased to specify the given tolerances **l. 193**

**l186: accurate: i would temper this by something like "reasonable". Do not forget that a correlation of 0.72 implies that about only 52% of the variance can be explained.**
Thanks for pointing it out. Text has been corrected as suggested **l. 219**.

**Figure 5 & 6 and others: please add units close to colorbars**
We added units on the top of each colorbar for all the figures.

**Figure 10 is introduced before Figure 9? and Figure 10 is really not insightful. Figure 10 should contain the definition of the water mass acronyms.**
We totally agree. Figure 10 was deleted based on our changes due to your comment and some of the other reviewer comments.

**l283: stabilizes? what do you mean? is present?**
"Stabilizes" was replaced with "is present" (**l. 363**)

**Figure 11: you should state in the caption that the CPIES estimate for the cyclone is missing**
In the new version of the manuscript, we chose another cyclonic eddy as its signature was much clearer in the moorings data with Lilly and Rhines (2002) method. At that time, both the SBE Microcat's and the CPIES were recording.

**l331: after its recovery and re-deployment: do you mean before the re-deployment?**
We agree. We delete this sentence as it was linked to results not presented in the new version of the manuscript.

**l365: what are the previous estimates?**
We now provide the previous estimates with references related to this issue (**l. 440-442**)

**l408: the decorrelation length scale is smaller than this distance: and what is it?**
We added this information to the new version of the manuscript (**l. 507**).

**line 411-413: "Future investigations with longer time series at these existing sites will lead to a better understanding of the eastern boundary current variability and Indo-Atlantic exchanges, and ultimately improve our understanding of the strength and variability of the AMOC." Ok, this is for further investigation, but you are not even hinting how this could be done. How are your mooring observations going to inform you on the variability of the AMOC? How are you results on water masses and eddies going to be utilized when constructing the trans-basin array. If you want to talk about the AMOC this is what you need to discuss.**
At the end of the discussion (**l. 509-516**) we add some additional words explaining the contribution of the mesoscale nonlinear dynamics we are discussing to larger scale processes such as the AMOC.

---

## Author Comment (AC3) · 25 Feb 2018

**Answer to: Anonymous Referee #3**

Thank you for your detailed, helpful suggestions on our manuscript. As we noted in the letter to the editor, we found the comments of the editor and reviewers to be very constructive, and we feel that the revised manuscript is greatly improved now that we have addressed these suggestions.

Below we detail the changes we have made to the manuscript, addressing point by point all the issues raised. The comments from the reviewers are in bold, while our responses are interspersed between the comments in non-bold text. All line numbers indicated in our responses correspond to the new version of the paper.

**General comments**

**There is little (or no) information about configuration and set up of the instruments and calibration issues, how have the authors performed the quality control of the different types of data, or which are the associated error bars of the different measured variables.**
In the Data and Methods section, as suggested we expanded the description of the configuration, set up, and calibration of the instruments used in our analysis (Microcats, ADCP, CPIES, and GEM field). We have now detailed the calibration process and specify the errors associated to each variables (**l. 167-171**). The information about the ADCP configurations has been added in Table I.

**For the CPIES/GEM analysis the robustness of the relation between vertical profiles of temperature, salinity, density and measured tau is not shown, even though it is key to the analysis presented comparing mooring and CPIES data. For instance, it would be good to know how does the GEM perform in the deep ocean, how many deep casts are used to build the GEM, etc.**
We agree, given the importance of this point, a figure is perhaps required. We have now added this figure (Figure 2), and we've added discussion about the scatter around the GEM fields (Figure 2c, d) to the text of the paper (**l. 134-138**). As the reviewer suggests, in the deep ocean, the scatter around the GEM field is a larger fraction of the observed variability, however this is because the signals at depth are smaller rather than because the scatter around the GEM field itself is getting significantly larger.
To illustrate how many casts are available, we've added gray lines to Figure 2c, d.

**The manuscript relies on an unpublished method (Laxenaire et al. 2017 submitted) for the eddy analyses. I suggest that the eddy detection method should be carefully explained to provide more guidance to the reader (for example, lines 128 and 154).**
It is true than the eddy detection algorithm of Laxenaire et al. (submitted, 2018) is still in review at this time as the paper had been re-submitted to JGR. We believe that method is good, but to address this comment and other comments from the reviewers, we chose a version of the algorithm comparable with the ones of Chaigneau et al. (2008, 2009) and Pegliasco et al. (2015). We have also added, in the Data and method section, a description of the eddy detection method and the main changes compared with the original algorithms **(l. 178-188)**.

**[...]Most of the introduction is dedicated to the MOC and MOC arrays, but there is very little explained about the connection between the findings in this study and MOC variability. The main objective of the manuscript should be clearly stated and the introduction must be targeted to provide the necessary background for what will be presented in the following sections. If the focus of the paper will be on the local dynamics (as the title and results suggest) more background on regional aspects is needed.**
As suggested, we have considerably revised and shortened the description of the MOC (**l. 34-36**), focusing mainly introducing the South Atlantic MOC (SAMOC) observing systems used in this study **(l. 86-94)**. Following your advice, we have also introduced in more detail the dynamical background of the Cape Basin, the processes and the local water masses exchanges associated with eddies, dipoles, and filaments **(l. 64-83)**.

**I suggest putting this work in the context of the SAMOC/SAMBA project strategy/objectives making it clear what the eastern part of the SAMBA array will address and how is it likely to contribute to improve the understanding of MOC variability and, specially, on which time scales.**

We added some additional words in the Introduction to explain why we mention the MOC variabilty (**l. 45-47**), and at the end of the discussion (**l. 509-516**) we add some additional words explaining the contribution of the mesoscale nonlinear dynamics we are discussing to larger scale processes such as the AMOC.

**The abstract needs to be revised in order to reflect the main findings of this work.**
It appears that our initial version of the abstract did not reflect the main findings of our work, our apologies. The abstract has been revised in the new version to order to focus on the key results of the paper.

**Specific comments**

**l14: change to South Atlantic Meridional Overturning circulation Basin-wide Array (SAMBA).**
We changed the text accordingly (**l. 14**)

**Please include previous literature regarding the main circulation patterns/boundary current systems in the Introduction. A schematic of the main currents/circulation patterns on Figure 1 could be useful.**
As explained in our reply to the general comments above, we have now introduce the main circulation patterns. With this new introduction, we believe that the readers have enough information about the dynamics, so we decided to keep Figure 1 like this for the sake of simplicity, apart from the addition of the C-line of Laxenaire et al. (submitted, 2018).

**l57: there are updated ACC transport estimates through Drake Passage**
We agree, we've added updated literature linked to transport estimates through Drake Passage. **l. 90**

**l64: It is not clear how the western boundary buoyancy anomalies have been looked at. Did these studies looked at the mesoscale eddy field?**
This sentence have been rewritten to clarify this point **(l. 97-99)**: "The transport and water mass anomalies associated with the deep current and migrating eddies near the western boundary have been well studied (e.g., Meinen et al., 2012, Meinen et al., 2017; Valla et al., submitted, 2018) but the eastern boundary anomalies have not yet been examined along 34.5°S."

**l85: '. . .some accurate information' I suggest rephrasing this to try to make a better statement.**
We modified this sentence, pointing out to the experimental evidences of advected mesoscale eddies trough the mooring line **l. 102-104.**

**l86-88: This a quite strong statement. I suggest leaving it for the outlook at the end of the manuscript with some explanation for why do you think that is true.**
We followed the reviewer's recommendation and moved this statement at the end of the manuscript with additional explanations. (**l. 509-516**)

**l93: The usage of the term 'tall' moorings is not clear. Would 'full depth' mooring suit better?**
"Tall" was replaced with "Full-depth". We have carefully gone through the manuscript to make sure that we consistently use this new term.

**l108: 'destroyed and lost'. Maybe it was just 'lost'?**
The PIES crashed hard against the hull of the vessel before being lost back into the sea, and subsequent attempts to contact it failed, so we are fairly confident that the glass sphere was breached and the instrument was destroyed. Nevertheless, we have simplified this sentence as the ultimate state of that instrument is not essential to this paper (**l. 127-128**).

**l113: replace 'various' with 'five'. Please add information about horizontal resolution of the hydrographic sections.**
"Various" was replaced with "Five". We added information about horizontal resolution of the hydrographic sections to the new version of the manuscript (**l. 162-165**).

**l116: Salinity and density are not directly 'measured' by the CTD. Please clarify.**
We agree, we have rephrased this statement (**l. 160-162**).

**l123: It may be worth clarifying here that satellite will give information of the dynamics at the surface.**
This has been done accordingly **l. 174.**

**l135-152: Part of this could go to the Introduction.**
Part of the original paragraph 3.1 has been moved in the introduction as suggested.

**l153: what is the 'C line'? Laxenaire et al method should be explained if their paper is still unpublished during the revision of this manuscript.**
We agree, this is a great point. We have added the position of the C-line to Figure 1, and we have added some explanations in the text as well **l. 226-227:** "This line extends from the southernmost tip of Africa (Cape Agulhas) and, after crossing various seamounts, ends at 45°S in the Southern Ocean (Figure 1)."

**l169: How do you know this? Which is the approximate depth of the Ekman layer here? Are there any seasonal variations?**
CTD observations and shipboard ADCP measurements conducted during the various cruises along the eastern section of the SAMBA transect have been used to estimate the mean depth of the Ekman layer. From these measurements, we have noticed that the mean ADCP shear deviates from the mean geostrophic shear above 36 m depth, indicating the presence of ageostrophic wind-driven Ekman currents. We have used this criterion to define the base of the mean Ekman layer along the section as 36-m depth. The estimated mean value of the Ekman layer depth has been added to the text **l. 198.**
Concerning the seasonal variations, we compared the values of the Ekman layer depth splitting the casts according to the time of the year (two different seasons: September and December), and we did not find any seasonal variability.

**l180-186: Which type of filtering is applied to compare the velocities from altimetry and velocity from ADCPs? How is aliasing by the sampling resolution of the altimetry, addressed? The conclusion about the validation of the altimetry-derived velocities with the ADCP velocities should be approached with more caution, given the values for the correlation coefficients. A correlation coefficient of 0.7 leaves about 50% of the variance unexplained. It could be useful to show a comparison between the time series of the velocities from altimetry and from the ADCP to provide more faith in the validation analysis.**
It appears that our initial version of the paper was not clear on this issue, our apologies. For the altimetry data, we chose the closest point from the mooring position to do the comparison. We do not do any additional treatment to compare the data. We agree than our conclusion were to radical according to the different values of the correlation coefficient, and we now addressed them with more caution. We thought than another figure to compare the time series of the velocities from the altimetry and the ADCP would not bring new elements as the comparison is done for each case studies in Figures 7, 8 and 9. Finally, the presence of the structures at the mooring is also supported by the Lilly and Rhines (2002) method, used to to detect eddies, filaments and dipole from the mooring measurements.

**l189: Does the upper layer include subsurface waters? Please give a depth range**
We have modified this sentence to make clear that we are speaking about the upper-layer dynamics between 40 and 60 m depth (**l. 221-222).**

**l229: Given the text structure below, I would replace 'four' case studies by 'three' case studies**
We agree, we consider only three case studies in the new manuscript.

**l276-277: Is ARMW a newly identified water mass or has it been identified before in all the studies mentioned there?**
This sentence has been rewritten to clarify this point **(l. 353-356).**

**l266-267: '. . .to recover the regional water masses for each mooring'. This is not clear, what is shown in Figure 10? Is it the depth range occupied by each water mass? Figure 10 is not very informative. The main information from the figure could go to a Table (see below).**
We agree. Figure 10 was deleted based on our changes due to your comment and some of the other reviewer comments.

**l270-282: It would be nice to condense the information about water masses on a Table to make it easier for the reader to follow.**
We agree with the reviewer and to clarify this paragraph we synthesized all the information about the characteristics of the water masses in Table V.

**l294 -295: Please add an interpretation: "Interestingly, no A-AAIW is observed in the mooring records during the whole 450-day deployment."**
We did not expect to find A-AAIW, but the absence of this water mass during the whole record confirms the different varieties of AAIW characterized by Rusciano et al. (2012). Nevertheless, we have deleted this sentence as it is not essential to this paper.

**l310: Please add an interpretation about the low correlation coefficients found for the deeper levels.**
We have added an interpretation about the low correlation coefficients. **l.390-391**

**l311: There are correlation coefficients values before in the text (Section 3.3), the description of the method for estimating their significance should be provided before.**
We provided this information with the first estimate of correlation coefficients **l. 206-210**.

**l318-322: It is not clear with respect to which means are these percent changes estimated from.**
We totally agree, the percent of changes on temperature and salinity were not clear and relevant in the previous version of the manuscript. We followed your suggestion and the ones of the editor in modifying this figure, i.e. we modified this figure to illustrate conservative temperature anomalies in °C and absolute salinity in g kg$^{-1}$ with neutral density as the vertical coordinate instead of depth (Figure 11). The temperature and salinity anomalies due to the presence of the structures are calculated relative to the hydrographic properties in "normal conditions" (just before each event occurred).

**l353: I suggest changing 'high spatial and temporal scales' with 'high spatial and temporal resolution'**
Text has been corrected as suggested **l. 429-430**

**l365: Please add references for the previous estimates**
We now provide the previous estimates with references related to this issue **(l. 440-442)**

**l366-367: 'accurate description'. . . this needs more evidence**
"Accurate" was replaced by "reasonable" **l. 443**

**l396: add 'Ocean' after 'Indian'**
Fixed as suggested l. 466

**l408: what is the decorrelation scale number here?**
We added the decorrelation length state to the new version of the manuscript (**l. 507**).

**l410: 'these first independent observations comparisons' Do you mean between ADCP and altimetry, and between moorings/CPIES?**
We meant the first comparison between full-depth-moorings and CPIES data sets. The sentence has been improved accordingly **l. 509-510.**

**l411: 'some confidence' Please try to rephrase.**

"confidence" was replaced by "evidences"  **l. 510**

**In the Discussion, I think that a better case could be made describing which are the essential new findings of this study compared with previous studies, and what have we learned from this new data set.**
We agree with the reviewer. The discussion have changed to highlight the main findings of this study.

**l411-413: It is not clear how this will be accomplished. How is the eastern boundary current system variability is going to be estimated from the present array configuration? How are these mesoscale features likely to impact MOC strength/variability? For instance, do you expect them to average out (or not) for the basin-wide MOC calculation?**
At the end of the discussion (**l. 509-516**) we add some additional words explaining the contribution of the mesoscale nonlinear dynamics we are discussing to larger scale processes such as the AMOC. We also add information about the array configuration that we will used to analyze the  the eastern boundary current system variability**.**

**General comments (2)**

**In general: To define an acronym first it is necessary to use the full words and then in brackets the acronym should be defined. For instance, Acoustic Doppler Profiler (ADCP).**
Done accordingly. We carefully checked the different acronyms throughout the manuscript.

*I find it confusing that the authors use the same nomenclature for an instrument's site (for instance C1) and an eddy indentified as a cyclone (also C1).*
We agree. We have changed the nomenclature for the instrument's site. The CPIES are identified in the new version of the manuscript by the letters: P1, P2, P3 and P4.

*Figure 4: For the caption I suggest adding 'for selected dates between' September 19, 2014 and October 15, 2015 '. . .*
Done accordingly.

*Figures 6-8: This is perhaps a matter of style, but I strongly suggest providing complete figure captions.*
In the revised version of the manuscript, we provided complete figure captions.

*Figure 9: Please specify that this diagram corresponds to the eastern part of the SAMBA array*
Fixed as suggested

*Figure 11: For identifying water masses it is more appropriate to use neutral density as the vertical coordinate instead of depth.*
Thanks for pointing it out. As noted in our response to one of the earlier comments, we have changed this figure to represent the anomalies versus neutral density.

---

## Author Response (AR1)

**Answer to Editor - Dr Chapman**

We would like to thank the editor and the reviewers for their positive evaluation of the manuscript and for their constructive comments that have helped us improve the manuscript. We feel that addressing the questions and suggestions has led to a better quality manuscript and that the robustness of our results has greatly improved.

Below we detail the changes we have made to the manuscript, addressing point by point all the issues raised. The comments from the reviewers are in bold, while our responses are interspersed between the comments in non-bold text. All line numbers indicated in our responses correspond to the new version of the paper.

To address the general comments of the editor and the reviewers, we have considerably shortened the description of the AMOC, keeping only the text necessary to introduce the South Atlantic MOC (SAMOC) observing systems in the Introduction. We added, also as recommended, a description of the regional dynamics of the Cape Basin and the processes and the local water mass exchanges associated with eddies, dipoles, and filaments. In the Data and Methods section, as suggested we expanded the description of the configuration, set up, and calibration of the instruments used in our analysis (Microcats, ADCP, CPIES, and GEM field). As suggested by the reviewer, we have applied the method of Lilly and Rhines (2002) to the detection of eddies, filaments and dipoles with our mooring data. The eddy detection method of Laxenaire et al. (submitted, 2018) is under review at this time, and we believe it has significant positive aspects. To address comments from the reviewers, we chose a version of the algorithm comparable with the ones of Chaigneau et al. (2008, 2009) and Pegliasco et al. (2015). We added, in the Data and method section, a description of the eddy detection method and the main changes compared with the original algorithms. We are happy that the presence of the structures at the mooring is supported by both that method and the Lilly and Rhines (2002) method. Finally, as suggested by the reviewers, for identifying water masses we now use neutral density as the vertical coordinate instead of depth.

In the data and methods section, no mention is made of any pressure sensors on the instruments. However, in lines 260-261 and again later, it is stated that pressure sensors showed how well the system worked. A statement on these sensors needs to be added under "Data and Methods." We agree, we did not include enough information about which instruments/levels had pressure sensors in the previous version of the manuscript. We have modified this sentence to add the missing information in the new version of the text (l. 121) and in the revised Table I.

**Line 153 – the authors mention "the C-line (criteria of Laxenaire et al (2017)". As this paper is still in submission, they need to define the meaning and position of the C-line.**

We agree, this is a great point. We have added the position of the C-line to Figure 1, and we have added some explanations in the text as well **l. 226-227:** "This line extends from the southernmost tip of Africa (Cape Agulhas) and, after crossing various seamounts, ends at 45°S in the Southern Ocean (Figure 1)."

**Line 156 – do these rings really move at 7 +/- 91 km/day? This suggests some are incredibly fast. While the mean agrees well with translation speeds in Olson and Evans (1986), the standard deviation seems about an order of magnitude too high.**

Thanks for pointing this out – you are correct, some of the 'speeds' that we had estimated were artificially high because we had some spurious features that were being tracked that exaggerated our standard deviation. We have redone this calculation and we have been careful to avoid artificial translation due to the intense merging and splitting in the area. We have now a comparable mean and standard deviation  $(11 \pm 6 \text{ km day}^{-1})$  to that found in previous estimates (**I. 440-442**).

**Lines 1980-221 – this is a horrible paragraph! I suggest the authors break it up into several shorter paragraphs so that readers can focus more on the various rings.**

We have revised and rewritten this paragraph as several shorter ones (**l. 246-272**).

**Line 224 – this line makes no sense as it stands. Is there something missing?**

Thanks for pointing this out; as you had guessed, this was a typo/editing problem (l. 273).

**In section 3.3, the authors give the velocities of various rings. Strictly, the maximum for ring A13 should be -60.4 cm/s, not 60.4 cm/s, as written. The same may be true for other velocities. Please check.** We agree. We have carefully gone through the manuscript to make sure that we consistently use a positive value for the azimuthal velocity of an anticyclonic eddy in the southern hemisphere and a negative value for a cyclonic one.

**Lines 367-369 – the authors state that "the eastern mooring array is strongly affected by the regional intense mesoscale dynamics generated by instabilities of the South Benguela upwelling front…" but they haven't really given any evidence of this.**

We agree, we have modified this statement to make a more general comment at this point **l. 445-446**: "Analyses of both satellite and mooring data show that the eastern mooring array is strongly affected by the intense regional mesoscale variability." Concerning the origins of the mesoscale features, the eddy detection method allows us to detect the location of their generations. Most of the cyclonic eddies and some of the anticyclonic eddies are firstly identified along the South Benguela upwelling front (Figure 3). We changed the text accordingly to explain the location of the generation without any statement on the instability of the front, as it's true, we did not give any evidence of it.

**In Fig. 11, it might be better to show the temperature changes in °C rather than as a percentage, as is done for the salinity plots.**

We followed your suggestion and the ones of the reviewers in modifying this figure, i.e. we modified this figure to illustrate conservative temperature anomalies in °C and absolute salinity in g kg-1 with neutral density as the vertical coordinate instead of depth.

**References: Line 73 – Olson (1986) should be Olsen and Evans (1986); Galdyshev et al is in the ref list twice; McCartney (1977) in reference list but not in text?**

These mistakes have been corrected as suggested. Thanks for pointing them out.

---

## Referee Report (RR1)

The revised manuscript is an improvement from the previous version. I appreciate the changes made by the authors, especially in the introduction and methods' sections. I find the manuscript interesting and I would recommend it for publication in Ocean Sciences. I have some relatively minor comments that I recommend to address before publication:

Line 39: replace 'lives' with 'leaves' (is this what you meant?)

Line 45: remove 'a' before implications

Line 85: 'On regional scales, the circulation is important for water mass distribution, local dynamics, ecosystem assessments and air-sea interactions.' This seems to be too generic, I suggest giving more details about the importance of your study for ecosystem assessments, for instance (just a few words).

Line 90: I suggest adding Chidichimo et al (2014, JPO), they looked at the water masses distributions and variability in Drake Passage. Donohue et al (2016, GRL) looked at the mean total ACC flow.

Line 91: I suggest adding updated references about the arrays south of Africa (ASCA, CROSSROADS, GOOD HOPE). For instance Swart et al (JGR 2008), Hutchinson et al (2016 JGR). Some of the authors here are the same as in Hutchinson et al, but it is not clear to me why this study is not mentioned.

Line 161: '...in order to measure temperature and derive salinity and density.' ... this reads weird. The CTD measures conductivity, temperature and depth. Salinity is derived from the measured conductivity while density needs to be calculated from temperature and salinity.

Lines 170-171: Standard error has units, please check this sentence.

Lines 221-223: I am not convinced about the conclusions from the analysis of the degree of correspondence between the ADCP in situ data and altimetry, given the moderate correlations found, but I'll leave this to the editor's judgment.

Line 376: Please add 'record from the' after pressure

Title section 3.4: I think it is 'three' case studies based on the section above (I also recommend typing 'three' instead of '3' in the title)

Line 434; I think you refer to the SAMBA-east array 'region'?

---

## Author Response (AR2)

**Answer to Editor - Dr Chapman**

We are very pleased that the editor sees the improvements from the previous version. We have addressed the constructive comments of the editor and the reviewers in our revised manuscript, and we think these helpful suggestions have led to an improved manuscript with more robust results.

Below we detail the changes we have made to the manuscript specifically addressing the comments raised by the editor, tackling point by point all the issues raised. (Our responses to the reviewers are in the following pages.) The original comments from the editor are below in bold, while our responses are interspersed between the comments in non-bold text. All line numbers indicated in our responses correspond to the new version of the paper.

**First, I apologize for the time taken to reach a decision on this paper. The authors are to be thanked for the revisions they have made to the paper based on the initial comments of the three reviewers. The paper was sent put to all three for a second review.**
**While one of the reviewers is now happy with it, the other two have a number of questions that they believe still need to be addressed. The recent review by referee #2, in particular, lists a large number of comments. Many of these are simple matters of improving the English, but others, especially the interpretation of the altimetry and the identification of the different eddies shown in the figures, are more substantial. This point was also made by reviewer #3. I agree with them in this respect.**

We concur that it is important that we support well the altimetry results eddy identification. To address these concerns, first we have added a new figure (Figure 3) that shows that our ADCP observations are strongly consistent with the altimetry observations. This provides additional support for the altimetry results and our new eddy detection method by using an independent data set.

Second, concerning the identification of the different eddies, we agree that verification with multiple methods is required as tracking individual eddies through the Cape Cauldron region is challenging. In addition to our new method that we have been using, we have now used the Mesoscale Eddy Trajectory Atlas product, distributed by AVISO and based on the eddy detection method of Chelton et al. (2011), to verify the detection of all of the features we discuss in the paper. We feel that the results are now quite robust given consistent detection from two independent altimetry-based methods and solid comparison with the ADCP method of Lilly and Rhines.

**Some of these supposed eddies look more like intrusions (as discussed by Stramma and England, JGR 1999 or Shannon and Nelson 1996 in "The South Atlantic").**

In this new version of the manuscript, we have now gone back carefully to ensure that we are only interpreting transient eddies. We agree completely that the intrusions mentioned by the editor, relating to equatorward shelf-break front jet, the Cape Peninsula jet and/or the Benguela Upwelling jet, do complicate the analysis of the circulation in the region. To address this suggestion/comment from the editor, we added in the new version of the manuscript (l. 75-89) a more complete discussion of how the eddies we discuss fit into the complex circulation systems of the Cape Cauldron. We believe that the eddies that we discuss in the new version of the paper, all of which have been identified by both our new method and by the Chelton-type method, are coherent and are not artifacts or intrusions associated with the shelf jets.

**It may be that this region of the South Atlantic is not well suited for CPIES data because of the**

**number of eddies and intrusions of all sizes that pass through and complicate the interpretation of the GEM fields. Fig. 3, for example, I find totally incomprehensible, and I think the authors are trying to make too much of their descriptions of what is happening in Fig. 5.**

Concerning the CPIES data, in reviewing these results based on the editor and reviewer comments we discovered that we had some computational issues with how the time period for each event had been identified in the CPIES-GEM records as compared to the MicroCat records. We have corrected this flaw in our code and we have updated Figure 11 (now Figure 12).  While the comparison is still not perfect, which is to be expected since mooring motion cannot be perfectly removed from the MicroCat's data among other reasons, we find a much better agreement between the two data sets. The discussion relating to these results was also updated in the text (l. 576-581).

Regarding the complicated interpretation of the eddy fields, we agree that the number of features in the "Cape Cauldron" region make the interpretation of in situ data challenging, and in our earlier draft we were perhaps trying to discuss these fields in too detailed a manner. To simplify our discussion and to focus more directly on only those eddies that are crucial to the points we are making, the number of eddies that are described in the paper has been reduced, and we are now showing only the important eddy features which are identifiable by both our new method and by the Chelton-type method. We have clarified/simplified Figure 3 (now Figure 4), and we believe that all the trajectories of the eddies are now distinguishable. Figure 5 (now Figure 6) was also updated to plot only the eddies that are detected by both of the two eddy detection methods. The definition of the eddy boundaries was also changed to the location of the maximal azimuthal velocity to have a more dynamical view of the eddy structures.

**If the authors can demonstrate a better case for their conclusions, I am prepared to reconsider this manuscript, but it will need to go out again for review. Until then, however, I do not think they have demonstrated the utility of CPIES in this region or explained what is happening.**

We believed than our conclusions are substantiated by our independent analysis of altimetry data and moored in situ measurements. This work presents the first independent observations comparison between full-depth-moorings and CPIES data sets within the eastern South Atlantic region that gives some evidence of eastern boundary buoyancy anomalies associated with migrating eddies. We should also point out that PIES and CPIES have been used quite successfully in the region in the past (see the Baker-Yeboah, Garzoli, and Elipot papers cited in the Introduction and Data and Methods).  With the new material we have added, we think we have now demonstrated that the CPIES are working and that the story we're telling is robust.

**Answer to Referee #1: Dr. Johannes Kartensen**

We are very pleased that Dr. Kartensen sees the value of our manuscript. We have addressed his final comments in this new revised draft. Below we respond to each of his specific suggestions: the comments from the reviewer are in bold, while our responses are interspersed between the comments in non-bold text. All line numbers indicated in our responses correspond to the new version of the paper.

**I am happy with the correction the authors applied and think the manuscript is now ready for publication. However, I have some few final comments:**

**line 39 – I guess you mean 'Agulhas Current leaves the ….'**
Yes, "lives" was replaced with "leaves" (l. 35)

**line 46 – ' … implications for climate (e.g. Beal…'**
Fixed as suggested (l. 41)

**line 136 – didn't you used absolute salinity (S_A)? – hence the unit is gr/kg (not psu)**
Thanks for pointing this out. In the revised manuscript, all of the salinity values are now reported in absolute salinity. We updated Figures 2, 8, 9 and 10 and the corresponding text.

**line 171 – is the uncertainty really smaller for SA then T?**
Unfortunately there were some issues with the CTD calibrations from the R/V Algoa cruise we discuss that cannot be fixed after the fact, and our MicroCat calibrations are based on those CTD calibrations. Based on this comment and a comment from one of the other reviewers, we have gone back over the accuracies and calibrations very carefully, and based on our review the temperature uncertainties are in fact a little larger than the salinity uncertainties. Because these uncertainties are unsatisfying to us as well, we have now also taken the additional step of comparing our deep (> 2000 dbar) calibrated data from the MicroCat sensors with the deep high quality observations of the WOA climatology. Variations in the deep ocean are known to be much smaller than in the upper water column, so by comparing the calibrated deep sensors to the relatively stable deep temperature and salinity we can get the upper bounds of the calibration errors. Based on these comparisons we identified an offset between the calibrated deep MicroCat values and the mean WOA climatology values of 0.14°C for temperature and -0.01 g kg$^{-1}$ for absolute salinity. While these offsets are unpleasantly large, being much larger than the commonly achieved accuracies for these instruments, the signals being analyzed in this study are much larger than these potential errors, so we are confident in the results we are presenting.

**Answer to: Anonymous Referee #2**

We are very pleased that the reviewer sees the improvements from the previous version. We have addressed his/her constructive comments and we think this has helped us improve the manuscript. Below are our responses to the reviewer's specific comments: the comments from the reviewer are in bold, while our responses are interspersed between the comments in non-bold text. All line numbers indicated in our responses correspond to the new version of the paper.

**General comments:**
**The authors should be commended for undertaking a major revision of this manuscript, addressing several of the points raised during my review of the original manuscript. However, many aspects of this paper remain problematic.**

**First, the main issue concerns the definition of eddies as captured by the still unpublished Laxenaire 2018 manuscript. Despite using the method of Lilly and Rhines, as suggested, I am still not convinced that the features identified by A13 and A16, as an example, really constitute coherent "eddies". I think it might be a too difficult approach to be willing to define such features in the "Cape Cauldron" region. Since the Laxenaire 2018 manuscript is not published yet, we do not know its performance. This is demonstrated by Figure 3 which is really incomprehensible. I think the authors would be better off talking in general about mesoscale features affecting the mooring measurements. In a revised, or different, version of this manuscript, the authors should start with analyzing more scrupulously the time series of temperature, salinity, and current rather than taking the approach of analyzing altimetry first. If the authors would like to pursue the coherent eddy approach, I suggest they use the Chelton database as a starting point.**

We acknowledge the reviewer's point that tracking individual eddies through the Cape Cauldron region is challenging, and we agree that to do it right, verification with multiple methods is required. We have now identified the eddies that we discuss using two independent methods: the new method we are presenting here for the first time (which we were calling the Laxenaire et al. (2017) method in the first version of this paper); and the Mesoscale Eddy Trajectory Atlas product, distributed by AVISO and based on the method of Chelton et al. (2011). The Laxenaire method still has not been published – but sometimes papers based on student's dissertations take time – this is why we have included additional details of the methods in the present paper. The eddies that are discussed in the present paper are all identifiable by both the new method and by the Chelton-type method. The case studies selected for inclusion in the paper were chosen because they were the largest, strongest, events in the time series. Furthermore, once these features were identified, they were also analyzed via the Lilly and Rhines method using the moored ADCP data. All three methods provide consistent interpretations of these features, so we feel these results are robust.

We also agree that the number of features in the "Cape Cauldron" region that we had been displaying in the old version of Figure 3 made it somewhat hard to understand. We have clarified/simplified this figure (now Figure 4), and we believe that all the trajectories of the eddies are now distinguishable.

**Second, I have concerns about the processing and calibration of the microcat data. In my detailed comments below I have many questions about the description of the processing of the data. Most of the statistics and numbers given do not make much sense at all.**

We agree and we apologize for all of the confusion and for the lack of clarity in our previous explanations. After a lengthy discussion amongst all of the co-authors, we agree that there are some issues/problems with the calibration of the CTD data that was collected on the R/V Algoa cruise that we use in this study. And because those CTD casts were used to calibrate the MicroCat data, there are also issues with that data. Unfortunately, hindsight is 20/20, and we cannot go back in time to change this calibration at this time. Rather than just give up and throw away the data, we have found a way to independently estimate an upper bound accuracy estimate for the MicroCat data – i.e. comparison of the deep (> 2000 dbar) data where the observed signals in the ocean are small to the mean vertical distribution of temperature and absolute salinity in the WOA climatology. Fortunately, the key signals that we are discussing and interpreting are much larger (in most cases at least an order of magnitude larger) than the 'worst case' accuracy estimates.  Below in our responses to your detailed comments we explain the details of the testing we did and the changes we have made to the manuscript.

**Figure 11 may also illustrates the issue as well. Why are the data from the microcats so distinct from the data from the CPIES (for the lightest density classes)?**
Thanks for pointing this out, this was a very helpful question as it helped us identify an error in our code! We had some computational issues with how the time period for each event was identified in the CPIES-GEM records as compared to the MicroCat records. We have corrected this flaw in our code and we have updated Figure 11 (now Figure 12).  While the comparison is still not perfect, which is to be expected since mooring motion cannot be perfectly removed from the MicroCat data among other reasons, we find a much better agreement between the two data sets. The discussion relating to these results was also updated in the text (l. 576-581).

**Finally, the whole topic of nonlinearity is dubious. The Rossby numbers provided (~0.1) suggest that linear geostrophic dynamics are taking place. What is left is the characterization of eddies as "nonlinear" as defined by Chelton where the translation speed is less than the tangential speed and water parcels are expected to be trapped within the eddies. I would have liked to see more clearly in the manuscript the connection with water mass intrusions in the mooring data.**
We agree, that the different definitions of non-linearity can be confusing – this is a common problem within the broader literature in the field of course, but we have tried to improve our discussion of these topics in our paper. We have clarified in the Introduction (l. 31-32) that when we refer to "nonlinear mesoscale eddies" we are using the definition based on the advective parameter of Chelton et al. (2011). This definition is maybe the most pertinent in the context of our present paper (l. 551-554), since it determines whether an eddy identified in altimetry data can advect a parcel of trapped fluid (Fierl, 1981).  The work of Chelton and others has also demonstrated that altimetry data is adequate for investigating the dynamics of these kinds of quasi-geostrophic-dominated mesoscale eddies.  The

distinction with the ideal case of a purely linear flow (Rossby number = 0) has also been detailed in Section 2 (l. 172-174). Finally, we have explained (l. 547-549) that features associated with a Rossby number ~ 0.1 are not highly nonlinear or ageostrophic, but that the features are nonlinear in the sense that the corresponding Rossby number is not nil, so some ageostrophic processes are occurring.

**Detailed comments:**

**l14-15: can delete "the latitude"**
"the latitude" has been deleted (l. 15)

**l16: official name is "Current and Pressure Recording Inverted Echo Sounder"**
As suggested we have modified the full-name of the CPIES (l. 16)

**l24-25: "Under three case studies, the full-water column hydrographic properties of each mesoscale feature has been evaluated.": I am not sure what this sentence means. Do you imply that with only 3 case studies you evaluated the properties of ALL the mesoscale features that passed your array? Is this demonstrated in the paper? I believe this sentence could be deleted from the abstract.**
We agree with the reviewer that this sentence was not relevant and did not clearly highlight the impacts of numerous mesoscale features in the area. As suggested the sentence has been deleted from the abstract.

**l34: what is "large-water mass distribution"? Do you mean large-scale water mass distribution?**
Thanks for pointing it out, as suggested we modified this sentence to describe the scale of the distribution (l. 29)

**l39: lives -> leaves**
"lives" was replaced with "leaves" (l. 35)

**l44: implications are for "climate" or "climate processes", not "studies"**
"studies" was deleted (l. 41)

**l52: its? do you mean theirs?**
"its" was replaced with "theirs" (l. 48)

**l54: "on pre-AVISO" -> "before". AVISO is not the same as altimetry.**
Thanks for pointing it out. To correct and clarify our statement, we replaced "on pre-AVISO altimetric statistics" by "statistics from the early constellation of altimetry satellites that resolved fewer features than the modern constellation" (l. 50-51)

**l60: "distinguished" is not the right word: "characterized", "identified" maybe?**
"distinguished" was replaced with "characterized" (l. 58)

**l62: "mixing": what type of mixing? vertical? horizontal?**
This sentence has been rewritten (l. 61) to provide clarifications about the horizontal mixing at intermediate depth associated with the co-existence of cyclonic and anti-cyclonic eddies.

**l67: "described"? Do you mean "found"? "identified"?**

Yes, "described" was replaced by "identified" (l. 64)

**l72: " theoretically described by numerical models": what does this mean?**
Thanks for pointing it out. The sentence has been rephrased (l. 68-69) to dissociate analytical theories from numerical simulations.

**l46 to l83: Could you break down this very long paragraph into several paragraphs? The last part is an extensive literature review, which is welcomed, but is not well organized.**
As suggested, this paragraph had been broken down into several paragraphs (l. 42-91). The last part was re-organized to focus on the interactions between the mesocale dynamics and the eastern boundary flow regime (l. 75-89).

**l101: how is it "ideal"?**
We rephrased this sentence to explain what we mean regarding the ideal position of the array (l. 112).

**l111: Current and Pressure Recording Inverted Echo Sounder**
As suggested we have modified the full-name of the CPIES (l. 123)

**Table 1: What is "ensemble lengths"?**
To clarify this technical terminology, "ensemble lengths" was replaced by "number of records" (p. 41)

**l121: "Some of these instruments also have pressure recorders ... but these sensors will not be used in this study": you did not use the pressure data to calculate salinity from conductivity? Did you use a nominal pressure/depth for the calculation? Was there any blowdown of your moorings?**
It appears that our previous draft of the paper was not clear on this issue, our apologies. Yes, we used the pressure recorders to calculate salinity from conductivity and to account for blowdown of the moorings. This sentence has been rewritten to clarify this point (l. 135). You can see in Figure 10c,d the blowdown of the mooring.

**l131: Why did you calculate daily averages when you already applied a low pass filter with a 3-day cut off? Calculating daily averages amounts to low pass filtering so it is redundant.**
We agree that our sentence was confusing. The data were not daily averaged but sub-sampled at one value per day. "sub-sampled to a daily averaged value" was replaced by "sub-sampled to one value per day" (l. 144).

**l133: What is the region from which you used hydrographic data to build the GEM? That is important information what should be indicated.**
We agree, the positions of the CTD and Argo casts used to create the GEM field have been added in Figure 2-a.

**l139: Beal et al. 2015 and Elipot and Beal 2015 also applied successfully the GEM method with CPIES in the nearby Agulhas Current.**
As suggested, we added citations of the two studies that had previously applied the GEM method to CPIES across the Agulhas Current off the southeast coast of South Africa (l. 154).

**l139: \tau: have you defined it earlier as travel time?**
You are right, we had not defined it earlier. "\tau" was replaced by "travel time" in this sentence (l. 154) as we did not use the tau symbol at any later point in the manuscript.

**l140: Tab?**
Thanks for catching that typo; "Tab" was replaced by "Table" (l. 155)

**l148-150: "This calculation leads to the estimate of the advection direction \Theta as the largest velocity is expected in the direction perpendicular to the mean advection flow (direction of V_n).": this sentence is odd, could you rephrase?**
We agree, this sentence was rephrased (l. 164-165).

**l166: Please update the van den berg 2015 reference by indicating where the report cab be found. I was actually able to find online the report for the RS Algoa voyage 221: I could not find anything in this report on the CTD calibration and quality control. This raises questions on the quality of the data and the validity of the microcat calibrations against the CTD data.**
As suggested, we have added a html link to provide access to the cruise report of van den Berg (2015) in the Bibliography (l. 836). We agree than the cruise report only mentioned the collection of discrete seawater samples for a future calibration of the CTD. This statement had been clarified in our revised manuscript (l. 184-185). The calibrations are done after every cruise as soon as the samples are processed without publishing an additional report. The residual error for the salinity, determined from the regression between the CTD and the discrete sample salinity values, is 0.003 psu. This value is close to the WOCE standard of 0.002 psu. We can not provide calibration estimates for the temperature, as you do not have a reversing thermometer.  And so, we can not disentangle the differences between CTD salinity and hydrographic bottle salinity from errors in conductivity, temperature and pressure measurements.

**MicroCat Calibration**

- **l167-173: The whole method of calibrating microcats originates from Kanzow et al. 2006 (doi:10.1016/j.dsr.2005.12.007). Please refer to the methodology in this publication and to the accuracy requirements for the instruments.**
- **l167: "were within an envelope of 0.005 wide"? I do not understand this, please explain better. What are the units?**
- **l168-173:"Once calibrated?" What? the CTD? The final processing of all data is surely done after the cruise; please revise.**
- **"were attached to the CTD": you mean the rosette? the package?**
- **"When regressed with the CTD temperature and salinity, all the SBE MicroCat's sensors had a root mean square (rms) values greater than 0.9999 for both temperature and salinity.": This does not make any sense. Are you talking about regression coefficients? RMS value of 0.999 in which units?**
- **"The average standard error for the temperature (salinity) regressions were equal to 0.012 (0.002).": Not sure what you mean here. Are you talking about residuals?**
- **The SBE 37-SM Microcat has an initial temperature accuracy of 0.002 C (see manual of instrument). If your standard error for temperature is 0.012 C based on your regression/calibration with the CTD, isn't this worth commenting on?**

As we mentioned in our reply to the reviewer's general comment above, we agree and apologize for all the confusions. We are not providing an answer point by point to all your very valuable comments as we have completely reevaluated our discussion of the calibration and rephrased most of the associated paragraph.
As suggested, we have added a citation to Kanzow et al. (2006) concerning the calibration method of

the MicroCat's data in the upper part of the water column (l. 185-186). As we mentioned earlier, there are issues with the CTD and MicroCat calibrations that we cannot go back and address after the fact. To develop 'worst case' estimates of the accuracy of the calibrated sensor records, we compared the deep (> 2000 dbar) temperature and absolute salinity from the calibrated MicroCat records to the high quality deep data from the region in the WOA climatology. Based on these comparisons, we have estimated final accuracies of 0.14°C for the temperature and -0.01 g kg$^{-1}$ for the absolute salinity. These 'worst case' accuracy estimates are unpleasantly large, one or two orders of magnitude larger than the instrument accuracies, but they are still relatively small (in most cases one order of magnitude or more) compared with the key signals we are analyzing in this study (e.g. see the new Figure 12). We have added gray bars to Figure 12 to illustrate which signals are statistically meaningful and which are not.

**Here you give numerical values for salinity. Can you give first numerical values for conductivity?**
We feel that it is more appropriate to focus on salinity, rather than report conductivity values, because a) salinity is more commonly reported in the literature, and b) the CPIES-GEM data only has salinity (i.e. no conductivity GEM fields have been, or really could be, built). For these reasons we think we should stay with salinity - however as the reviewer has suggested elsewhere, we have improved our error analysis of the salinity - which we think is the point that the reviewer was driving towards here.

**Figure 2: It would be better for legibility to set the contour colors to "none" (if using Matlab as an example) and use discrete (instead of continuous) color scales. Water depth is usually positive, as in your Table 1.**
Figure 2 was modified as suggested.

**l174: "In addition": to what?**
"In addition" was deleted

**l181: extremum -> extrema?**
"extremum" was replaced by "extrema", thanks (l. 205).

**l182: Only 1mm? Are you sure this is the right value? Eddies should be associated with SLA on the order of 10 cm or so.**
We agree that our sentence was confusing. The 1 mm value is a increment (decrement) in the algorithm to look for closed contours around each possible cyclonic (anticyclonic) center as in the original algorithm of Chaigneau et al. (2008, 2009). This sentence was rephrased (l. 205-206).
We agree than given the accuracy of satellite data, a minimum of amplitude of 1 or 2 cm can be applied to filter the eddy once detected (e.g., Chaigneau et al.,2008, 2009; Chelton et al., 2011). Nevertheless, Faghmous et al. (2015) have shown that eddies associated with amplitudes inferior than 1cm can be very coherent over time. Following the filtering criteria of Faghmous et al. (2015) for regional study, we only consider eddies with a minimum lifetime of 7 days. We have added a brief sentence to the text explaining this (l. 208-209).

**l194-195: do you mean Figure 4?**
Yes, it was Figure 4, thanks. As suggested by another reviewer, a new figure was added (new Figure 3) for this specific paragraph. We have carefully gone through this paragraph to make sure each call to a Figure is correct.

**l228: "the median radius and standard deviation of these features are equal to 85 ± 43 km (66 ± 38 km considering the solid body rotation)": the two type of analyses (solid rotation or not?) are**

**not necessarily obvious. This requires more explanation. Please expand.**
We added some additional explanation regarding the two definitions of the eddy radius in Section 2 (l. 209-212).

**l232: It looks like Figure 4 might have been introduced before Figure 3?**
We have gone through the paper carefully to make sure the Figure order corresponds to the order in which we discuss them, thanks!

**Tab III caption: generate -> generated?**
"generate" was replaced by "generated" (l. 940)

**l242: Sometimes you spell "Figure", sometimes you use "Fig"**
"Figure" was replaced by "Fig." throughout the manuscript now, thanks.

**Figure 3 is not very insightful: Take as an example the blue lines connecting C8, C9, and C25: are we supposed to be able to distinguish where these eddies went?**
We agree that Figure 3 (now Figure 4) was not clear. To improve the readability of this figure, we have separated the trajectories of the cyclonic and anticyclonic eddies into separate panels. We also put in bold the trajectories of eddies after a merging. These improvements will help the reader to distinguish the different trajectories of the eddies. The number of eddies plotted had been also reduced as we represent only the features detected by the two different eddy detection methods.

**Figure 5 caption:**
**l764: "are identified as close enough to affect the measurement at the different moorings.": I think you mean something like: "are shown to affect the measurements ..." (even if I am not always convinced)**
We deleted this specific sentence in the caption as we modified Figure 5 (now Figure 6) to represent the eddies detected by the two different eddy detection methods, and give some additional confidence to our results.

**Figure 6 caption:**
**l771: dote -> dots?**
"dote" was changed to "dots" (l. 890)
**l772: same ... than -> same ... as**
"same ... than" was replaced by "same ... as" (l. 891)

**l257: A12 is really dubious in Figure 5i: Figure 5 does not show all contours of ADT so it is rather difficult to assess what the Laxenaire 2018 algorithm does. However, it can be seen that the western side of this anticyclone has no velocity signature, nor a SST signature. It looks like the east side is a branch of the Benguela current. I am finding this sequence of eddies hardly convincing. How can you explain the size of A13? It goes back to my comments about the original version of this manuscript.**
Figure 5 (now Figure 6) was updated to plot only the eddies that are detected by both of the two eddy detection methods. The definition of the eddy boundaries were changed to their maximal azimuthal velocities definition to have a more dynamical view of their structures. For example, the shape of A12 (now called A3, Figure 6c) is now coherent, and its boundary is associated with its velocity signature. The size of A13 (now called A4, Figure 6d) was clearly reduced with this new definition.

**l266: which intense dipole? April 18?**

The two eddies associated with this dipole were now specified (l. 209)

**Figure 4: As it stands, I am not convinced that the time series of u and v really allow the reader to see the influence of the alleged eddies. Maybe the Figure could show stick vector diagrams instead of u and v time series curve?**
We agree, stick vector diagrams have now been used in this figure (now Figure 5) to highlight the rotation effect of the eddy. We also added, as suggested by reviewer 3, the full time series of the u and v time series curves at the surface for the comparison with altimetry data (the new Figure 3).

**l289-290: "A filament or a front is also characterized by a straight line as for an eddy sliced through its exact center.": as? What do you mean? Do you mean that a filament or an eddy sliced throught its center lead to the same type of hodograph?**
Yes, we mean that a filament or an eddy sliced through its center leads to the same type of hodograph. The sentence was clarified as suggested to make this more clear. (l. 323-324)

**l293: "impulsive-like"? impulsive does not describe a visual feature but an emotional state. Do you mean a pulse maybe? Please explain better.**
The term "impulsive-like feature" was used by Lilly and Rhines (2002) to describe the hodograph of a dipole as we had done in our previous draft, however we see the point the reviewer was making, so we have replaced the term by "pulse" (l. 327).

**l294: "theoretical"? That has more to do with kinematics**
"theoretical points" was replaced by "detection method" (l. 329)

**l296 and l299: "associated to" -> associated with**
"associated to" was replaced by "associated with" (l. 336)

**l297: "first SBE Microcat's": please indicate the depth**
The depth range of the first Microcat's sensor was added (l. 337-338)

**l300-302: check the subjects and tense of your verb in this sentence.**
Thanks for pointing it out, the tense of the verb was corrected (l. 342).

**Figure 6: This is confusing: you have 3 case studies but 6 events?**
We see the point the reviewer raises here. Each 'case study' that we discuss is associated with one feature which can generate different events. We have explained the details of what we mean by case study more carefully to clarify this point at the beginning of the paragraph (l. 330-333).

**l309: It may be a matter of definition but should you not drop the term "eddy" in the case of A13? It looks more like a circulation. A13 is 600km long at least in the meridional direction …**
To address this point and other points that were raised, the definition of the eddy boundaries were changed to the location of their maximal azimuthal velocities. The size of A13 (now called A4, Figure 6d) was clearly reduced with this new definition.

**l316: is it at 450 m (text) or 500 m (Figure 7 caption)?**
We agree and we have carefully checked to make sure the depth of the sensors is referred to in the same way in both the text and the captions.

**Figure 8 and others: The issue with displaying the velocity vectors in these figures is that the**

**velocities at the SAMBA line are so much weaker than in the retroflection region. As a result, it does not help your case when arguing that eddies are present at the line. Could you find another way to display these arrows? Use two different scaling? or just show the direction of the velocity? Or show the velocities only near the SAMBA line so that you can adjust the scaling?**

Thanks for pointing it out. The area shown in Figures 8, 9 and 10 was reduced and the scale of the vectors increased, to highlight the velocity at the SAMBA line.

**Figure 9: A16 is an eddy? This is not credible.**

Even with the new definition of the eddy boundary that we are using (see our responses to earlier comments), the size of A16 (now called A5) is still very large (Figure 10). The eddy is detected by both detection methods that we are employing. The radius estimated is about 156 km with our new eddy detection method, and about 128 km in the Mesoscale Eddy Trajectory Atlas product at this date. We should point out that previous estimates of Agulhas rings in this region have been shown to have a diameter of as large as 400 km (*e.g.* Arhan et al, 1999), so the large eddy we are observing is not implausible.

**Figure 10: something strange is happening in the top left corner of panel a: please indicate what the contours are and the displayed gray numbers. It looks like something is wrong with the labeling of your contours. The caption needs to be improved. "temperature and salinity ... from the SBE37 MicroCATs colored dots with their associated depth" : what does this mean? Temperature from dots? Finally, I do not understand the difference between panel a on one hand and panels b and c on the other hand.**

The contours display in the top left corner had been fixed for this figure (now Figure 11). The caption now indicates that the gray contours and contour labels are the potential density relative to a reference pressure of 1000 dbar. The caption was improved to explain all the different features observed in this figure and the differences between each panel.

**Finally, I am not sure about the systematic use of "#" for #Anticyclonic and #Cyclonic. What is the point?**

We have eliminated the "#" symbols, as they were not essential.

**l371-372: "While the signature of these two features is clearly separated": in panel b I do not see the blue dots and red dots being clearly separated for the densest classes of water. Can you explain better?**

We agree, in the revised manuscript we now detail in the text at which sensors this distinction is evident (l. 421-422)

**l378: associated to -> associated with**

"associated to" was replaced by "associated with" (l. 428)

**l410: said -> written?**

This sentence was deleted (l. 463)

**l411: "correlated compared" -> correlated with?**

This sentence was deleted (l. 463)

**section 3.4: Could you conclude something about the analysis summarized in Figure 11: what have we learned? Is this section about comparing CPIES data and microcat data? Or is it about learning something about water intrusions? Looking at Figure 11, none of the anomalies from the**

**CPIES seem to match the anomalies from the microcats. What does this mean?**

Thanks for pointing this out. As explained in our earlier response to one of the general comments, we have updated Figure 11 (now Figure 12) to fix the results after discovering an error in our code, and the result is that we now find a better agreement between the CPIES-GEM and MicroCat data sets. Furthermore this figure aids in our discussion of isopycnal heave versus intrusion of new water masses in the case study events we highlight; the main results of this comparison are now highlighted and discussed in better detail in the text (l. 450-487, l. 576-581).

**l477 to 480: The two following statements are a little bit contradictory, can you try to reconcile? In the intro you talk a lot about nonlinear, then if your Rossby number estimates are small, the conditions are not that "nonlinear"? Or maybe you refer to the definition of nonlinear eddies by Chelton etc.? Can you please add a reference and discuss more.**

As explained our earlier reply to the general comment, in our new revised version we have clarified the definition of nonlinear mesoscale eddies that we are using, i.e. defined by the advective parameter of Chelton et al. (2011), in the Introduction (l. 31-32). We have also discussed more the implications of two different parameters on the dynamics of these features in Section 4 (l. 547-554).

We are very pleased that the reviewer sees the value in our manuscript. We have addressed their helpful comments in the revised draft. Below are our responses to the reviewer's specific comments: the comments from the reviewer are in bold, while our responses are interspersed between the comments in non-bold text. All line numbers indicated in our responses correspond to the new version of the paper.

**The revised manuscript is an improvement from the previous version. I appreciate the changes made by the authors, especially in the introduction and methods' sections. I find the manuscript interesting and I would recommend it for publication in Ocean Sciences. I have some relatively minor comments that I recommend to address before publication.**

**I would recommend that the authors attempt to present more strongly the validation of the altimetry data with the ADCP in situ observations. Showing the time series could be helpful to provide more faith in the validation analyses.**

This is an excellent suggestion, as our ADCP observations are strongly consistent with the altimetry observations. To address this suggestion, we have added a new figure (Figure 3) that shows the good correspondence between the ADCP and altimetry velocity time series. This visual representation provides additional confidence in our new eddy detection method. Moreover, to address a comment from one of the other reviewers, we have also used the Mesoscale Eddy Trajectory Atlas product, distributed by AVISO and based on the eddy detection method of Chelton et al. (2011), to verify the detection of all of the features we discuss in the paper. We feel that the results are now quite robust given consistent detection from two independent altimetry-based methods and solid comparison with the ADCP method of Lilly and Rhines.

**Specific Comments**

**Line 39: replace 'lives' with 'leaves' (is this what you meant?)**
"Lives" was replaced by "leaves" l. 35

**Line 45: remove 'a' before implications**
"a" was removed l. 41

**Line 85: 'On regional scales, the circulation is important for water mass distribution, local dynamics, ecosystem assessments and air-sea interactions.' This seems to be too generic, I suggest giving more details about the importance of your study for ecosystem assessments, for instance (just a few words).**
Thanks for pointing this out. We have provided some additional details by including an example of the implication of the mesoscale dynamics on the southern Benguela ecosystem (l. 93-95).

**Line 90: I suggest adding Chidichimo et al (2014, JPO), they looked at the water masses distributions and variability in Drake Passage. Donohue et al (2016, GRL) looked at the mean total ACC flow.**
As suggested, we added (l. 100) a citation to Chidichimo et al. (2004), which seemed more pertinent.

**Line 91: I suggest adding updated references about the arrays south of Africa (ASCA,**

**CROSSROADS, GOOD HOPE). For instance Swart et al (JGR 2008), Hutchinson et al (2016 JGR). Some of the authors here are the same as in Hutchinson et al, but it is not clear to me why this study is not mentioned.**

Our focus here is primarily within the South Atlantic basin, i.e. west of Africa, whereas most of the arrays south of Africa in this list are more explicitly focused on the Indian Ocean circulation. However we agree that there is some degree of interaction between the two regions, and we have added citations to Swart et al. (2008) and Hutchinson et al. (2016) and included a few words in the Introduction to discuss them (l. 101). The studies of Beal et al. (2015) and Elipot and Beal (2018) were added in Section 2 as they are pertinent as examples of where the GEM method has previously been applied to CPIES data within the broader region (l. 154).

**Line 161: '...in order to measure temperature and derive salinity and density.' ... this reads weird. The CTD measures conductivity, temperature and depth. Salinity is derived from the measured conductivity while density needs to be calculated from temperature and salinity.**

We agree, this sentence was rephrased (l. 180).

**Lines 170-171: Standard error has units, please check this sentence.**

The errors for the temperature and the absolute salinity were re-estimated and added in the manuscript with their respective units (l. 193).

**Lines 221-223: I am not convinced about the conclusions from the analysis of the degree of correspondence between the ADCP in situ data and altimetry, given the moderate correlations found, but I'll leave this to the editor's judgment.**

We agree with the reviewer than our conclusion was perhaps a bit too bold given the modest values of the correlation coefficient. The correlations are statistically different from zero given the number of degrees of freedom that we have in the records, but nevertheless we have addressed this comparison with more caution in the new version of the manuscript.

**Line 376: Please add 'record from the' after pressure**

"record from the" was added as suggested (l. 426)

**Title section 3.4: I think it is 'three' case studies based on the section above (I also recommend typing 'three' instead of '3' in the title)**

Yes, you are right, it is "three". We corrected the title as suggested (l. 431).

**Line 434; I think you refer to the SAMBA-east array 'region'?**

"region" was added, thanks (l. 497)

[revised manuscript text omitted]

---

## Author Response (AR3)

**Answer to Editor - Dr Chapman**

We are very pleased that the editor believes that our manuscript is now suitable for publication subject to minor corrections. Below we detail the changes we have made to the manuscript, tackling point by point the remaining minor comments that the editor asked us to address prior to acceptance. The original comments from the editor are below in bold, while our responses are interspersed between the comments in non-bold text. All line numbers indicated in our responses correspond to the new version of the paper.

**I thank the authors for their latest revision of the paper, which I believe is now suitable for publication. The reduction in the number of eddies from the original claim has certainly made interpretation easier, especially as the eddies remaining have been identified by two separate methods, which adds credence to their results. I have four very minor points that may need correcting:**

**1. In line 206, mention is made of the closed ADT contour lines having an increment of plus or minus 1 mm. Do you actually mean this, or should it be 1 cm? 1 mm seems very small given that the changes in SSH can be more than 20 cm in large eddies.**
It appears that the revised version of the manuscript was still not clear on this issue, our apologies. Yes, the increment applied in the eddy detection method is equal to 1 mm as in the study of Chaigneau et al. (2011). But, we understand the confusion of the editor, as the typical value for this criterion is usually 1 cm. Nevertheless, Faghmous et al. (2015) showed that using a finer threshold step than the typical value used in the literature leads to more accurate eddy sizes and amplitudes. This sentence was rephrased to explain in more details the choice of this threshold value (l. 205-209).

**2. In lines 311-312, the authors talk about the warm and cold intrusions. Yet given the color bar used for Fig. 6, it is actually quite hard to see this. Perhaps changing the wording slightly to point out that the intrusions are "relatively" warmer or colder may help.**
Thanks for pointing this out. We agree than the characterization of the intrusions was perhaps a bit too bold given their modest satellite temperature signature. In the revised version we have described these features more along the lines of what was suggested (l. 313-314).

**3. Line 427 references Fig 9c,d. I believe this should be Fig 10c,d.**
Yes, you are right, this was a typo. In the revised version "Fig 9" was replaced with "Fig10" (l. 429).

**4. Line 503 – the authors say that they found 16 anticyclonic eddies that were Agulhas rings. I assume this is left over from the earlier version of the manuscript as you are now only discussing 7 anticylones.**
We agree that our sentence was confusing. The total number of 16 Agulhas Rings is referring to the number of anticyclones that enter the Cape Basin crossing the C-line (Fig. 1). But you are right, only 7 anticyclonic eddies influenced the mooring measurements of the SAMBA-East line, which is the primary focus of our paper. These sentences have been rewritten in the revised version to clarify this point (l. 505-509).

[revised manuscript text omitted]